# Further evidence for CCN aerosol concentrations determining the height of warm rain and ice initiation in convective clouds over the Amazon basin

*Ramon Campos Braga[1], Daniel Rosenfeld[2], Ralf Weigel[3], Tina Jurkat[4], Meinrat O. Andreae[5,9], Manfred Wendisch[6], Ulrich Pöschl[5], Christiane Voigt[3,4], Christoph Mahnke[3,8], Stephan Borrmann[3,8], Rachel I. Albrecht[7], Sergej Molleker[8], Daniel A. Vila[1], Luiz A. T. Machado[1], and Lucas Grulich[10]*

[1] Centro de Previsão de Tempo e Estudos Climáticos, Instituto Nacional de Pesquisas Espaciais, Cachoeira Paulista, Brasil

[2] Institute of Earth Sciences, The Hebrew University of Jerusalem, Israel

[3] Institut für Physik der Atmosphäre, Johannes Gutenberg-Universität, Mainz, Germany

[4] Institut für Physik der Atmosphäre, Deutsches Zentrum für Luft- und Raumfahrt (DLR), Oberpfaffenhofen, Germany

[5] Multiphase Chemistry and Biogeochemistry Departments, Max Planck Institute for Chemistry, 55020 Mainz, Germany.

[6] Leipziger Institut für Meteorologie (LIM), Universität Leipzig, Stephanstr. 3, 04103 Leipzig, Deutschland

[7] Instituto de Astronomia, Geofísica e Ciências Atmosféricas, Universidade de São Paulo, Sao Paulo, Brazil

[8] Particle Chemistry Department, Max Planck Institute for Chemistry, 55020 Mainz, Germany.

[9] Scripps Institution of Oceanography, University of California San Diego, La Jolla, CA 92098, USA

[10] Institut für Informatik, Johannes Gutenberg-Universität, Mainz, Germany

*Correspondence to*: Ramon C. Braga (ramonbraga87@gmail.com)

**Abstract:** We have investigated how aerosols affect the height above cloud base of rain and ice hydrometeor initiation and the subsequent vertical evolution of cloud droplet size and number concentrations in growing convective cumulus. For this purpose we used in-situ data of hydrometeor size distributions measured with instruments mounted on HALO aircraft during the ACRIDICON-CHUVA campaign over the Amazon during September 2014. The results show that the height of rain initiation by collision and coalescence processes ($D_r$, in units of meters above cloud base) is linearly correlated with the number concentration of droplets ($N_d$ in cm$^{-3}$) nucleated at cloud base ($D_r \approx 5 \cdot N_d$). Additional cloud processes associated to $D_r$ such as GCCN, cloud and mixing with ambient air and other processes produce deviation of ~21% in the linear relationship, but it does not mask the clear relationship between $D_r$-$N_d$ which was also found at different regions around the globe (e.g. Israel and India). When $N_d$ exceeded values of about 1000 cm$^{-3}$, $D_r$ became greater than 5000 m, and the first observed precipitation particles were ice hydrometeors. Therefore, no liquid water raindrops were observed within growing convective cumulus during polluted conditions. Furthermore, also the formation of ice particles took place at higher altitudes in the clouds in polluted conditions, because the resulting smaller cloud droplets froze at colder temperatures compared to the larger drops in the unpolluted cases. The measured vertical profiles of droplet effective radius ($r_e$) were close to those estimated by assuming adiabatic conditions ($r_{ea}$), supporting the hypothesis that the entrainment and mixing of air into convective clouds is nearly inhomogeneous. Additional CCN activation on aerosol particles from biomass burning and air pollution reduced $r_e$ below $r_{ea}$, which further inhibited the formation of raindrops and ice particles and resulted in even higher altitudes for rain and ice initiation.

**1. Introduction**

Understanding cloud and precipitation forming processes and their impacts on the global energy budget and water cycle is crucial for meteorological modeling. Therefore, many studies have focused on improving cloud parameterization in numerical weather and climate models (e.g., Frey et al., 2011; Khain et al., 2005, 2000; Klein et al., 2009; Lee et al., 2007; Machado et al., 2014).

Cloud droplets form when humid air rises and becomes supersaturated with respect to water. Then water vapor condenses onto surfaces provided by pre-existing cloud condensation nuclei (CCN, a list of abbreviations and symbols is given in Table 1) aerosols. For ice formation, the ambient temperatures must reach values lower than 0 °C. At temperatures between 0 °C and -36 °C, ice in convective clouds mostly forms inhomogeneously on ice nuclei (IN) aerosols, often when they interact with supercooled liquid water droplets (Pruppacher et al., 1998). Ice multiplication is an important mechanism that masks the primary ice nucleation activity when cloud droplets are sufficiently large to promote also warm rain by coalescence, at the temperatures of -3 to -8 °C (Hallet and Mossop, 1974). At much colder temperatures (less than -37 °C), cloud particles freeze due to homogeneous ice nucleation (Rosenfeld and Woodley, 2000).

A cloud predominantly consists of droplets with diameters larger than about 3 μm, except for transient smaller sizes right at cloud base. The number concentration of cloud droplets ($N_d$ in cm$^{-3}$) at cloud base mainly depends on the conditions below cloud base, i.e., the updraft wind speed ($W$) and the supersaturation ($S$) activation spectra of cloud condensation nuclei [$CCN(S)$] (Twomey, 1959). In very clean conditions, values of $N_d$ near cloud base are in the range of ~50-100 cm$^{-3}$, while in polluted condition $N_d$ may reach values between 1000-2000 cm$^{-3}$ (Andreae, 2009; Rosenfeld et al., 2008, 2014a).

Below the freezing level, raindrops are formed due to cloud droplet coagulation (collision-coalescence) processes (warm rain process). Mixed phase precipitation results from interactions between ice particles and liquid water droplets (Pruppacher et al., 1998). Several studies based on aircraft, radar and satellite measurements support that warm rain

formation requires that the cloud consists of droplets with values of the effective radius ($r_e$) larger than 13-14 μm (Freud and Rosenfeld, 2012; Konwar et al., 2012; Prabha et al., 2011; Chen et al., 2008; VanZanten et al., 2005; Pinsky and Khain, 2002; Gerber, 1996; Rosenfeld and Gutman, 1994).

The effects of aerosol particles on clouds and precipitation have been studied in different parts of the globe (e.g., Fan et al., 2014; Li et al., 2011; Ramanathan et al., 2001; Rosenfeld and Woodley, 2000; Rosenfeld et al., 2014a; Tao et al., 2012; Voigt et al., 2017; Wendisch et al., 2016). A particularly interesting region is the Amazon basin, which presents contrasting environments of aerosol particle concentration between dry and wet seasons as well as steep aerosol concentration gradients within regions with near-constant thermodynamic conditions (Andreae et al., 2004; Artaxo et al., 2013). The background number concentrations of aerosol particles and CCN over the pristine parts of the Amazon region are about a factor of 10 times lower than those of polluted continental regions, including polluted conditions over the Amazon (Martin et al., 2016). During the dry-to-wet transition season in the Amazon region, total aerosol number concentrations reach values up to 10,000 cm$^{-3}$, mostly due to forest fires (Andreae, 2009; Andreae et al., 2012; Artaxo et al., 2002). On the other hand, in the rainy season aerosol number concentrations are about 500-1000 cm$^{-3}$ with CCN concentrations on the order of 200-300 cm$^{-3}$ for 1 % supersaturation, mainly consisting of forest biogenic aerosol particles (Artaxo, 2002; Martin et al., 2016; Pöhlker et al., 2016; Pöschl et al., 2010). Additionally, Manaus city, which is located at the central Amazon basin, releases significant concentrations of urban pollution aerosol particles (e.g., due to traffic, combustion-derived particles, or different types of industrial activities). This increases CCN concentrations by up to one order of magnitude (for 0.6% supersaturation) from the wet (Green Ocean) to the dry season (Kuhn et al., 2010).

Rosenfeld et al. (2012b) showed that by estimating the adiabatic number of droplets nucleated at cloud base ($N_a$), the height above cloud base at which the first raindrops evolve can be parameterized. This approach is based on the assumption that the entrainment and mixing of air into convective clouds is almost completely inhomogeneous (Beals et al., 2015; Burnet and Brenguier, 2007; Freud et al., 2011; Paluch, 1979). The inhomogeneous mixing occurs when evaporation rate of cloud droplets exceeds significantly the mixing rate of the cloud with ambient air. This causes the droplets that are at the boundary of the entrained air filament to evaporate completely and moisten that air until it is saturated. Further mixing of the saturated entrained air would not cause additional evaporation, but only decreasing of $N_d$ and LWC, while maintaining $r_e$ of the remaining droplets unaffected. This implies that the vertical profile of the actual cloud droplet effective radius behaves nearly as in an idealized adiabatic cloud. This connects uniquely the adiabatic drop number concentration, which is approximated by $N_a$ at cloud base, with the adiabatic droplet effective radius ($r_{ea}$), based on an adiabatic parcel model for which droplet growth is dominated by condensation (Freud and Rosenfeld, 2012; Pinsky and Khain, 2002). This parameterization can be applied to estimate the height above cloud base at which raindrops start to form, when $r_{ea}$ reaches 13 μm ($D_{13}$) [Freud and Rosenfeld, 2012; Konwar et al., 2012; Rosenfeld et al., 2012b; Prabha et al., 2011; VanZanten et al., 2005]. However, uncertainties associated to the calculated $N_a$ decrease the agreement between $r_{ea}$ and $r_e$. Most of these uncertainties arise when additional CCN activation of droplets happens above cloud base because the adiabatic model does not predict that $N_d$ increases with height, but decrease due to evaporation and deviations from inhomogeneous cloud mixing (Pinsky and Khain, 2012).

Braga et al. (2017) applied the methodology described by Freud et al. (2011) to calculate $N_a$ at the base of growing convective cumulus clouds for the Amazon region during the ACRIDICON-CHUVA (Aerosol, Cloud, Precipitation, and Radiation Interactions and Dynamics of Convective Cloud Systems)-CHUVA (Cloud processes of tHe main precipitation systems in Brazil: A contribUtion to cloud resolVing modeling and to the GPM [Global Precipitation Measurements]) campaign (Wendisch et al., 2016). The $N_a$ is calculated from $N_a = CWC_a / M_{va}$, where $CWC_a$ is the adiabatic cloud water content (CWC$_a$) as calculated from cloud base pressure and temperature, and $M_{va}$ is the adiabatic mean

volume droplet mass, as approximated from the actually measured mean volume droplet mass ($M_v$) by the cloud probe DSDs obtained during the cloud profiling measurements. Measurements of $M_v$ with height are considered only for cloud passes where CWC is greater than 25 % of the adiabatic CWC and $r_e$ is lower than 11 μm (i.e. for cloud droplets which have grown mostly via condensation). The calculated $N_a$ based on the measured vertical profile of $r_e$ agreed well (within 20-30 %) with the actual measurements of cloud droplet number concentrations at cloud base. This approach provides the opportunity to test the agreement between estimated $r_{ea}$ and the height above cloud base of warm rain initiation ($D_r$) within clouds for the Amazon region. In addition, measurements of the height above cloud base of ice initiation ($D_i$) in convective clouds are also available from flights that include cloud penetrations at ambient temperatures as low as -60 ºC with the HALO aircraft (Wendisch et al., 2016).

This study analyzes the vertical development of cloud and precipitation particles (water drops and ice crystals) in growing convective cumulus over the Amazon, based on measurements of cloud microphysical properties from instruments mounted on HALO during ACRIDICON-CHUVA (Wendisch et al., 2016). The vertical profile of $r_{ea}$ is used to estimate the depth above cloud base at which warm rain initiation occurs. The dominance of inhomogeneous mixing causes the $r_e$ profile to behave almost as in adiabatic clouds, constrained by $N_d$ at cloud base (Burnet and Brenguier, 2007; Freud et al., 2011). This means that the height above cloud base for reaching $r_e$ of 13-14 μm, which is required for rain initiation, is also determined by cloud base $N_d$ (Freud and Rosenfeld, 2012). Rain initiation depends strongly on $r_e$ because the rain production rate by collision and coalescence is proportional to $\sim r_e^5$ (Freud and Rosenfeld, 2012). Here we test and quantify these relationships for the measurements conducted with HALO aircraft during ACRIDICON-CHUVA.

The HALO flights during the ACRIDICON-CHUVA campaign were performed over the Amazon region under various conditions of aerosol concentrations and land cover (Wendisch et al., 2016). Figure 1a shows the flight tracks during which cloud profile sampling in growing convective cumulus was performed. Figure 1b shows a schematic sketch of the flight pattern while sampling cloud clusters (the locations in three dimensions of each flight are available at Figure 1 on supplementary material). The aircraft obtained a composite vertical profile by penetrating young and rising convective elements, typically some 100-300 m below their tops.

The cloud droplet size distributions (DSDs) between 3-50 μm diameter were measured at a temporal resolution of 1 second by the CAS-DPOL and CCP-CDP probes (Baumgardner et al., 2001; Lance et al., 2010; Brenguier et al., 2013). Each DSD spectrum represents 1 s of flight path (covering ~150 m of horizontal distance for a typical aircraft speed). The value of $r_e$ was calculated for each 1-s DSD. The two probes (CAS-DPOL and CCP-CDP) were mounted on opposite wings of HALO (horizontal distance of ~15 m). Similar values of $N_d$ and derived $r_e$ were measured by CAS-DPOL and CCP-CDP (they agree within 30 %), even though they were mounted on different wings. A previous study (Braga et al, 2017) showed that both probes were in agreement within the measurement uncertainties with respect to the measured cloud droplet number concentrations at cloud base and in accordance with the expected values for different conditions of CCN concentration and updraft wind speed below cloud base. In addition, the CWC calculated from the measured DSDs shows similar values to those measured with a hot wire device for different heights above cloud base [the probes' measurements agree within their uncertainty range (16% for probe DSDs and 30% for hot-wire device)] (Braga et al., 2017).

The determination of the height of rain initiation is based on the drizzle water content (DWC) calculation from the CCP-CIP probe (Brenguier et al., 2013). The DWC is defined as the mass of the drops integrated over the diameter range of 75–250 μm (Freud and Rosenfeld, 2012). This size range is selected because it includes only drops with terminal fall speed of 1 m s$^{-1}$ or less, which maximizes the chance that the drizzle was formed *in situ* and did not fall a

large distance from above. Rainwater content (RWC) is defined as the CCP-CIP integrated liquid water mass of droplets with diameters between 250 and 960 µm. The CCP-CIP images were used to distinguish raindrops and ice particles during cloud passes. The hydrometeor type is identified visually by their shapes. The phase of the smaller CCP-CIP particles cannot be distinguished. Therefore, the precipitation is considered as mixed phase when ice particles are identified, and the combined DWC and RWC are redefined as mixed phase water content (MPWC). Table 2 summarizes the calculated cloud microphysical properties with respect to the instrumentation used and its size ranges.

## 2. The scientific motivation

The aircraft based *in situ* measurements of cloud properties were collected within convective clouds formed over the Amazon from cloud base up to cloud top above the glaciated level. These measurements provided a unique opportunity to evaluate previous theoretical knowledge about aerosol impacts on convective clouds characteristics over the Amazon. In this study the impact of $N_a$ (adiabatic cloud drop concentrations) in determining the initiation of rain and ice within convective clouds is evaluated. This is performed through the analysis between the calculated $N_a$, $D_r$ and $D_i$ for several different environmental conditions over the Amazon (cloud base updrafts, aerosol concentration, surface cover etc.). The relationship of $N_a$ and $D_r$ was previously analyzed for regions of Israel and India where a linear relationship was found ($D_r \approx 4 \cdot N_a$) [Freud and Rosenfeld, 2012]. For Amazon region a similar analysis is performed here also taking in account the impact of $N_a$ in $D_i$. This is the first study which analyzes the impact of $N_a$ on $D_r$ and $D_i$ at Amazon region using *in situ* measurements of convective cloud properties. The results obtained from comparisons of $N_a$ estimates and the measured effective number of droplets nucleated at cloud base ($N_d*$), shown at Braga et al. (2017) for the same flights in the Amazon region, support the methodology of deriving $N_a$ based on the rate of $r_e$ growth with cloud depth, and under the assumption that the entrainment and mixing of air into convective clouds is extremely inhomogeneous. This is important because the characteristics of convective clouds based on $N_a$ values can be extended in space and time by their application to satellite-calculated $N_a$ (which is obtained with the same parameterization that has been recently developed from the satellite-retrieved vertical evolution of $r_e$ in convective clouds) [Rosenfeld et al., 2014b].

## 3. Instrumentation

### 3.1 Cloud particle measurements

The instrumentation used to measure cloud particles and rain or ice formation consists of three cloud probes: CAS-DPOL, CCP-CDP and CCP-CIP (Brenguier et al., 2013). In this study, cloud particle counts are accumulated for bin diameters larger than 3 µm from the CCP-CDP and CAS-DPOL; the lower size bins from these probes overlap with haze particles. Nucleated cloud drops in convective clouds grow quickly beyond 3 µm. Details about the cloud probe measurements characteristics are described in the following sub-sections and in Braga et al. (2017).

### 3.2.1 CCP-CDP and CCP-CIP measurements

The Cloud Combination Probe (CCP) combines two detectors, the Cloud Droplet Probe (CDP) and the greyscale Cloud Imaging Probe (CIPgs). The CDP detects forward scattered laser light of cloud particles when penetrating the CDP detection area (Lance et al., 2010). The CIP records 2-D shadow cast images of cloud elements. In this study, we deduced the existence of ice from the occurrence of visually non-spherical shapes of the shadows. The particle detection size range is 2 µm to 960 µm when measuring with the CCP at 1 Hz frequency (Wendisch et al., 2016). The combination of CCP-CDP and CCP-CIP information provides the ability to measure cloud droplets and raindrops within clouds for nearly the same air sample volume. The maximum number of particles measured by CCP-CDP and CCP-CIP

are about 2,000 and 500 cm$^{-3}$ for 1 Hz cloud pass, respectively. For the data processing of the CIP measurements, ice is assumed as the predominant particle phase in the mixed-state cloud conditions throughout the ACRIDICON-CHUVA campaign. The assumption of ice density instead of water density implies a slight overestimation (~10 %) of the calculated rain water content for particles greater than 75 µm. An additional data processing assuming water density as the predominant particle phase was performed for flights which warm rain was initiated below the 0 °C isotherm.

### 3.2.2 CAS-DPOL measurements

The CAS-DPOL measures particle size distributions between 0.5 and 50 µm at 1Hz time resolution (Baumgardner et al., 2011; Voigt et al., 2010; Voigt et al., 2011). Number concentrations are derived using the probe air speed measured at the instrument. Particle inter-arrival time analysis did not show influences of coincidence (Lance, 2012). The data analysis and uncertainties are described in detail in Braga et al. (2017).

Braga et al. (2017) have shown sufficient agreement between both CAS-DPOL and CCP-CDP measurements of cloud droplet number concentration to distinguish convective clouds that develop above clean vs. polluted regions during the ACRIDICON-CHUVA campaign. In addition, the CWC estimated by integration of the DSDs measured with both probes showed good agreement with hot wire CWC measurements (Braga et al., 2017).

### 3.3 Meteorological data

The HALO aircraft was equipped with a meteorological sensor system (BAsic HALO Measurement And Sensor System - BAHAMAS) located at the nose of the aircraft (Wendisch et al., 2016). The uncertainties for measurements of temperature, relative humidity and vertical wind speed are 0.5 K, 5 % and 0.3 m s$^{-1}$, respectively (Mallaun et al., 2015).

### 3.4 Aerosol measurements

Aerosol particle measurements were performed using the Passive Cavity Aerosol Spectrometer Probe 100X (PCASP-100X), which is an airborne optical spectrometer that measures aerosol particles in the 0.1 µm to 3 µm diameter range (Liu et al., 1992). The maximum number of particles measured by PCASP is about 3,000 cm$^{-3}$ for 1 Hz cloud pass. During ACRIDICON-CHUVA campaign PCASP was not operated with a heated inlet, thus, the measured aerosol particles below cloud base (about 200 m) can be larger than the original dry size due to swelling.

## 4. Methods

The analyses are performed along the following general steps:

a) The relationship between $r_e$ and the probability of drizzle is found. The value of $r_e$ is calculated from the size distributions measured by the CAS-DPOL and the CCP-CDP (two different values). DWC, RWC, and MPWC are obtained from the CCP-CIP data. The calculations of these cloud properties are detailed at section 4.1.

b) The $N_a$ at cloud base is estimated through the vertical profile of $r_e$. The calculation of $N_a$ is detailed at section 4.2.

c) The height of rain initiation based on the modeled adiabatic growth of $r_e$ with height is estimated for different aerosol condition as a function of estimated $N_a$. The value of $D_{13}$ is estimated as the cloud depth for which the adiabatic $r_e$ reaches 13 µm (as described also at section 4.2).

d) The extent of agreement between the directly measured $D_r$ within convective clouds and the estimated $D_{13}$ based on the assumption of adiabatic $r_e$ growth and on the measured $r_e$ is discussed at sections 5 and 6.

### 4.1. Estimation of $r_e$, rain and ice initiation

Rain is initiated during the warm phase of growing convective cumulus by intensification of the collision and coalescence (coagulation) processes with height. The efficiency of the process of droplet coalescence is determined by the collection kernel ($K$) of the droplets and their concentrations (Pruppacher et al., 1998). Freud and Rosenfeld (2012) have shown through model simulations and aircraft measurements that $K \propto r_v^{4.8}$, where $r_v$ is the mean volume radius obtained from the cloud probe DSDs in the absence of ice. $r_v$ is defined as follows:

$$r_v = \left( \frac{3\ CWC}{4\ \pi\ \rho\ N_d} \right)^{\frac{1}{3}} \qquad (1)$$

where $\rho$ is the water density (1 g cm$^{-3}$), CWC is in g m$^{-3}$, and $N_d$ is in cm$^{-3}$. The values are obtained from the 1-Hz data of droplet size distributions from the cloud probes. The calculation of CWC is performed separately with CAS-DPOL and CCP-CDP probe droplet concentrations as follows:

$$CWC = \frac{4\pi}{3} \rho \int N(r) r^3 \mathrm{d}r \qquad (2)$$

where $N$ is the droplet concentration and $r$ the droplet radius. The calculations of DWC, RWC, and MPWC are done in similar fashion to CWC but with different cloud probes and particle size ranges (see Table 2).

The definition of $r_e$ is:

$$r_e = \frac{\int N(r) r^3 dr}{\int N(r) r^2 dr} \qquad (3)$$

Freud and Rosenfeld (2012) showed that $r_v \approx 1.08 \cdot r_e$, depending on the droplet size distribution. Using this relationship, they derived $r_e$ from $r_v$ and showed that warm rain initiates within clouds when $r_e$ is about 13-14 µm (Klein et al., 2009; Rosenfeld and Gutman, 1994; Rosenfeld and Lensky, 1998; Rosenfeld et al., 2012a, 2014c).

Only measurements with CWC larger than 25% of the adiabatic water content are considered in order to exclude convectively diluted or dissipating clouds. It is assumed that rain (or ice) formation starts when calculated DWC exceeds 0.01 g m$^{-3}$ (Freud and Rosenfeld, 2012). For rain initiation in liquid phase the DWC threshold is ~10% greater due to the overestimation of DWC during CIP measurements in warm clouds (as stated at Section 3.2.1). The small terminal fall speed of the drizzle drops ($\leq 1$ m s$^{-1}$) allows to focus on in-situ rain (or ice) initiation while minimizing the amount of DSDs affected by rain drops fallen from above into the region of measurements. In addition, cloud passes with rain were eliminated when cloud tops were visibly much higher than the penetration level (> ~1000 m), based on the videos recorded by the HALO's cockpit forward-looking camera. However, cloud tops higher than few hundred meters above the penetration level occurred only rarely.

Table 3 shows the cloud depth above cloud base at which warm rain initiation ($D_r$) occurs (i.e., DWC > 0.01 g m$^{-3}$) for all flights as a function of estimated $N_a$. The $D_r$ is taken as the cloud depth for ice initiation ($D_i$) if ice particles are evident in the CCP-CIP images. Here, the $D_i$ is visually ascribed for sizes greater than ~ 0.25 mm and it does not mean that frozen smaller particles cannot be present. The assumption of water or ice density as the predominant particle phase on DWC calculation based on CCP-CIP probe did not impact $D_r$ and $D_i$ measured because the DWC threshold (i.e., DWC > 0.01 g m$^{-3}$) for warm rain or ice initiation was achieved at the same cloud depth for both particles densities. Additional details about the cloud profiling characteristics for each flight as the number of altitude levels sampled (NLS), highest cloud depth without raindrop ($D_{r-1}$) or ice particles ($D_{i-1}$) etc. are also available in Table 3. Furthermore, Appendix A discusses the uncertainty calculations of the estimated parameters of cloud properties.

The $N_a$ for the convective clusters is estimated based on the slope between the calculated adiabatic CWC (CWC$_a$) and the mean volume mass of the droplets ($M_v$), which is the mass of a water sphere having the radius $r_v$. $Mv$ is calculated for 1-s DSD measurements of CAS-DPOL and CCP-CDP for non-precipitating cloud passes (Braga et al., 2017). The underlying assumption is that the measured $r_v$ is approximating the adiabatic $r_v$ ($r_{va}$) due to the nearly inhomogeneous mixing behavior of the clouds with the ambient air (Beals et al., 2015). Therefore, the measured $M_v$ approximates the adiabatic $M_v$ ($M_{va}$, where $M_{va}$ = CWC$_a$ / $N_a$). This methodology does not account for cloud mixing losses from droplet evaporation of additional drop activation. Both incur an overestimation of $N_a$. It was found that the calculated $N_a$ based on the vertical profile of $r_e$ commonly overestimate the measured $N_a$ near cloud base by 30 % (Freud et al., 2011). Therefore, in calculating $N_a$ a factor of 0.7 is applied to $N_a$ estimates. Braga et al. (2017) have shown that this estimated $N_a$ was in a reasonably good agreement with the directly measured cloud base droplet number concentration, $N_d$, as obtained from the CCP-CDP and CAS-DPOL during ACRIDICON-CHUVA. Once $N_a$ is estimated, the adiabatic $r_e$ ($r_{ea}$) can be calculated based on a simple adiabatic parcel model where droplet growth is dominated by condensation (Pinsky and Khain, 2002), where $r_{ea}$ = 1.08· $r_{va}$. The value of $D_{13}$ is defined as the cloud depth for which $r_{ea}$ reaches 13 µm. The uncertainties calculations of cloud properties estimated from cloud probes were described in Braga et al. (2017). The uncertainties of $r_e$, $r_v$, $r_{ea}$, $r_{va}$ are about 10%, while for CWC and $M_v$ the uncertainties are about 30%. The $N_a$ calculation does not take into account the possibility of new nucleation above cloud base (Freud et al., 2011). Braga et al. (2017) have shown that the assumption of adiabatic growth of droplets via condensation from cloud base to higher levels within cloud can lead to an overestimation by ~20-30% of the number of droplets at cloud base when calculating $N_a$ in cases with additional droplet nucleation above cloud base.

The $N_a$ calculated for cloud base was used to classify clouds as having developed in clean, polluted, or very polluted regions. A clean cloud case was defined as $N_a < 500$ cm$^{-3}$, polluted for $500$ cm$^{-3} < N_a < 900$ cm$^{-3}$, and very polluted for $N_a > 900$ cm$^{-3}$. During ACRIDICON-CHUVA, a flight in clean clouds (AC19) was performed over the Atlantic Ocean. Clouds observed during flights over the northern Amazon were classified as polluted, mainly due to diluted smoke from biomass burning advected by long-range transport. This region represents the Amazon background condition for aerosol concentration during the dry season. Very polluted conditions were met over the Central Amazon, which was affected strongly by biomass burning over the Amazonian deforestation arc (southern Amazon).

## 5. Results

### 5.1 Threshold of $r_e$ for warm rain initiation

The values of $r_e$ derived from integrating the cloud probe DSDs were used to identify rain initiation. Some caution is required to eliminate possible bias resulting from peculiar shapes of the drop size spectrum. An $r_e$ value of 13-14 µm represents the rain initiation threshold for growing convective cumulus observed at different locations in the world, as long as there is no significant influence from giant CCN (GCCN; dry soluble diameter > 1 µm) (Freud and Rosenfeld, 2012).The presence of GCCN during cloud droplet formation at cloud base can lead to a faster formation of raindrops due to both, the rain embryo effect and the competition effect that reduces cloud base maximum supersaturation and consequently reduces $N_d$ (Rosenfeld, 2000; Segal et al., 2007). Such cases are very common over the ocean due to sea spray aerosols; there, the values of $r_e$ at which raindrops start to form are commonly smaller than the usual threshold of 13-14 µm (Freud and Rosenfeld, 2012). In our study the DSDs from flight AC19 performed over the Atlantic Ocean did not show a large drop tail near cloud base (see Figure 2 in the supplementary material). The cumulative sample volume

from CCP-CDP probe at cloud base was about 5.8 L$^{-1}$ for 176 s of measurements. The figure shows the scarcity of large cloud droplet (with diameters > 20 µm) near cloud base, where the mean concentration of such droplets is smaller than 0.1 drop cm$^{-3}$. Such small concentration of large droplets at cloud base is insufficient to have any significant effect on supersaturation.

Figures 2a-b show the precipitation initiation probability as a function of $r_e$ calculated from the CCP-CDP and CAS-DPOL probes for all flights analyzed over the Amazon. The probability of precipitation is the fraction of 1-Hz in-cloud measurements which exceed certain DWC thresholds (i.e. for DWC > 0.01 g m$^{-3}$). This was done as a function of $r_e$ value. These figures show that for the CCP-CDP probe rain initiation is expected to occur at $r_e$ > 13 µm, whilst for CAS-DPOL the rain initiation threshold is $r_e$ > 12 µm. Difference of the two instruments in the $r_e$ range below ~7 µm and above ~11 µm have been discussed in Braga et al. (2017). For $r_e$ < 7 µm, they are related to a higher sensitivity of the CAS-DPOL for small cloud and aerosol particles, whereas for $r_e$ > 11 µm CAS-DPOL has lower sensitivity to large particles than CCP-CDP; however the differences are not significant within the uncertainties of the measurements. Because the CCP-CDP was mounted very close to the CCP-CIP, results from this probe are shown in subsequent sections; similar results were found from data collected with the CAS-DPOL probe.

### 5.2 Comparing estimated $r_{ea}$ with measured $r_e$

The comparison between the values of $r_{ea}$ (calculated from the estimated $N_a$ at cloud base described in Section 4.2) with the measured $r_e$ is the basis for analyzing the evolution of cloud particle size until rain or glaciation initiation occurs within the cloud. Rosenfeld et al. (2012b) showed that a tight relationship between the $N_a$ calculated for cloud base and the evolution of $r_{ea}$ with height ($r_{ea}$-$D_c$) provides a useful proxy of the depth in convective clouds at which raindrops start to form.

### 5.2.1 Case study: Flight AC07 over the Amazon deforestation arc

Flight AC07 was performed over the deforestation arc (see Figure 1a). Figure 3 shows the number of droplets measured at different heights in the convective clouds. Droplet concentrations reaching ~2000 cm$^{-3}$ were measured at cloud base, which is characteristic for very polluted clouds. The cloud base was located at about 1900 m above sea level, with ambient air temperature at about 15°C. Figure 4a shows the mean DSD for a cloud penetration at cloud base. It emphasizes the higher number concentration of small droplets (< 10 µm) that are observed in convective clouds forming in polluted environments. Figure 4b shows the evolution of $r_e$ measurements and estimated $r_{ea}$ as a function of temperature. The figure also shows that the values of $r_e$ do not exceed the 13 µm threshold at warm temperatures. These results suggest that cloud droplets formed at cloud base grow mainly via condensation and no raindrops were formed during the warm phase of convective cloud development. However, to rule out coalescence processes as a possible reason for droplet growth, further analysis using CCP-CIP images was done.

Figures 5a-c show the evolution of DSD and CWC (mean values) as a function of height above cloud base and the cloud particle images from the CCP-CIP. Figure 5a plots the data for a cloud pass at warm temperatures and Figures 5b-c result from measurements during cloud passes at cold temperatures. The DSDs show that most droplets have a diameter smaller than 20 µm, and only very few large droplets are observed for warm temperatures. The CCP-CIP detected only cloud droplets and no raindrops, as evident by both RWC and DWC < 0.01 gm$^{-3}$. At cold temperatures, the CCP-CIP images show the irregular shapes of large ice particles. No spherical raindrop shapes were found in these data for any of the cloud passes, including those collected at warm temperatures. The DWC and RWC calculated from the mean DSDs show values greater than zero only when ice particles were observed on the CCP-CIP images. Also, for a cumulative sample volume of 1.24 m$^{-3}$ from 89 s of CCP-CIP measurements, no raindrop were observed between the

heights above cloud base of 2,900 m (0ºC) and 7,100 m (-26.25 ºC). This means that the raindrop concentration, if any, was smaller than 1 drop m$^{-3}$. This is a negligible rain rate, and supports the notion of practical shut of coalescence. Furthermore, the CCP-CIP did not detect any raindrops at lower levels (warm temperatures) for a cumulative sample volume of 5.9 m$^{-3}$ from 426 s of measurements. These results indicate a strong inhibition of raindrop formation within growing convective cumulus for this flight over the deforestation arc of the Amazon. Even though some of the indicated effective radii values are larger than 13 μm for colder temperatures, these values do not indicate rain formation when only ice particles are observed. This does not exclude the possibility that small raindrops froze soon after their formation in such low temperatures.

The mean DSD and CIP images shown in Figure 5c result from a passage through a convective cloud with lightning activity. Figure 6 shows a photo of the cloud taken from the HALO cockpit just before the cloud penetration. The CCP-CIP has imaged graupel in this case. The presence of these type of ice particles within convective clouds is very common in thunderstorms, and previous studies highlight the large frequency of lightning occurrence during the dry-to-wet season over the deforestation arc region of the Amazon (Albrecht et al., 2011; Williams et al., 2002). These results also highlight the role of aerosols from biomass burning on warm rain inhibition and on the aerosol invigoration effect due to the generation of large ice particles and lightning (Rosenfeld et al., 2008).

Regarding the values of $r_e$ as a function of $D_c$, Figure 7a shows the estimated $r_{ea}$ (calculated from the adiabatic CWC shown at Figure 7b) and measured $r_e$. The figure shows that the estimated values for $r_{ea}$ are close to the $r_e$ measurements for convective cloud passes at different $D_c$. Even though no raindrops were observed in the convective cloud, the figure shows similar values of $r_{ea}$ and measured $r_e$ (with $r_{ea}$ slightly larger) as a function of $D_c$.

*5.2.2 Results of analysis of $r_e$ and $D_c$ in clean and polluted regions*

*- Clean region*

Figure 8a shows the measured $N_d$ of a convective cluster over the Atlantic Ocean off the Brazilian coast (flight AC19). This region was classified as clean because $N_a$ is about 300 cm$^{-3}$ (see Table 3). The cloud base was located at 600 m above sea level at a temperature of 22 ºC. Given the clean conditions over the ocean, the high relative humidity at surface level and the low concentration of CCN lead to the formation of large droplets already close to cloud base. Figure 8b shows the estimated $r_{ea}$ and the measured $r_e$ as a function of $D_c$. Several cloud passes showed large droplets with $r_e$ ~ 13 μm at only 1660 m above cloud base. Figures 9a-b show the DSDs and CCP-CIP images for the cloud passes at the height where rain starts to form and at the greatest height measured above cloud base, respectively. Figure 9a shows that rain is initiated (DWC > 0.01 g m$^{-3}$) already when the droplets become larger than about $r_e$ > 12 μm. This is probably due to the presence of GCCN over this maritime region.

Figure 10 shows the mean aerosol particle size distribution (PSD), as measured by the PCASP, just below cloud base for clean, polluted, and very polluted regions. The mean total number concentration of aerosol particles with sizes larger than 0.1 μm is about 1000 cm$^{-3}$ over the Atlantic Ocean, whilst for polluted (very polluted) case this value is about three (ten) times larger. In addition, the mean total number concentration of particles measured by the CCP-CDP show concentration ten times greater for particles larger than 10 μm over the ocean in comparison with inland Amazon region. This figure indicates the presence of large aerosols particles with sizes greater than 1 μm (possibly GCCN) over the ocean. When it nucleates droplets, this type of aerosol accelerates the growth of droplets during the warm phase leading to a faster formation of raindrops than predicted by the adiabatic parcel model. About 3500 m above cloud base, large raindrops are observed in the CCP-CIP images (see Figure 9b). The low CWC indicates that most of it was already converted into raindrops. These results highlight that under clean conditions, raindrops were formed mainly by warm

phase processes of cloud development. Even if the convective clouds reach colder temperatures, the low remaining amount of cloud water reduced a key ingredient for cloud electrification.

Before raindrops start to form ($Dc$ ~1,660 m) updrafts were observed with most values < 4 m s$^{-1}$, and when rain starts downdrafts starts to be evident (see Figure 3g at supplementary material). The values of vertical velocities measured at flight AC19 (clean region) were smaller than measured for flight AC07 (very polluted region). However, for both cases updrafts are more evident during droplets growth via condensation and downdrafts are stronger when precipitation particles are observed in the cloud. Strong updrafts (~10 m s$^{-1}$) are observed in polluted cases after ice starts to form (see Figure 3a at supplementary material), probably due to the latent heat release during freezing processes. An alternative explanation of updraft enhancement due to environmental conditions in these cases cannot be excluded.

*-Polluted regions*

The flights AC09 and AC18 were classified as polluted (see Table 3). These flights were performed over the northern Amazon region (see Figure 1a). Figure 11a shows the measured $N_d$ from flight AC09. The cloud base was located about 1200 m above sea level at a temperature of 19.5 ºC. Figure 11b shows the estimated $r_{ea}$ and the measured $r_e$ as a function of $D_c$. Values of $r_e > 13$ μm were observed for temperatures around 0 ºC, indicating the possibility of rain starting at this height. Figure 12a-b shows the DSDs and CCP-CIP images from flight AC09 at the height where rain starts to form ($D_r$ ~3000 m) and at the greatest height with measurements above cloud base. The CIP image at Figure 12b shows the first pass in which ice hydrometeors are observed mixed with supercooled rain drops. For lower levels only raindrops were observed. For flight AC18 cloud base was located about 1700 m above sea level at a temperature of 17 ºC, and rain started to form in convective clouds when $D_r$ ~3800 m. The measured $N_d$ and the estimated $r_{ea}$ and measured $r_e$ as a function of $D_c$ from flight AC18 are shown at Figures 4a-b at supplementary material. Figure 5a at the supplementary material shows that first rain drops in AC18 are observed at the -5.7 ºC isotherm, and that they still remain liquid, or at least spherical, at the -11.4 ºC isotherm (see Figure 5b at the supplementary material). Larger raindrops and a high amount of DWC were observed on AC09 for warmer temperatures than on flight AC18 (not shown). These results show that differences in cloud particle formation are associated with the $D_c$ at which convective clouds start to form raindrops or ice, defined earlier as $D_r$ and $D_i$. Flight AC18 has a droplet concentration, $N_d$, of up to 100 cm$^{-3}$ greater than the measurements during AC09 (see Figure 4a at supplementary material). With higher $N_d$ at cloud base, droplet growth via condensation in convective clouds is a less pronounced function of height due to the water vapor competition between droplets. Under these conditions, the collision and coalescence process and freezing of droplets are initiated at higher $D_c$ (Freud and Rosenfeld, 2012; Rosenfeld et al., 2008). For this the reason, the formation of raindrops and ice particles on flight AC09 starts at lower $D_c$ than on flight AC18 (assuming non-significant additional CCN activation above cloud base).

*-Very polluted regions*

Five flights were classified as very polluted (see Table 3): AC07, AC08, AC12, AC13, and AC20. The microphysical analysis of the measurements collected in growing convective cumulus during flight AC07 was already presented in Section 5.2.1. Figure 13a show the measured $N_d$ from flight AC13, which was made in the same region as flight AC07. The figure shows that the values of $N_d$ near cloud base on flight AC13 reach 2000 cm$^{-3}$, similar to AC07. However, the rate of decrease of $N_d$ with height above cloud base is much smaller in AC13 compared to AC07. During flight AC13 the measurements of large updrafts (which increase supersaturation and induce new droplets activation) and large aerosol concentration above cloud base suggest the occurrence of additional CCN activation leading to the observed relative increase of $N_d$ with height. This is supported by the fact that the observed $r_e$ are smaller than the calculated $r_{ea}$, as shown

in Figure 13b. Only values below 13 μm are observed (maximum of 12 μm), indicating the suppression of raindrop formation. Indeed, no raindrops were observed in the CCP-CIP images from growing convective cumulus passes on this flight, and only cloud droplets and ice particles were detected. Figure 14 shows the DSD and CCP-CIP images at the start of glaciation ($D_i$ ~4800 m). These results highlight the role of aerosols in inhibition of raindrop formation due to inducing a larger $N_d$ and respective lower $r_e$, which leads to suppression of collision and coalescence processes in very polluted regions.

The measured $N_d$ during flights AC08, AC12, and AC20 was greater above cloud base than at cloud base on several cloud passes (especially in flights AC08 and AC20; see Figures 6 and 7 in the supplementary material for these flights). In these flights the estimated $r_{ea}$ values were larger than the measured $r_e$ as a function of $D_c$ and strong updrafts (up to 15 m s$^{-1}$) were observed above cloud base (see Figures 3b,d and h in the supplementary material). The acceleration of updrafts above the height of cloud base increase supersaturation and thus can induce additional CCN activation. For flights which we observed the increase of $N_d$ with height, high aerosol concentration was observed indicating additional CCN activation. During these flights, cloud profiling was performed up to $D_c$ ~3500 m, and the values of measured $r_e$ were smaller than 13 μm, indicating the suppression of raindrop formation. The analysis of the data from the cloud probe DSDs and CCP-CIP images indicates that indeed no raindrops were present on these flights (not shown). The measurements from AC07 and AC13 over very polluted regions in the Amazon suggest that no raindrops are formed in growing convective clouds under these conditions. Instead, large precipitation particles are formed at cold temperatures in the form of ice. The $D_c$ at which these ice particles are formed depends on the size of the cloud droplets ($r_e$) at colder temperatures (larger droplets freeze earlier or at lower $D_c$) [Pruppacher et al., 1998]. This was previously documented by satellite retrievals (Rosenfeld et al., 2011), where glaciation temperatures of convective clouds were strongly dependent on $r_e$ at the -5 °C isotherm, where smaller $r_e$ were correlated with lower glaciation temperatures.

**6. Discussion**

The results from cloud probe measurements under clean, polluted, and very polluted conditions highlight the role of aerosol particles in rain and ice formation for growing convective cumulus. Figure 15 summarizes the estimated depths above cloud base at which initiation of rain and ice formation is observed ($D_r$ and $D_i$), as well as the estimated $D_c$ for rain initiation as indicated from $r_{ea}$ by $D_{13}$. This figure shows a close relationship between $N_a$ and $D_r$ of $D_r =$ (5±1.06)·$N_a$, demonstrating the capability to predict the minimum height at which raindrops are expected to form based on cloud base drop concentrations. For flights in which rain was observed (AC19, AC18, and AC09), $D_r$ occurs at heights slightly greater than $D_{13}$. For cases where neither rain nor ice were observed (AC08, AC12, and AC20), the estimated $D_{13}$ was not reached during the HALO cloud profiling flights. In addition, $D_{13}$ and $D_i$ show similar values for flight AC07, whereas for flight AC13 the values are less comparable (probably due to an overestimation of $N_a$, and thus $D_{13}$, caused by additional CCN activation above cloud base).

The linear relationship between $N_a$ and $D_{13}$ indicates a regression slope of about 5 m (cm$^{-3}$)$^{-1}$ between $D_{13}$ and the calculated $N_a$ for the Amazon during the dry-to-wet season. This value is slightly larger than the values observed by Freud and Rosenfeld (2012) for other locations around the globe (e.g., India and Israel). These clear linear relationships found between $N_a$ and $D_r$ (~$D_{13}$) for different regions highlight the efficiency of the adiabatic parcel model to estimate the height of rain initiation within convective clouds in this study. Additional cloud processes associated such as GCCN, cloud and mixing with ambient air and other processes, which are not accounted for in this study, would produce deviations that are likely to be the cause of the observed scatter in the results.

For the flight in cleanest conditions (AC19), the presence of larger aerosol particles (possibly GCCN from sea spray) below cloud base leads to a faster growth of cloud droplets via condensation with height, and consequently $r_e$ is smaller than 13 μm (see Figure 9a) for warm rain initiation. A similar decrease of $r_e$ for rain initiation over ocean was observed by Konwar et al. (2012). While $D_r$ is explained by $N_d$ and well correlated with it, there is no correlation between $N_d$ and $D_i$.

Figure 16 illustrates the vertical development of precipitation water content by symbols representing the amount of DWC and MPWC as a function of $D_c$ and CDP-measured $r_e$. Also shown are the lines of $r_{ea}$ as a function of $D_c$. The figure shows that raindrops began to form at $r_e$ of 13 μm for AC09 and AC18. The $r_e$ for rain initiation is slightly smaller (12 μm) on AC19; probably due to the sea spray giant CCN, which accelerate the coalescence for a given $r_e$. Mixed phase precipitation was initiated on flights AC07 and AC13, well below the height of $D_{13}$ at an $r_e$ of 11.5 and 10.2 μm, respectively. Ice starts to form at lower temperatures when the cloud droplets are smaller, as manifested by $D_i$ of -9 and -14 °C for flights AC07 and AC13, respectively. The remaining flights did not reach the height for rain initiation (AC08, AC12, and AC20).

It is evident that raindrops form faster via collision and coalescence process in a cleaner atmosphere. For the polluted cases, raindrops form at colder temperatures (~0 °C and colder) via collision and coalescence than for clean conditions. Rain can initiate at supercooled temperatures, e,g., -5 °C on AC18. The raindrops were documented to start freezing at -9 °C in AC09. In very polluted conditions, only cloud droplets, but no raindrops were observed at $D_c < 4000$ m. In these cases, precipitation was initiated as ice particles at $D_c > 4000$ m. These flights with completely suppressed warm rain were performed over the smoky deforestation arc. Measurements of large updrafts which increase supersaturation within cloud and the higher $N_d$ above cloud base indicate new activation of cloud droplets for flight AC13 (not observed at AC07) in the course of the development of convective cumulus. This additional CCN activation leads to smaller $r_e$. For flights where additional CCN activation was significant, the differences between the estimated $r_{ea}$ and the $r_e$ measurements at same height are larger, because the adiabatic estimation does not consider the additional CCN activation of droplets above cloud base and thus overestimates the observed size.

Figure 17 summarizes the findings from the vertical profiling flights. It illustrates the vertical microstructure of growing cumulus above the Amazon and the adjacent ocean in varying aerosol conditions. The figure highlights the differences of aerosol concentrations and cloud particles distribution within convective clouds over the Amazon basin (including the Atlantic Ocean, forested and deforested regions). The aerosol concentration is smaller over the Atlantic Ocean and increase significantly at continental Amazon, especially over the deforestation arc due to biomass burning emissions from forest fires. As polluted is the atmosphere, larger is the number of droplets nucleated at cloud base and less efficient is the growth of cloud droplets via condensation with $D_c$. The new activation of CCN above cloud base also has shown to decrease the efficiency of cloud droplets growth due to the higher competition for water vapor available. The increase of aerosol concentration over the Amazon basin according to our findings has shown to suppress the warm rain formation because larger cloud depths were necessary to raindrop starts to form (when cloud droplets have $r_e$ ~13-14 μm). The additional aerosol concentrations observed at polluted regions from forest fires suppress rain such that most of hydrometeors are ice when they are at a size that allows distinguish their phase (~0.25 mm). In addition, the formation of ice particles was also delayed (or occurred at higher $D_c$) in more polluted atmosphere, because smaller cloud droplets freeze at lower temperatures.

## 7. Conclusions

This study focused on the effects of aerosol particle number concentration on the initiation of rain drops and ice hydrometeors in growing convective cumulus over the Amazon. Data from aerosol and cloud probes on board of the

HALO aircraft were used in the analysis. The values of the estimated $N_a$ at cloud base were applied to classify the atmospheric conditions where convective clouds developed as a function of aerosol particle number concentration (i.e., clean, polluted, and very polluted regions). From the estimated $N_a$, the evolution of $r_{ea}$, the theoretical $r_e$ assuming adiabatic growth of droplets, with cloud depth above cloud base ($D_c$) were compared with the observed $r_e$ at the various heights. A DWC value of 0.01 g m$^{-3}$ was used as a threshold for rain initiation or glaciation within clouds. Images from the CCP-CIP probe were used to detect the presence of raindrops and/or ice hydrometeors. The results shown in previous sections support the following conclusions:

1. The use of $r_e \sim$ 13-14 µm as a threshold for rain initiation is suitable for convective clouds formed at the Amazon basin during the dry season. It is in agreement with $r_e$ of rain initiation elsewhere.

2. The evolution of the directly observed $r_e$ follows generally that of the calculated $r_{ea}$ due to the nearly inhomogeneous mixing behavior of the convective clouds with the ambient air. Convective clouds are usually non-adiabatic systems because of strong wind/turbulence effects, heating and other factors, but the similarities of $r_e$ and $r_{ea}$ provided the capability to estimate $D_r$ over the Amazon and others regions around the globe (e.g. India and Israel).

3. Rain initiation occurred higher in more polluted clouds, as manifested by higher $D_c$. Rain was initiated at supercooled levels in moderately polluted clouds. In very polluted conditions, warm rain was suppressed completely. This was exacerbated by the occurrence of additional CCN activation above cloud base, which further reduced $r_e$ compared to $r_{ea}$.

4. The initiation of ice hydrometeors is also delayed to greater $D_c$ in more polluted clouds, because smaller drops freeze at colder temperatures due to suppressed ice multiplication processes (Hallett and Mossop, 1974). Ice was initiated mostly by freezing raindrops in cases when warm rain formation was not completely suppressed.

5. Both the $D_{13}$ and $D_r$ increased linearly with $N_a$, in agreement with the theoretical considerations of Freud and Rosenfeld (2012). Despite the suspected occasional additional CCN activation, $r_e$ was sufficiently close to $r_{ea}$ to allow a linear relationship in the form of $D_r = (5\pm1.06) \cdot N_a$. The deviation from exact linear relationship can be associated to additional cloud processes such as GCCN, cloud and mixing with ambient air etc. The magnitude of these additional processes is insufficient to mask the linear relationship. The observations suggest also that, in the absence of new droplet activation above cloud base, $D_{13}$ is very similar to $D_i$ under very polluted conditions, where raindrops are not formed at warmer temperatures.

These results show that even moderate amounts of smoke, which fill most of the Amazon basin during the drier season, are sufficient to suppress warm rain and elevate its initiation to above the 0 ℃ isotherm level. This results in a suppression of rain from small clouds and an invigoration in the deep clouds, as hypothesized by Rosenfeld et al. (2008). While the net effect on rainfall amount is unknown, the redistribution of rain intensities and the resulting vertical latent heating profiles are likely to affect the regional hydrological cycle in ways that need to be studied further.

**Acknowledgements**

The first two authors of this study were supported by project BACCHUS European Commission FP7-603445. The generous support of the ACRIDICON-CHUVA campaign by the Max Planck Society, the German Aerospace Center (DLR), FAPESP (São Paulo Research Foundation), and the German Science Foundation (Deutsche Forschungsgemeinschaft, DFG) within the DFG Priority Program (SPP 1294) "Atmospheric and Earth System Research with the Research Aircraft HALO (High Altitude and Long Range Research Aircraft)" is greatly appreciated. This study was also supported by EU Project HAIC under FP7-AAT-2012-3.5.1-1. C. Mahnke and R. Weigel received funding by the German Fed-

eral Ministry of Education and Research (BMBF, Bundesministerium für Bildung und Forschung) within the joint RO-MIC-project SPITFIRE (01LG1205A). In addition, the German Science Foundation within DFG SPP 1294 HALO by contract no VO1504/4-1 and contract no JU 3059/1-1 contribute to support this study. The first author also acknowledges the financial support from the Brazilian funding agencies CAPES and CNPq during his Ph.D. studies.

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

**Appendix – A**

*Cloud properties uncertainties*

The uncertainties calculations of cloud properties estimated from CCP-CDP probe were described in Braga et al. (2017). The uncertainties of $r_e$, $r_v$, $r_{ea}$, $r_{va}$ are $\sim 10\%$, while for CWC and $M_v$ the uncertainties are about 30%. The calculation of $N_a$ uncertainty is described below.

*$N_a$ uncertainty*

The uncertainty of $N_a$ is calculated based on the uncertainty of slope between CWCa and $M_{va}$. The two maximum/minimum acceptable slope lines of $N_a$ can be used to estimate the uncertainty of the $N_a$ of the best fit line. The principle behind this is that if we were to take another complete set of data, we would find a new best fit slope. The maximum amount by which it is likely to differ from our existing best fit slope is about half the difference of the maximum and minimum acceptable slopes that we have. So this can be used as an uncertainty estimate:

$$Slope\ uncertainty = \frac{[(maximum\ slope)-(minimum\ slope)]}{2} \qquad (A.1)$$

The absolute values of $N_a$ uncertainty is shown at Table 3. The relative uncertainty of $N_a$ values in mean terms is ~21 % for all flights analyzed.

 **Figure captions**

Figure 1 a) HALO flight tracks during the ACRIDICON-CHUVA experiment. The flight number is indicated at the bottom by colors; b) Flight patterns below and in convective clouds during the ACRIDICON-CHUVA campaign.

Figure 2 a) Precipitation probability as a function of $r_e$ for the CCP-CDP probe for different DWC thresholds (black – DWC > 0.01 g m$^{-3}$; blue – DWC > 0.02 g m$^{-3}$; green – DWC > 0.03 g m$^{-3}$; gold – DWC > 0.05 g m$^{-3}$; red – DWC > 0.1 g m$^{-3}$). The dashed line indicates the number of cases (in seconds for each 1-s cloud pass) for each $r_e$ size interval (right axis); b) Similar for the CAS-DPOL probe.

Figure 3 Cloud droplet concentration measured with CCP-CDP as a function of temperature for flight AC07. Each dot indicates a 1-Hz average concentration. The sample number (N) and the approximate start time of the cloud profile are shown at the top of the panel.

Figure 4 a) Mean cloud droplet size distribution calculated from the CCP-CDP data for a cloud pass at cloud base during flight AC07. The flight number, initial time of cloud pass, and duration in seconds are shown at the top of graph. The mean total number of droplets ($N_{dmean}$), the maximum total number of droplets ($N_{dmax}$) in one second for this cloud pass, and the approximate height (H) and temperature (T) are shown at the upper-right corner of the graph; b) Cloud droplet effective radius ($r_e$) calculated from CCP-CDP as a function of temperature indicated with dots. The black line indicates the estimated adiabatic effective radius ($r_{ea}$) as a function of temperature.

Figure 5a-c. Droplet size distribution composite from the CCP-CDP and CCP-CIP probes (left panel). Similar for indicated cloud water content (CWC) in the right panel. Indicated at the top of the panels are the HALO flight number, date, time of flight (UTC), duration of cloud pass in seconds, temperature (T) and altitude (H) above sea level, and the mean values for the total number of droplets ($N_d$), CWC, DWC, RWC, and $r_e$. The color bars indicate the height of HALO during the cloud pass. On the right side of the panels CCP-CIP images corresponding to the cloud pass are shown.

Figure 6 Image taken from the HALO cockpit just before the aircraft penetration of a convective cloud with lightning activity during flight AC07. In this case, the cloud pass height was 9,022 m (temperature ~ -25 ℃) and the maximum CWC measured was 0.55 g m$^{-3}$.

Figure 7 a) Cloud droplet effective radius ($r_e$) as a function of cloud depth ($D_c$) for flight AC07. The line indicates the $r_e$ estimated for adiabatic growth ($r_{ea}$) from cloud base (dashed lines indicate the $r_{ea}$ values considering the uncertainty of the estimate). The height of 0 ℃ is indicated by a black horizontal bar across the $r_{ea}$ line. The estimated adiabatic number of droplets ($N_a$) at cloud base is shown at the top of the figure. b) Similar to a) for Cloud water content (adiabatic values are shown by lines).

Figure 8 a) Cloud droplet concentrations measured with the CCP-CDP as a function of temperature for flight AC19. Each dot indicates 1Hz average concentration. The sample number in seconds (N) and the start time of the cloud profile are shown at the top of the panel; b) Similar to Figure 7 for flight AC19.

Figures 9 a-b) Similar to Figures 5a-c for flight AC19.

Figure 10 Cumulative aerosol size distribution below cloud base calculated from the PCASP probe for typical clean, polluted, and very polluted regions (solid line) for flights AC12 (very polluted), AC18 (polluted), and AC19 (clean). Similar for cumulative cloud droplet size distribution calculated with CCP-CDP (dashed line). The flight numbers are indicated by colors at the top of the panel.

Figure 11 a) Cloud droplet concentrations measured with the CCP-CDP as a function of temperature for flight AC09. Each dot indicates 1-Hz average concentration. The sample number in seconds (N) and the start time of the cloud profile are shown at the top of the panel; b) Similar to Figure 9 for flight AC09.

Figures 12 a-b) Similar to Figures 5a-c for flight AC09.

Figure 13 a) Cloud droplet concentration measured with the CCP-CDP probe as a function of temperature for flight AC13. Each dot indicates a 1-Hz average concentration. The sample number and the approximate time of the cloud profile are shown at the top of the panel; b) Similar to figure 7 for Flight AC13.

Figures 14 Similar to Figures 5a-c for flight AC13.

Figure 15 Cloud depth ($D_c$) as a function of the estimated adiabatic number of droplets ($N_a$) at cloud base. $D_c$ for adiabatic cloud droplet effective radius ($r_{ea}$) equal 13 μm (or $D_{13}$) are indicated by triangles. Similar for cloud depth of rain initiation ($D_r$) [indicated by circles] and cloud depth for ice initiation ($D_i$) [indicated by asterisk]. The flight numbers are indicated by colors on the right side of the panel. The values of $D_{13}$, $D_r$, and $D_i$ are shown in Table 1. The uncertainties of $N_a$ estimates are shown by horizontal error bars. The vertical error bars indicate the cloud depth between $D_r$ and $D_{r-1}$ or $D_i$ and $D_{i-1}$. The black line indicates the linear equation for $D_{13}$ as a function of $N_a$ for all flights, where: $D_r = (5\pm1.06)N_a$.

Figure 16 CDP-measured cloud droplet effective radius ($r_e$) (colored dots) and estimated cloud droplet adiabatic effective radius ($r_{ea}$) (colored lines) as a function of cloud depth ($D_c$) for all flights (indicated by colors). The height of 0 °C is indicated by a horizontal bar across the $r_{ea}$ line. The circles indicate the approximate values of drizzle water content (DWC) calculated from the CCP-CIP data, the range of DWC values is indicated in the table at the upper-right side of the figure. The star symbols indicate approximate mixed phase drizzle water content (MPWC) values calculated from the CCP-CIP data (indicated in the table at the bottom-right side of the figure). The temperature in °C of rain or ice initiation ($D_r$ and $D_i$, respectively) is indicated by colored numbers close to the circle or star symbols.

Figure 17. General characteristics of growing convective cumulus formed over the Amazon basin during the dry season. The heights of cloud base are higher over continental Amazon due to the smaller relative humidity in comparison with the maritime region. Convective clouds formed over the Atlantic Ocean, near the Brazilian coast, have smaller cloud droplets concentration ($N_d$) at cloud base due to the smaller concentration of aerosol and updraft speeds below cloud base. The initiation of warm rain ($D_r$) is observed at lower cloud depths (~2 km or ~10°C) from collision and coalescence processes. When convective clouds are more continental, larger aerosol concentration and updrafts are observed below cloud base, leading to larger $N_d$ nucleated at cloud base (as observed above forested and deforested

regions). Over the forest $D_r$ is observed near 0°C, whilst for the deforestation arc region the collision and coalescence processes are totally suppressed and the formation of ice particles took place at higher altitudes in the clouds in very polluted conditions, because the resulting smaller cloud droplets froze at colder temperatures compared to the larger drops in the less polluted cases.

**Table captions**

Table 1. List of abbreviations and symbols.

Table 2. Description of cloud probes, size range intervals and hydrometeor shapes observed on CCP-CIP images used to calculate CWC, DWC, RWC and MPWC.

Table 3. Classification of each flight as a function of $N_a$ at cloud base. The values of cloud base height ($Cbh$) and temperature ($T$), $D_{13}$, $D_r$ and $D_i$ in m and temperatures in °C are also shown for convective cloud measurements of each flight. Additionally, information about the height of $D_{r-1}$, $D_{i-1}$, $T_r$, $T_i$, $T_{i-1}$, NLP and $W_{max}$ is also shown for each flight. The uncertainties of $N_a$ and $D_{13}$ estimates are described at Appendix A.

**Tables**

Table 1. List of abbreviations and symbols.

| Abbreviation/notation | Description | Units |
|---|---|---|
| ACRIDICON-CHUVA | Aerosol, Cloud, Precipitation, and Radiation Interactions and Dynamics of Convective Cloud Systems - CHUVA (Cloud processes of tHe main precipitation systems in Brazil: A contribUtion to cloud resolVing modeling and to the GPM [Global Precipitation Measurements]) | - |
| CAS-DPOL | Cloud and Aerosol Spectrometer | - |
| $Cbh$ | Cloud base height | m |
| CCP-CDP | Cloud Combination Probe - Cloud Droplet Probe | - |
| CCP-CIP | Cloud Combination Probe - Cloud Imaging Probe | - |
| CCN | Cloud Condensation Nuclei | $cm^{-3}$ |
| CWC | Cloud water content | $g\ m^{-3}$ |
| $CWC_a$ | Adiabatic cloud water content | $g\ m^{-3}$ |
| $D_c$ | Cloud depth - distance from cloud base | m |
| $D_r$ | Cloud depth where first drizzle with drop shape was detected | m |
| $D_{r-1}$ | Nearest cloud depth below $D_r$ without raindrop | m |
| $D_i$ | Cloud depth where first drizzle with ice shape was detected | m |
| $D_{i-1}$ | Nearest cloud depth below $D_i$ without ice particles | m |
| DWC | Drizzle Water Content | $g\ m^{-3}$ |
| DSD | Cloud-droplet size distribution | $cm^{-3}\ \mu m^{-1}$ |
| $D_{13}$ | Cloud depth where $r_{ea} = 13\ \mu m$ | m |
| IN | Ice Nuclei | $cm^{-3}$ |
| $K$ | The collection kernel of a pair of droplets | $cm^{-3}\ s^{-1}$ |
| LWC | Liquid Water Content | $g\ m^{-3}$ |
| MPWC | Mixed Phase Water Content | $g\ m^{-3}$ |
| $M_v$ | Mean volume cloud droplet | $\mu m^{-3}$ |
| $M_{va}$ | Adiabatic mean volume cloud droplet | $\mu m^{-3}$ |
| $N_a$ | Adiabatic number concentration of droplets | $cm^{-3}$ |
| $N_d$ | Number concentration of droplets | $cm^{-3}$ |
| $N_d^{*}$ | Effective number of droplets concentration at cloud base | $cm^{-3}$ |
| $NLS$ | Number of altitude levels sampled | - |
| PCASP | Passive Cavity Aerosol Spectrometer Probe | - |
| PSD | Aerosol particle size distribution | $cm^{-3}\ \mu m^{-1}$ |
| $r_e$ | The effective radius of the cloud droplet spectra | $\mu m$ |
| $r_{ea}$ | The adiabatic effective radius of the cloud droplet spectra | $\mu m$ |
| $r_v$ | The mean volume radius of the cloud droplets | $\mu m$ |
| RWC | Rainwater Content | $g\ m^{-3}$ |
| $S$ | Supersaturation | % |
| $T$ | Temperature | °C |
| $T_r$ | Temperature of rain initiation | °C |
| $T_i$ | Temperature of ice initiation | °C |
| $T_{i-1}$ | Nearest temperature greater than $T_i$ without ice particles | °C |
| $W$ | Vertical velocity | $m\ s^{-1}$ |
| $W_{max}$ | Maximum vertical velocity during the cloud profiling flight | $m\ s^{-1}$ |

Table 2. Description of cloud probes, size range intervals and hydrometeor shapes observed on CCP-CIP images used to calculate CWC, DWC, RWC and MPWC.

| Abbreviation/Notation | Instrument | Size range | Hydrometeor shapes |
|---|---|---|---|
| **CWC** | CCP-CDP/CAS-DPOL | 3-50 μm | Cloud droplets |
| **DWC** | CCP-CIP | 75-250 μm | Cloud droplets and raindrops |
| **RWC** | CCP-CIP | 250-960 μm | Cloud droplets and raindrops |
| **MPWC** | CCP-CIP | 75-960 μm | Cloud droplets and ice particles |

Table 3. Classification of each flight as a function of $N_a$ at cloud base. The values of cloud base height ($Cbh$) and temperature ($T$), $D_{13}$, $D_r$ and $D_i$ in m and temperatures in °C are also shown for convective cloud measurements of each flight. Additionally, information about the height of $D_{r-1}$, $D_{i-1}$, $T_r$, $T_i$, $T_{i-1}$, NLP and $W_{max}$ is also shown for each flight. The uncertainties of $N_a$ and $D_{13}$ estimates are described at Appendix A.

| Flight | $Cbh$ (m)/ $T$ (°C) | $N_a$ (cm⁻³) | $D_{13}$(m) | $D_{r-1}$ (m) | $D_r$ (m) | $T_r$(°C) | $D_{i-1}$ (m) | $T_{i-1}$(°C) | $D_i$ (m) | $T_i$(°C) | NLS | $W_{max}$ (m s⁻¹) | Classification |
|---|---|---|---|---|---|---|---|---|---|---|---|---|---|
| AC07 | 1900 / 15 | 963 ± 236 | 4500 ± 1104 | - | - | - | 3631 | -5.3 | 4537 | -9.1 | 14 | 13.09 | very polluted |
| AC08 | 1100 / 20 | 920 ± 162 | 3900 ± 690 | - | - | - | - | - | - | - | 10 | 8.2 | very polluted |
| AC09 | 1200 / 19.5 | 566 ± 98 | 2400 ± 420 | 2300 | 3000 | 2.4 | 4570 | -6.0 | 5217 | -9.2 | 15 | 8.8 | polluted |
| AC12 | 2200 / 15.5 | 1546 ± 434 | 9000 ± 2540 | - | - | - | - | - | - | - | 12 | 18.9 | very polluted |
| AC13 | 2200 / 15.5 | 1080 ± 234 | 5500 ± 1194 | - | - | - | 4240 | -9.0 | 4800 | -14.1 | 12 | 9.2 | very polluted |
| AC18 | 1700 / 17 | 666 ± 114 | 2900 ± 512 | 3100 | 3800 | -5.7 | - | - | - | - | 13 | 19.9 | polluted |
| AC19 | 600 / 22 | 276 ± 54 | 1000 ± 198 | 1150 | 1660 | 10 | - | - | - | - | 13 | 8.49 | clean |
| AC20 | 1900 / 16.5 | 987 ± 224 | 5000 ± 1130 | - | - | - | - | - | - | - | 8 | 16.6 | very polluted |




**Figures**

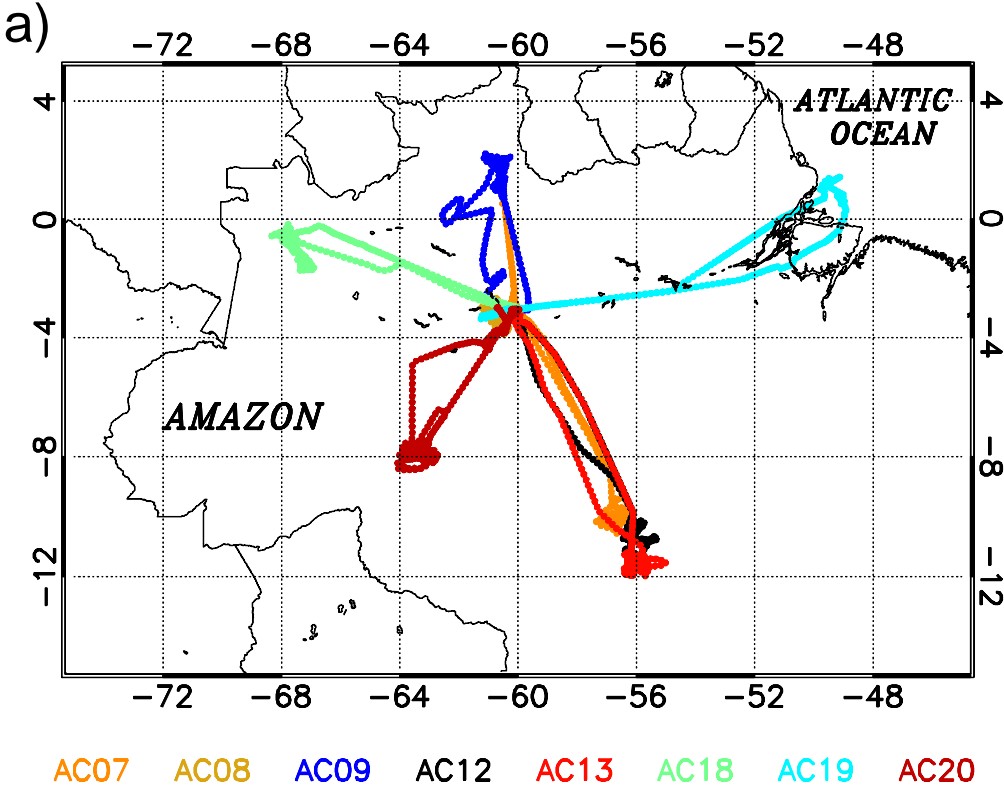

AC07    AC08    AC09    AC12    AC13    AC18    AC19    AC20

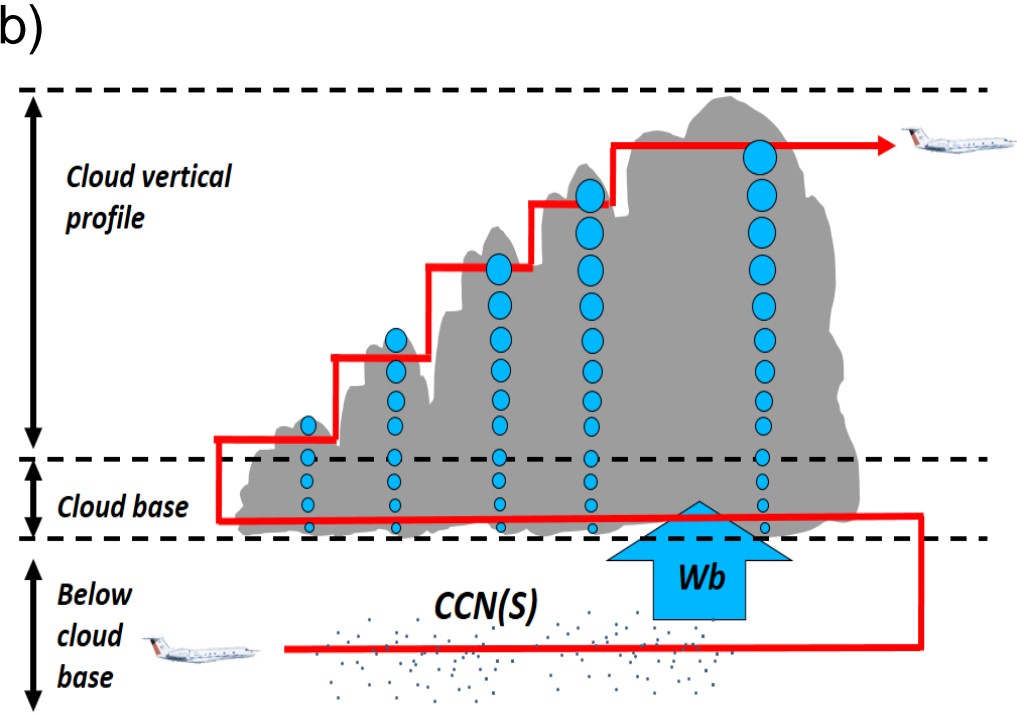


**Figure 1 a) HALO flight tracks during the ACRIDICON-CHUVA experiment. The flight number is indicated at the bottom by colors; b) Flight patterns below and in convective clouds during the ACRIDICON-CHUVA campaign.**

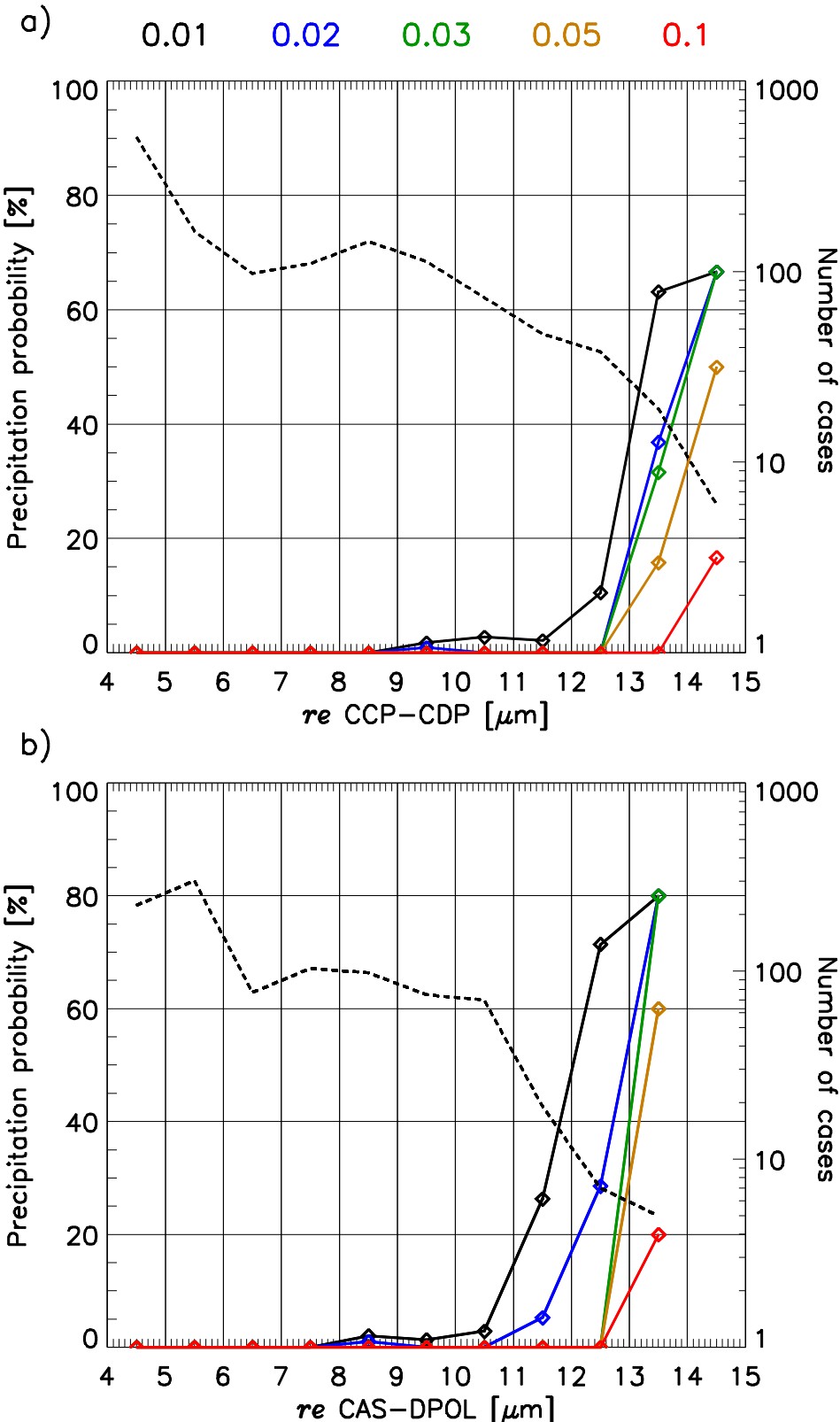

**Figure 2 a) Precipitation probability as a function of $r_e$ for the CCP-CDP probe for different DWC thresholds (black – DWC > 0.01 g m$^{-3}$; blue – DWC > 0.02 g m$^{-3}$; green – DWC > 0.03 g m$^{-3}$; gold – DWC > 0.05 g m$^{-3}$; red – DWC > 0.1 g m$^{-3}$). The dashed line indicates the number of cases (in seconds for each 1-s cloud pass) for each $r_e$ size interval (right axis); b) Similar for the CAS-DPOL probe.**

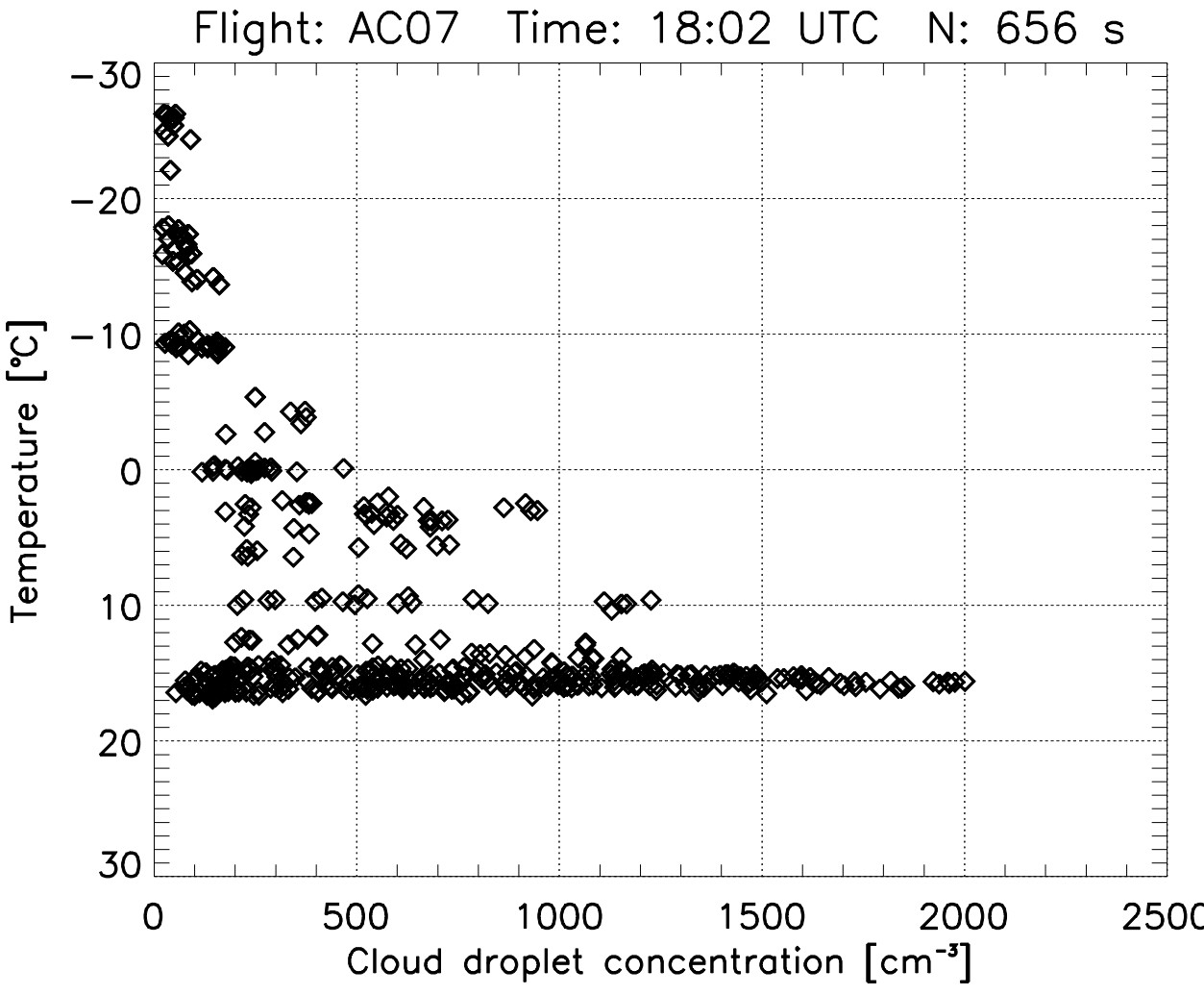


**Figure 3 Cloud droplet concentration measured with CCP-CDP as a function of temperature for flight AC07. Each dot indicates a 1-Hz average concentration. The sample number (N) and the approximate start time of the cloud profile are shown at the top of the panel.**

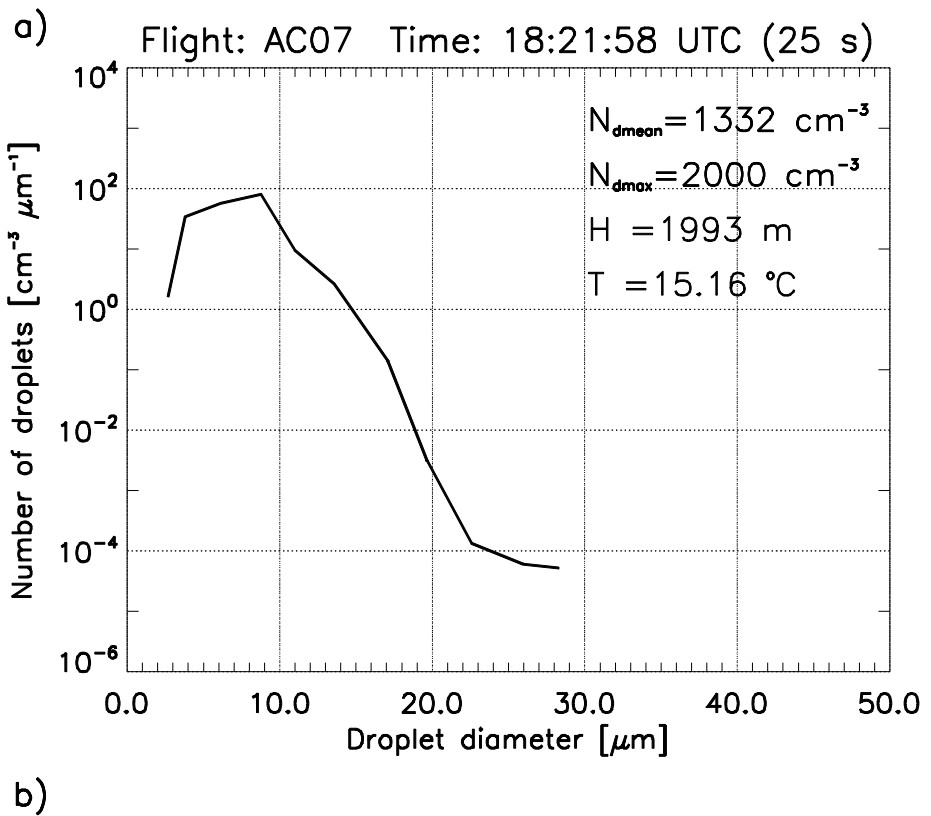

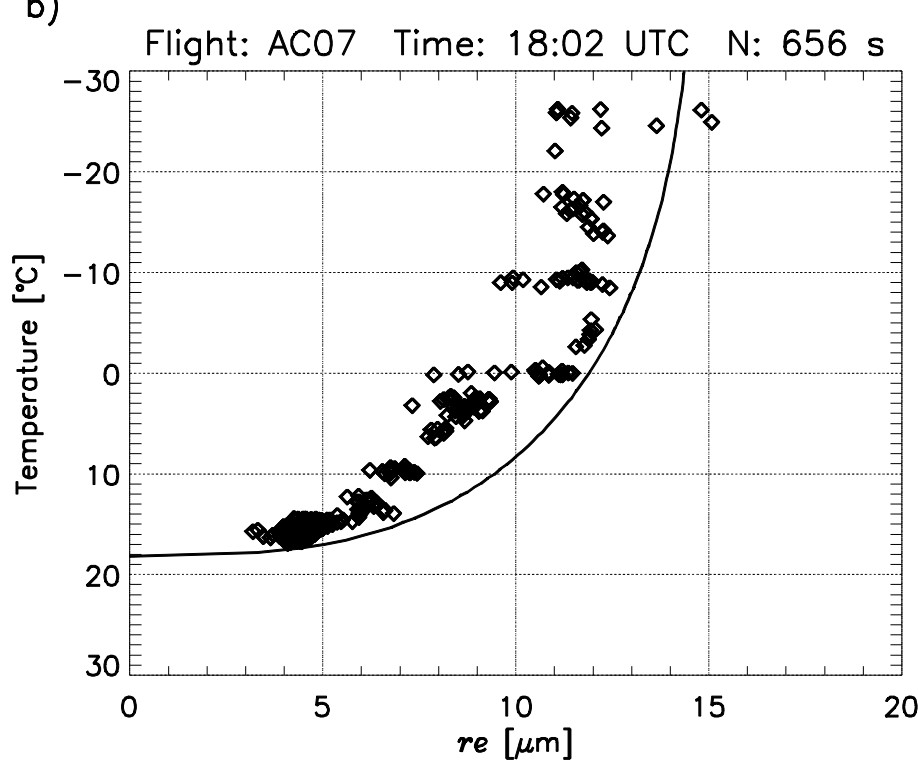

Figure 4 a) Mean cloud droplet size distribution calculated from the CCP-CDP data for a cloud pass at cloud base during flight AC07. The flight number, initial time of cloud pass, and duration in seconds are shown at the top of graph. The mean total number of droplets ($N_{dmean}$), the maximum total number of droplets ($N_{dmax}$) in one second for this cloud pass, and the approximate height (H) and temperature (T) are shown at the upper-right corner of the graph; b) Cloud droplet effective radius ($r_e$) calculated from CCP-CDP as a function of temperature indicated with dots. The black line indicates the estimated adiabatic effective radius ($r_{ea}$) as a function of temperature.




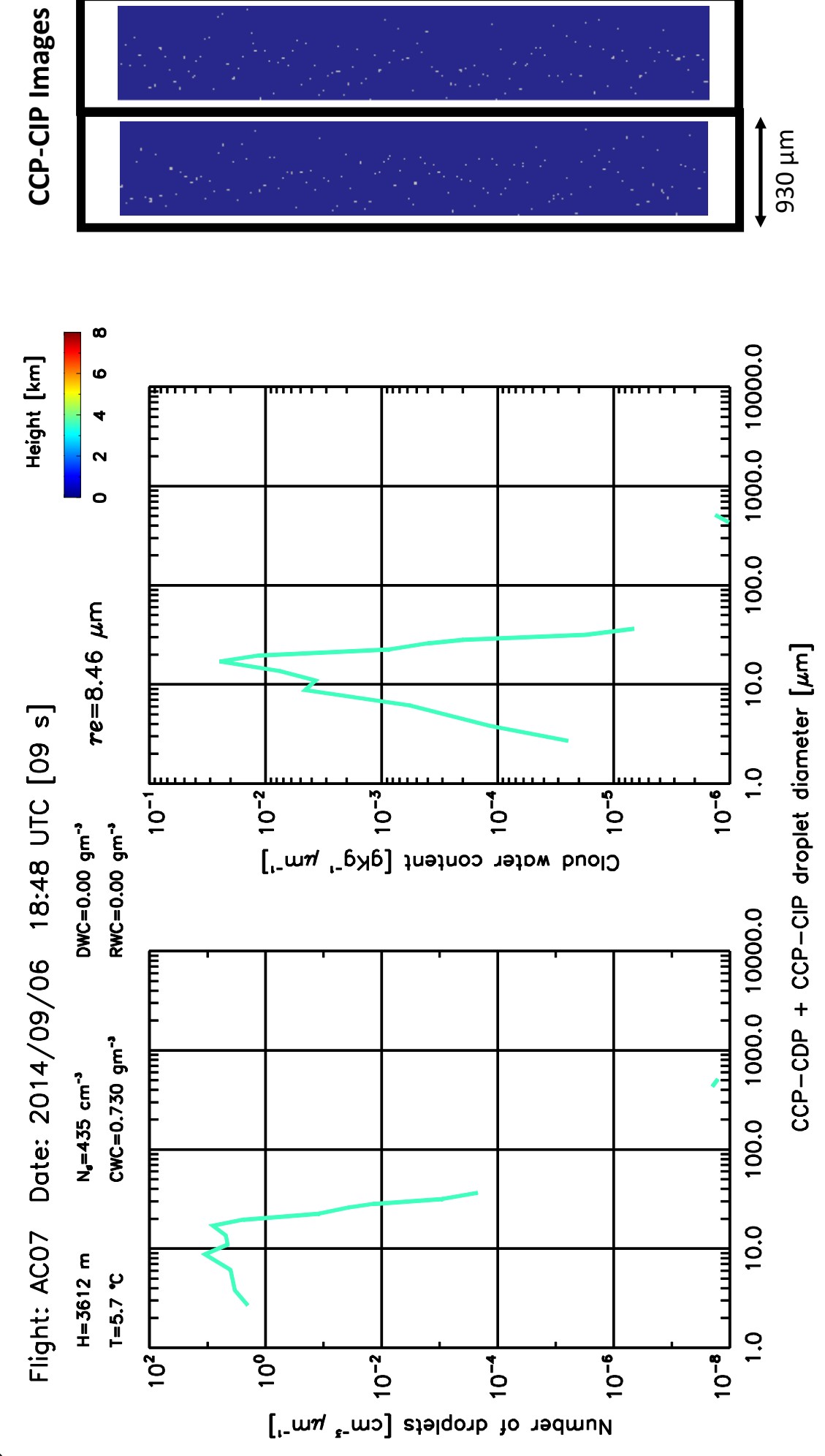

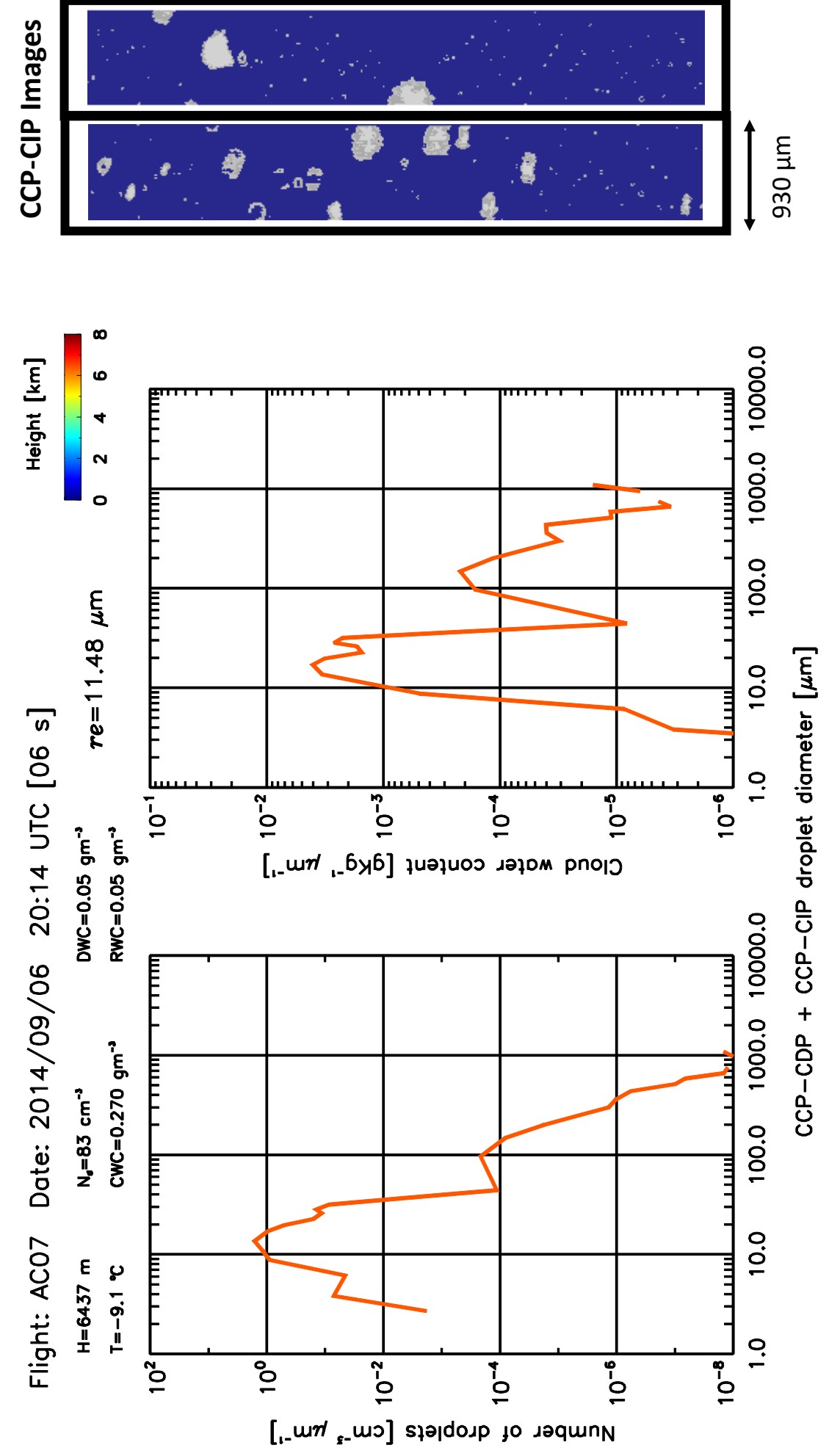

CCP-CIP Images

930 µm

b)

Flight: ACO7   Date: 2014/09/06   20:14 UTC [06 s]

H=6437 m      $N_s$=83 cm$^{-3}$     DWC=0.05 gm$^{-3}$     $re$=11.48 µm

T=-9.1 °C     CWC=0.270 gm$^{-3}$    RWC=0.05 gm$^{-3}$

Height [km]

Number of droplets [cm$^{-3}$ µm$^{-1}$]

Cloud water content [gkg$^{-1}$ µm$^{-1}$]

CCP-CDP + CCP-CIP droplet diameter [µm]

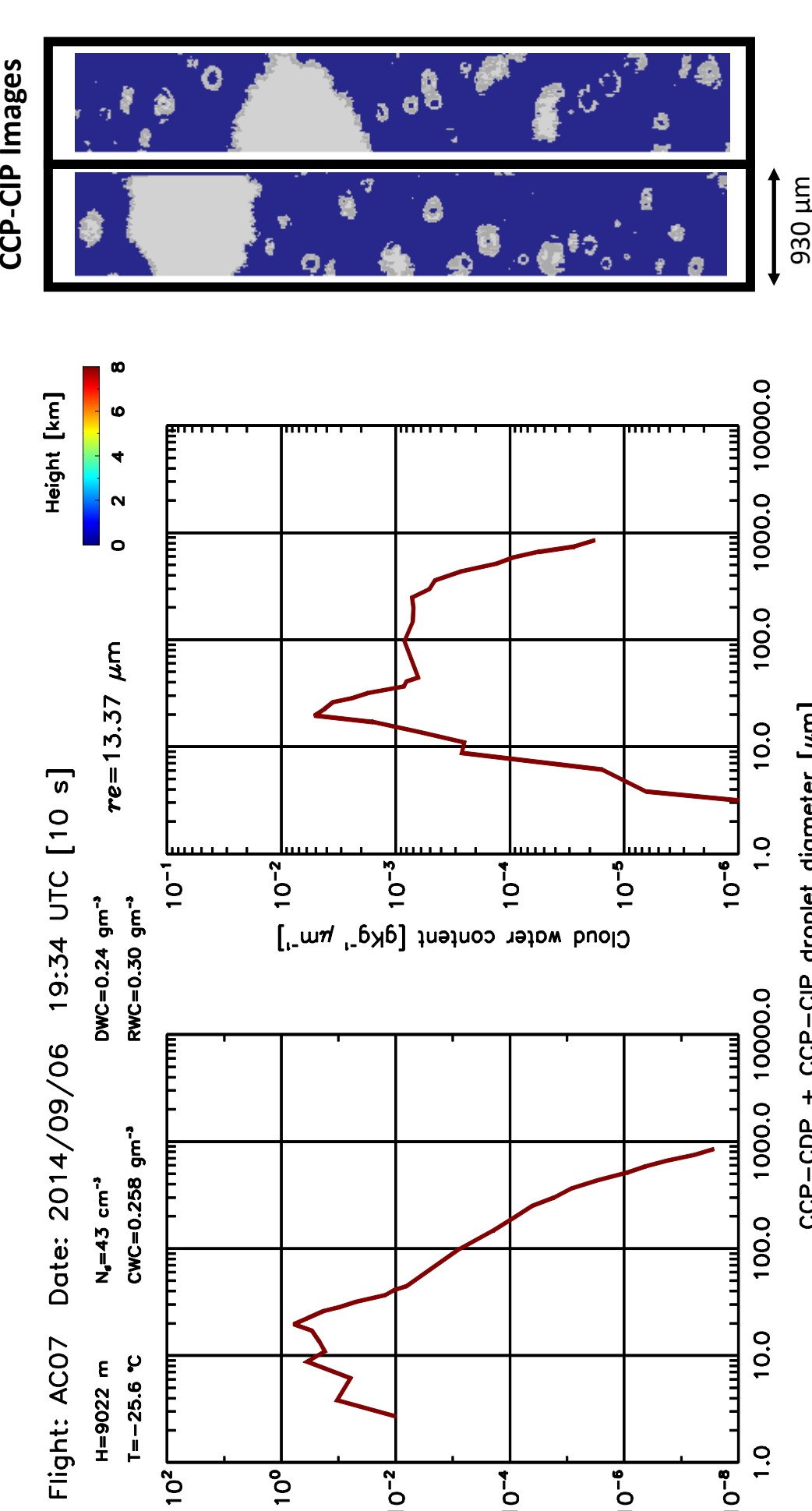

Figure 5a-c. Mean droplet size distribution composite from the CCP-CDP and CCP-CIP probes (left panel). Similar for indicated cloud water content in the right panel. Indicated at the top of the panels are the HALO flight number, date, time of flight (UTC), duration of cloud pass in seconds, temperature (T) and altitude (H) above sea level, and the mean values for the total number of droplets ($N_d$), CWC, DWC, RWC, and $r_e$. The color bars indicate the height of HALO during the cloud pass. On the right side of the panels CCP-CIP images corresponding to the cloud pass are shown.

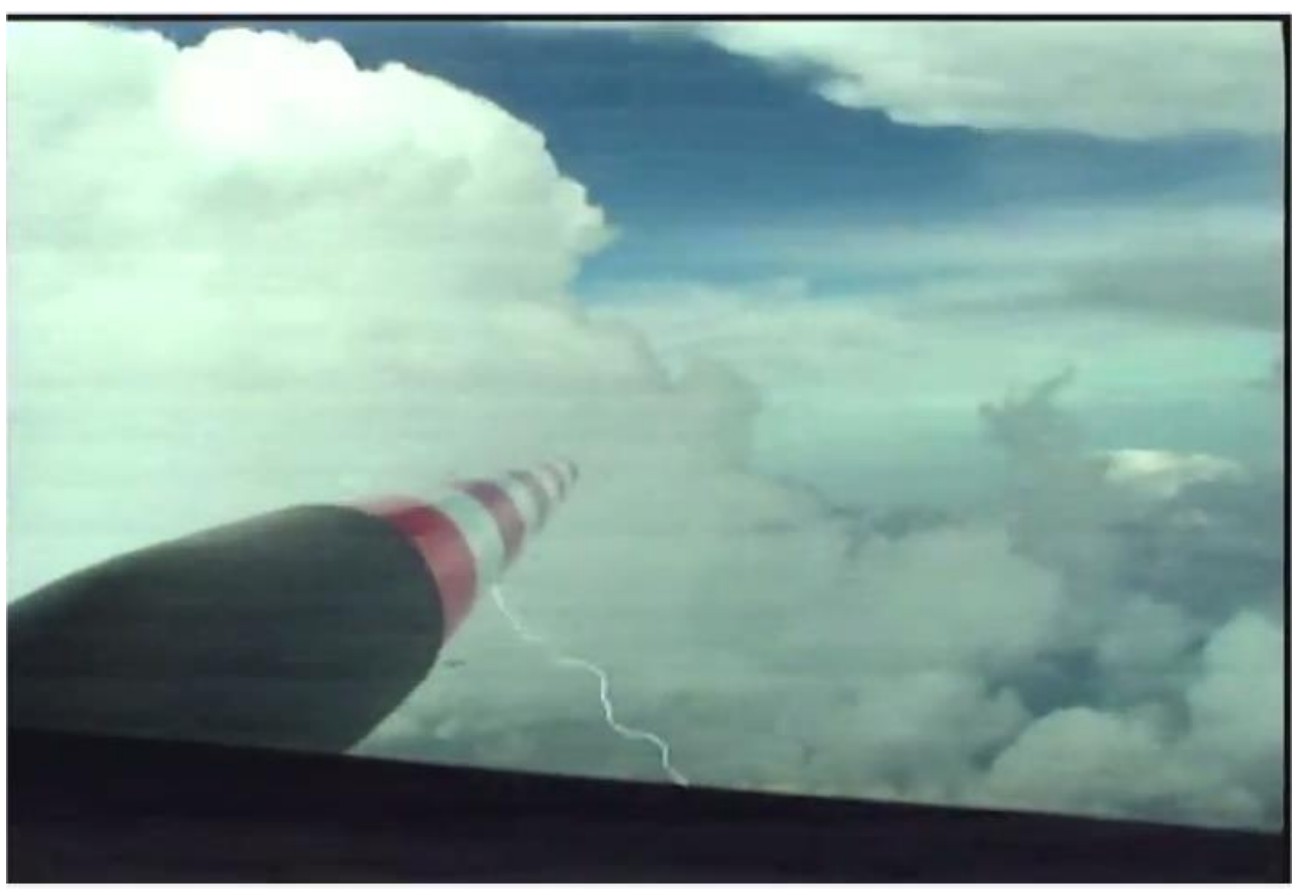

**Figure 6 Image taken from the HALO cockpit just before the aircraft penetration of a convective cloud with lightning activity during flight AC07. In this case, the cloud pass height was 9,022 m (temperature ~ -25 ºC) and the maximum CWC measured was 0.55 g m⁻³.**

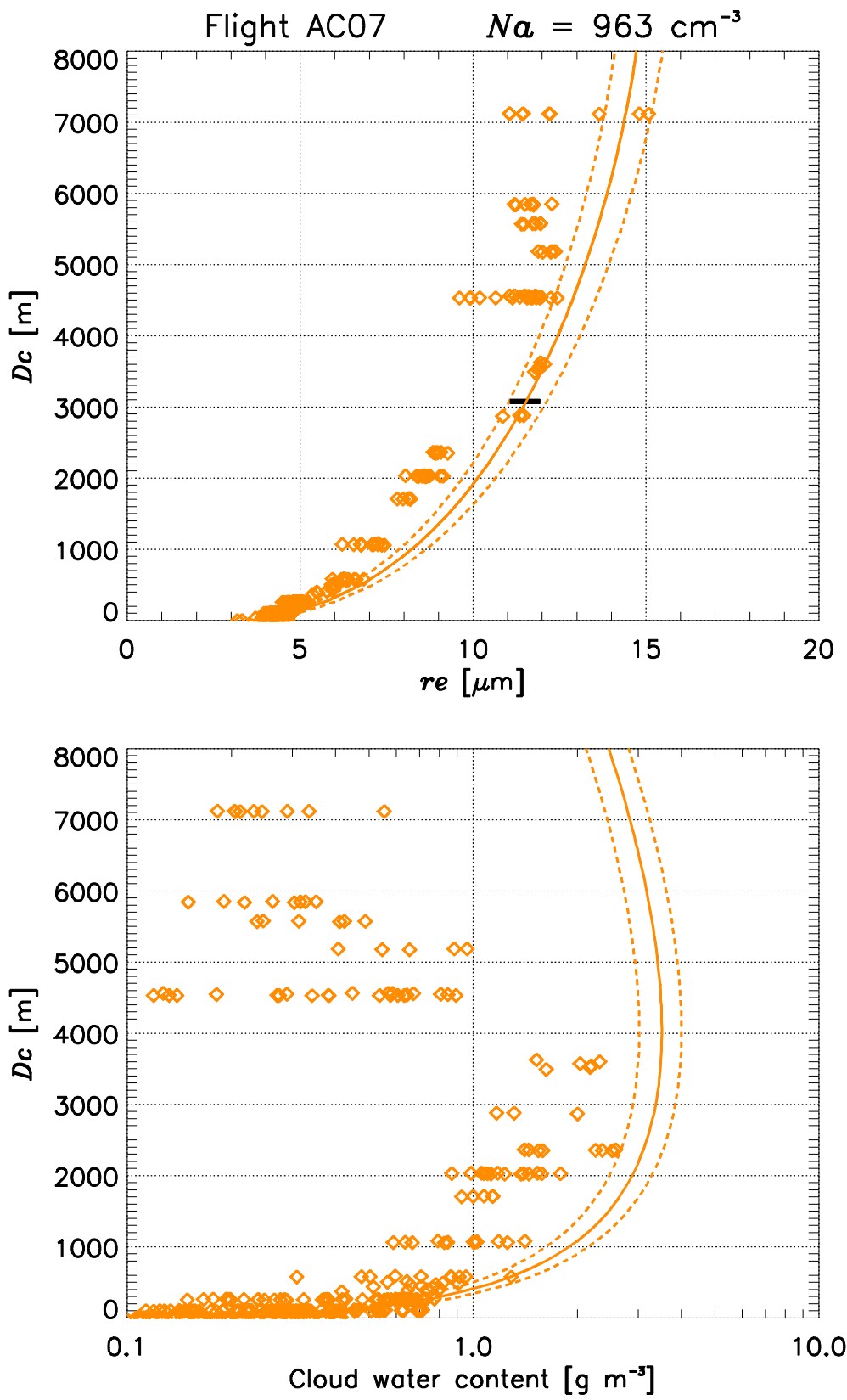

**Figure 7 a) Cloud droplet effective radius ($r_e$) as a function of cloud depth ($D_c$) for flight AC07. The line indicates the $r_e$ estimated for adiabatic growth ($r_{ea}$) from cloud base (dashed lines indicate the $r_{ea}$ values considering the uncertainty of the estimate). The height of 0 ºC is indicated by a black horizontal bar across the $r_{ea}$ line. The estimated adiabatic number of droplets ($N_a$) at cloud base is shown at the top of the figure. b) Similar to a) for Cloud water content (adiabatic values are shown by lines).**

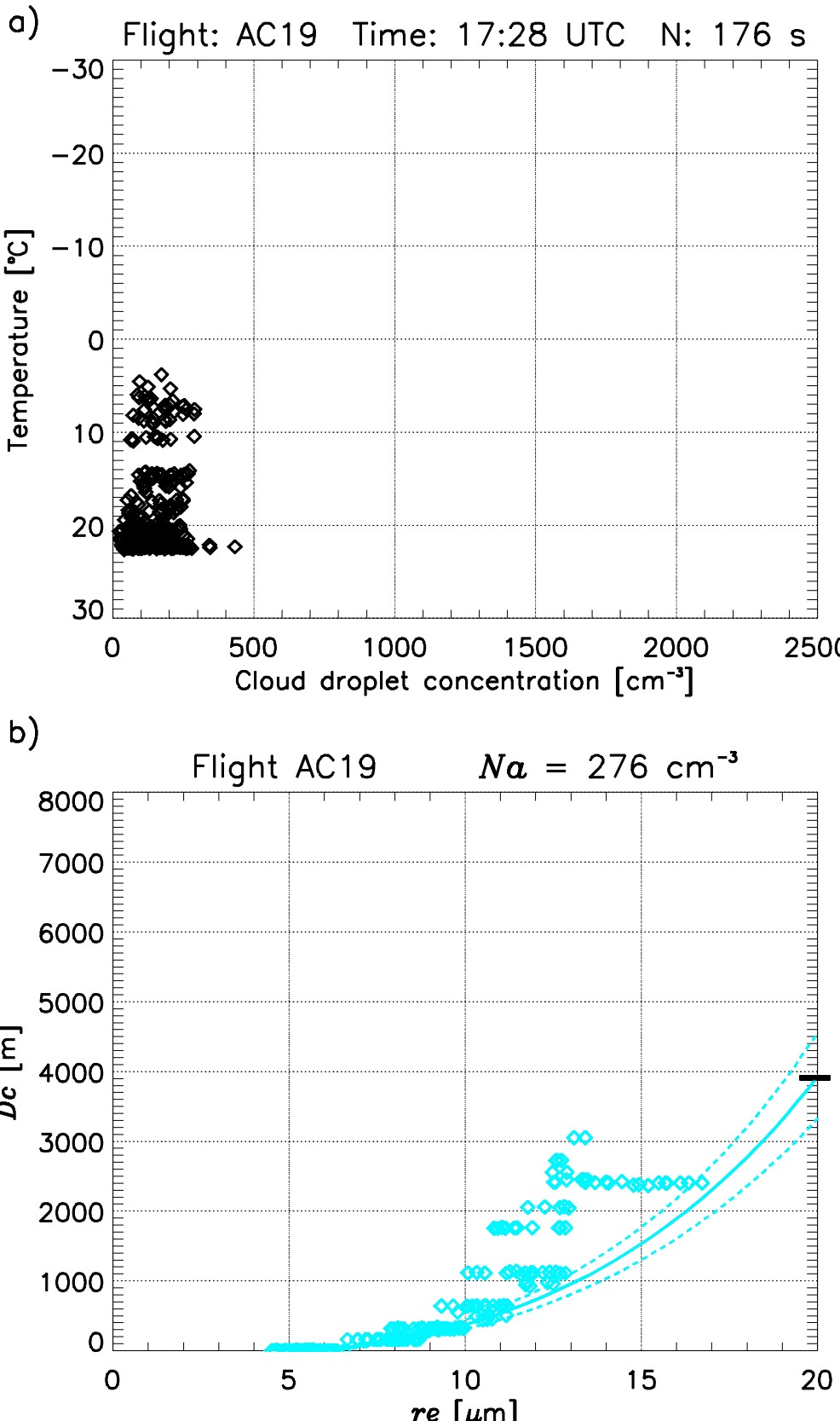

**Figure 8 a) Cloud droplet concentrations measured with the CCP-CDP as a function of temperature for flight AC19. Each dot indicates 1Hz average concentration. The sample number in seconds (N) and the start time of the cloud profile are shown at the top of the panel; b) Similar to Figure 7 for flight AC19.**

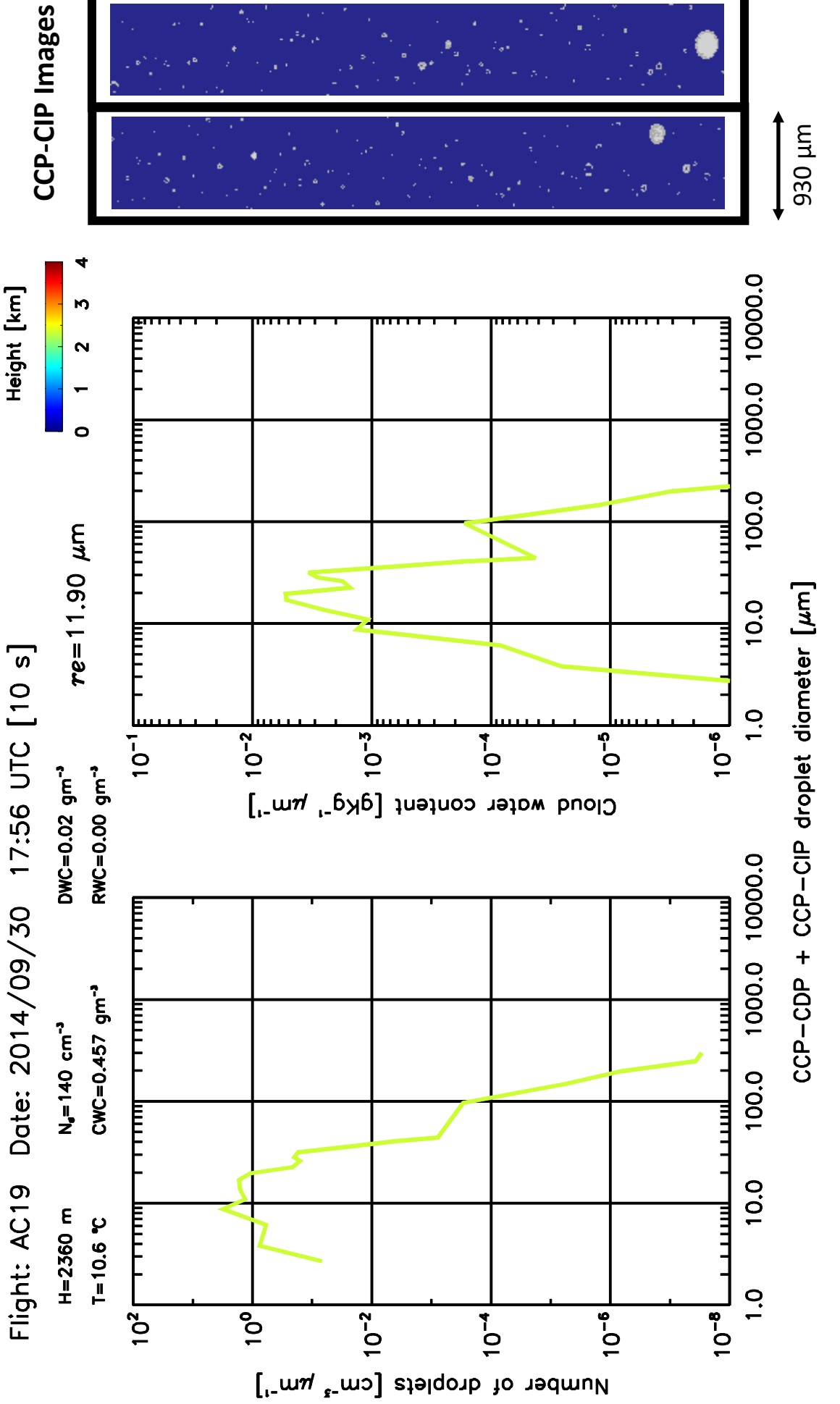

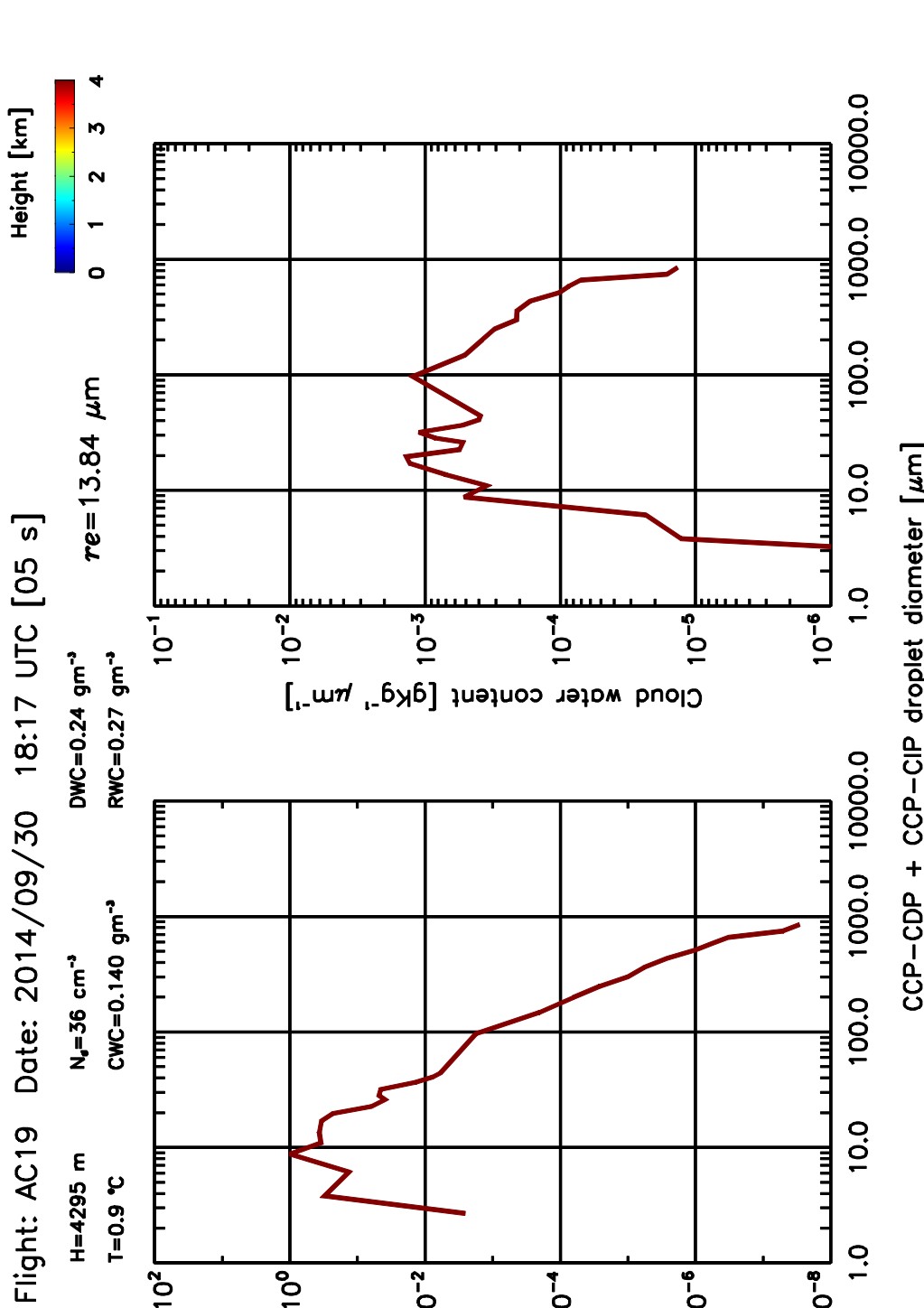

Figures 9 a-b) Similar to Figures 5a-c for flight AC19.

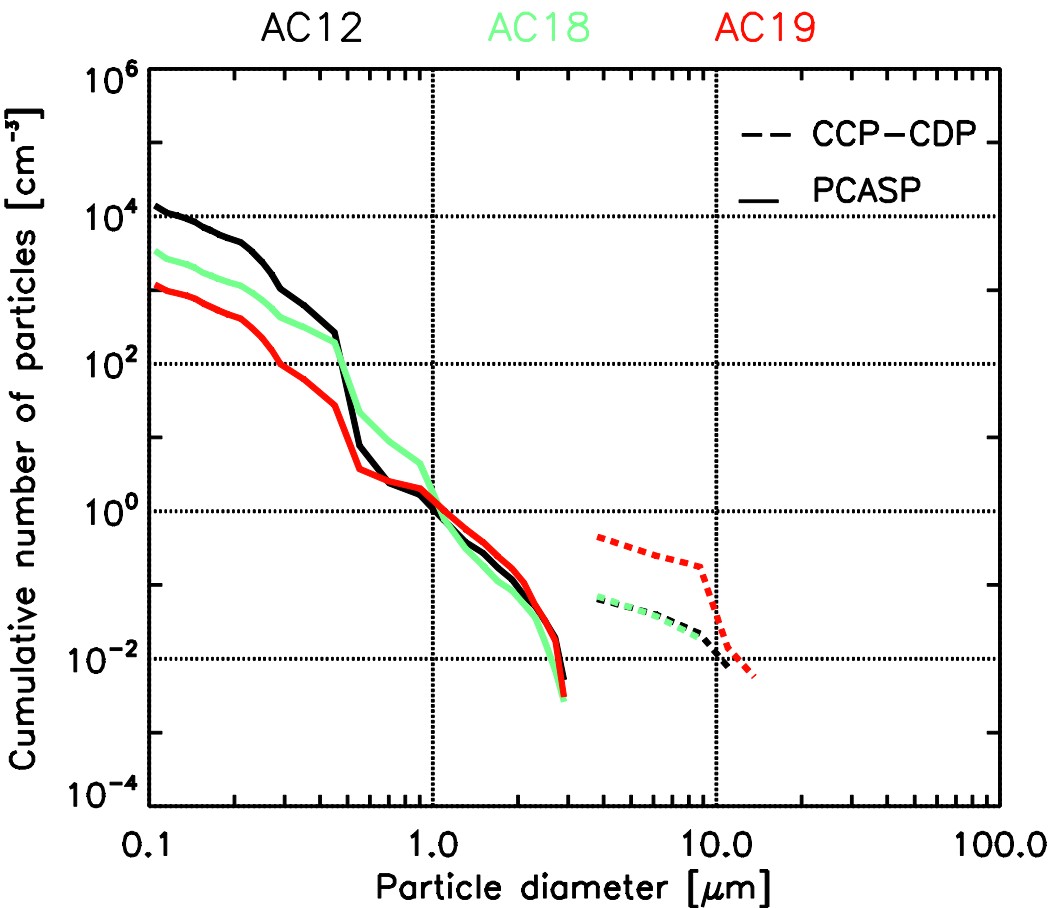

**Figure 10** Cumulative aerosol size distribution below cloud base calculated from the PCASP probe for typical clean, polluted, and very polluted regions (solid line) for flights AC12 (very polluted), AC18 (polluted), and AC19 (clean). Similar for cumulative cloud droplet size distribution calculated with CCP-CDP (dashed line). The flight numbers are indicated by colors at the top of the panel.

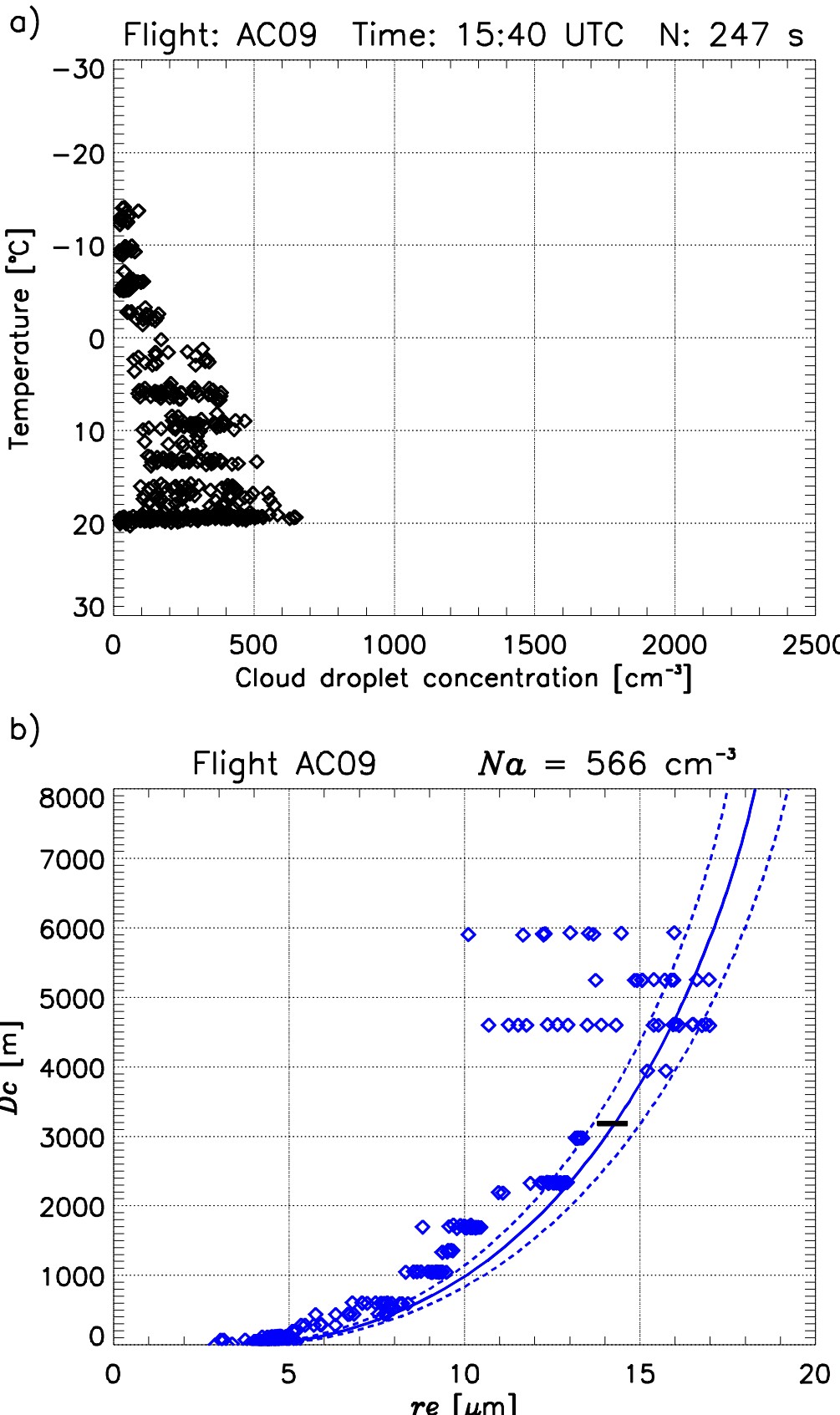

**Figure 11 a) Cloud droplet concentrations measured with the CCP-CDP as a function of temperature for flight AC09. Each dot indicates 1-Hz average concentration. The sample number in seconds (N) and the start time of the cloud profile are shown at the top of the panel; b) Similar to Figure 7 for flight AC09.**

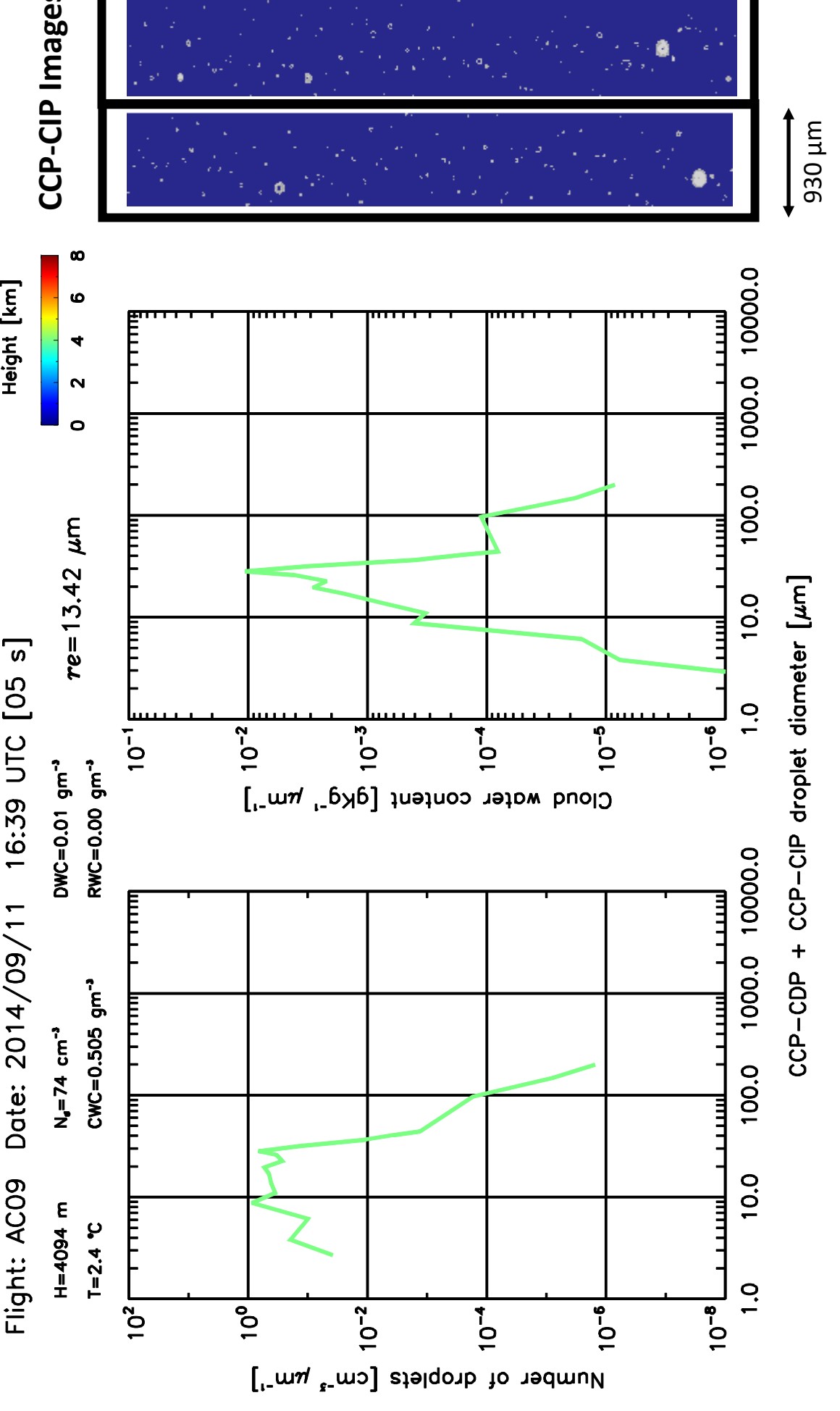

**CCP-CIP Images**

930 μm

**b)**

Flight: AC09   Date: 2014/09/11   17:13 UTC [04 s]

H=6417 m      $N_a$=31 cm⁻³   CWC=0.247 gm⁻³

T=−9.2 °C

DWC=0.16 gm⁻³

RWC=0.27 gm⁻³

$re$=15.97 μm

Height [km]

0   2   4   6   8

Cloud water content [gkg⁻¹/μm⁻¹]

Number of droplets [cm⁻³/μm⁻¹]

CCP−CDP + CCP−CIP droplet diameter [μm]

Figures 12 a-b) Similar to Figures 5a-c for flight AC09.

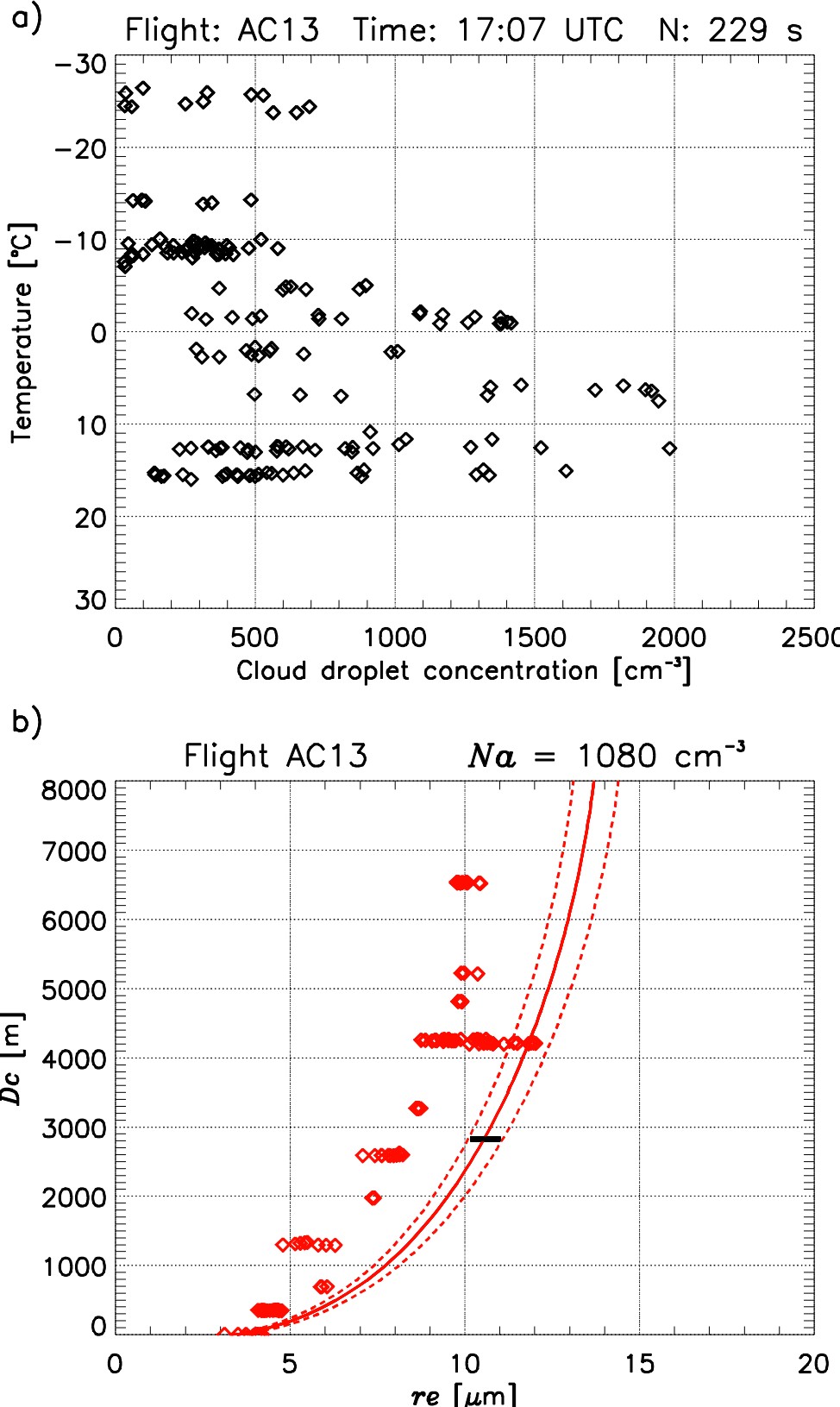

**Figure 13 a) Cloud droplet concentration measured with the CCP-CDP probe as a function of temperature for flight AC13. Each dot indicates a 1-Hz average concentration. The sample number and the approximate time of the cloud profile are shown at the top of the panel; b) Similar to Figure 7 for Flight AC13.**

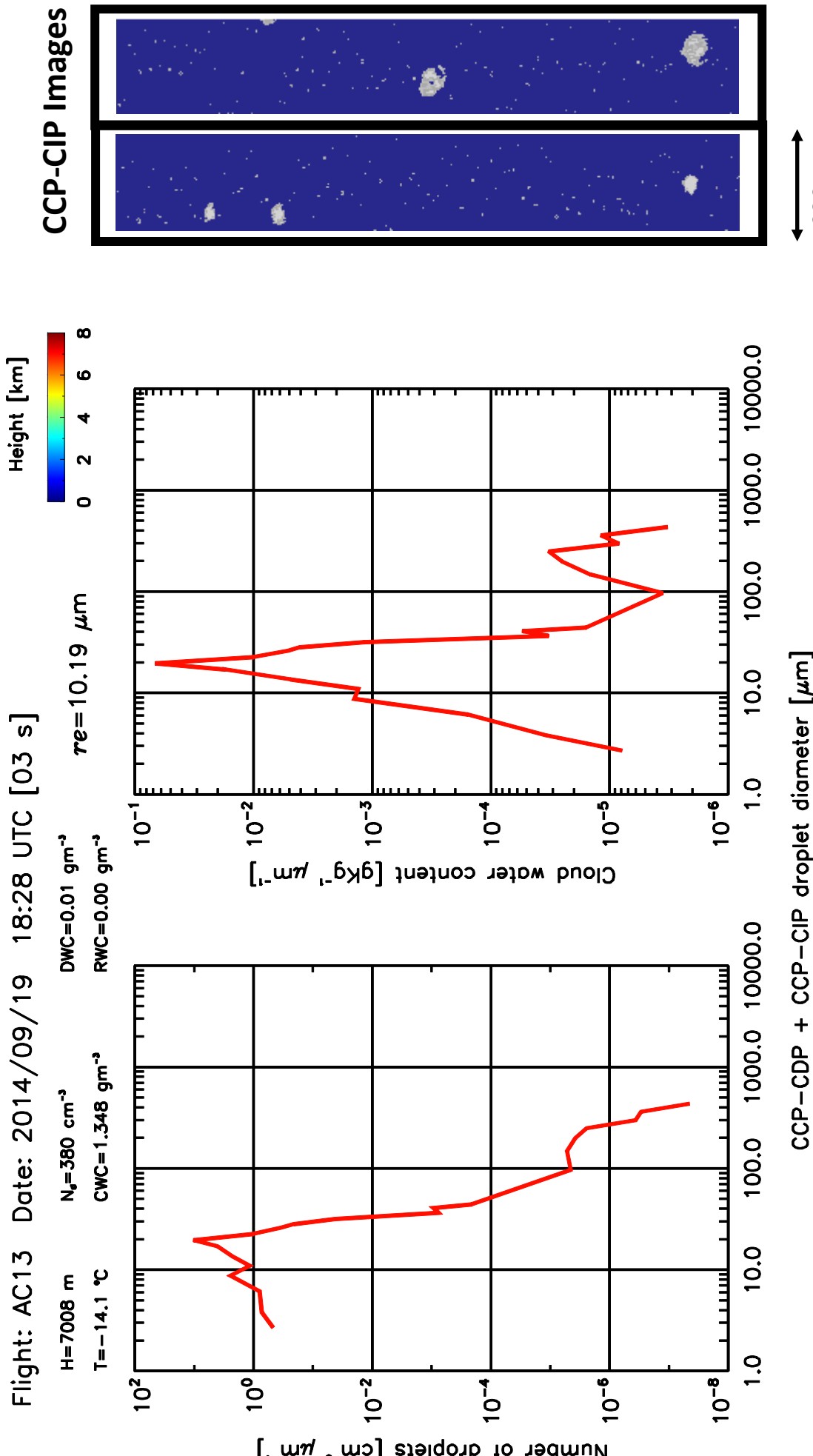

Figures 14 Similar to Figures 5a-c for flight AC13.

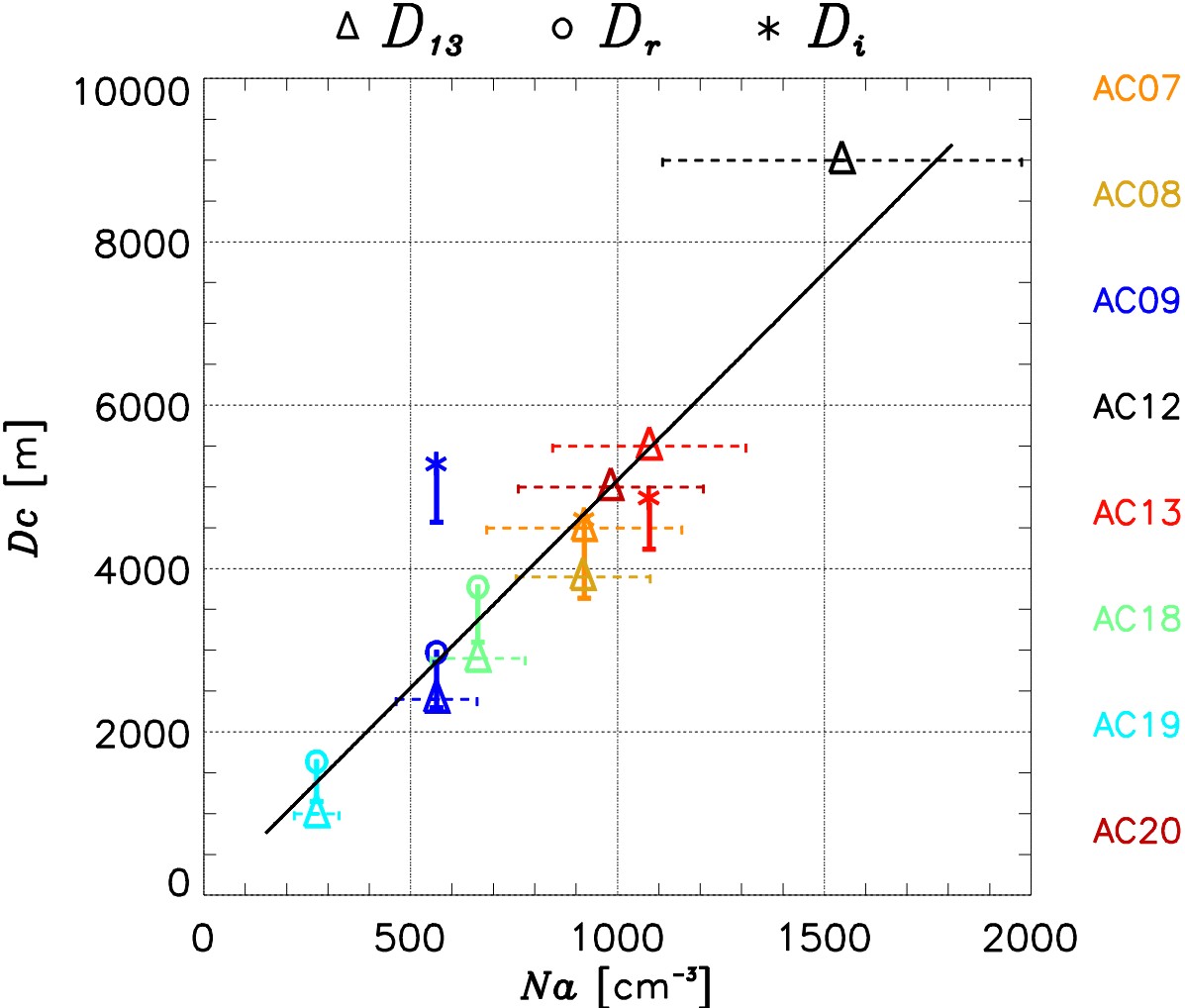

**Figure 15** Cloud depth ($D_c$) as a function of the estimated adiabatic number of droplets ($N_a$) at cloud base. $D_c$ for adiabatic cloud droplet effective radius ($r_{ea}$) equal 13 μm (or $D_{13}$) are indicated by triangles. Similar for cloud depth of rain initiation ($D_r$) [indicated by circles] and cloud depth for ice initiation ($D_i$) [indicated by asterisk]. The flight numbers are indicated by colors on the right side of the panel. The values of $D_{13}$, $D_r$, and $D_i$ are shown in Table 1. The uncertainties of $N_a$ estimates are shown by horizontal error bars. The vertical error bars indicate the cloud depth between $D_r$ and $D_{r-1}$ or $D_i$ and $D_{i-1}$. The black line indicates the linear equation for $D_{13}$ as a function of $N_a$ for all flights, where: $D_r = (5\pm1.06)N_a$.

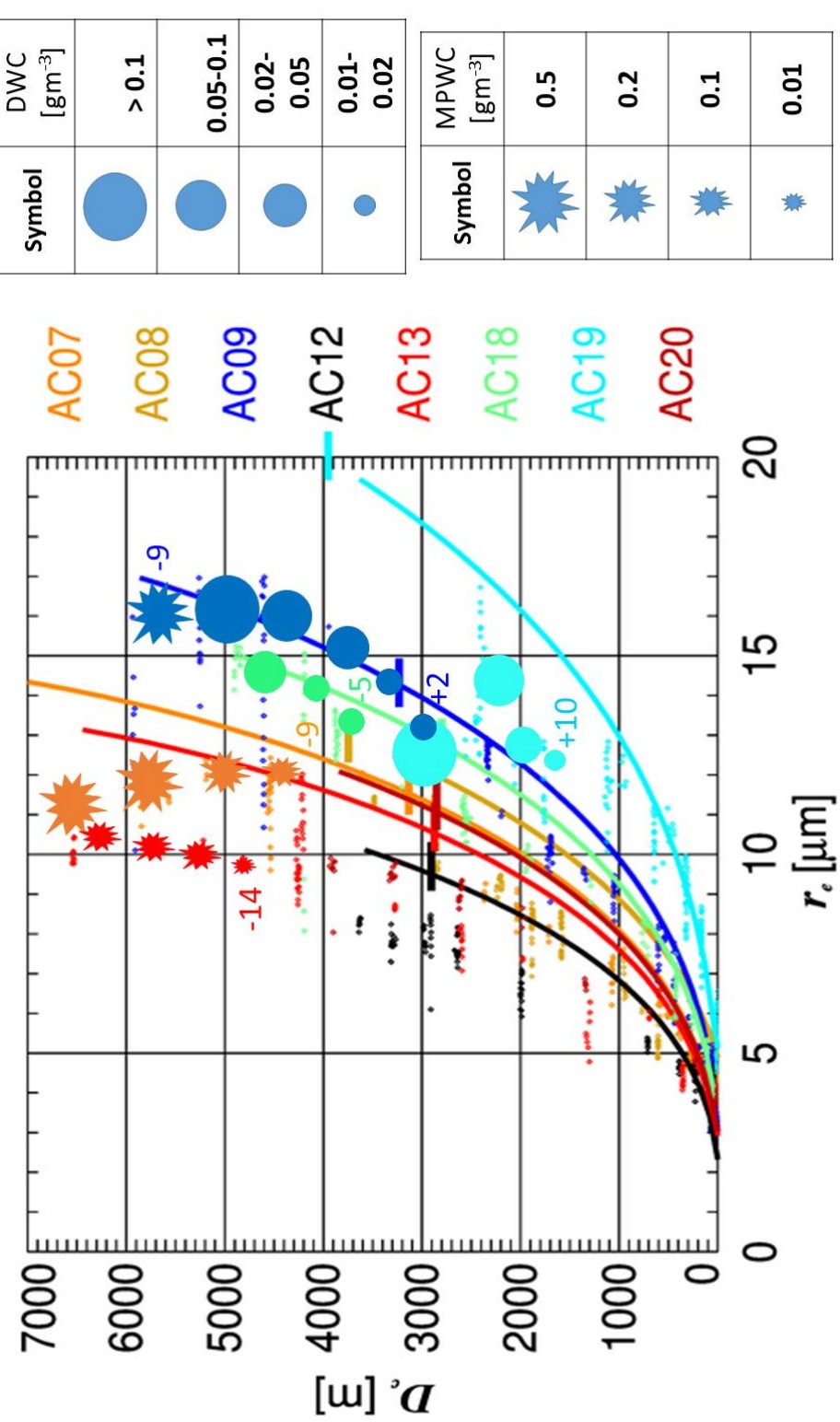

**Figure 16** CDP-measured cloud cloud droplet effective radius ($r_e$) (colored dots) and estimated cloud droplet adiabatic effective radius ($r_{ea}$) (colored lines) as a function of cloud depth ($D_c$) for all flights (indicated by colors). The height of 0 °C is indicated by a horizontal bar across the $r_{ea}$ line. The circles indicate the approximate values of drizzle water content (DWC) calculated from the CCP-CIP data, the range of DWC values is indicated in the table at the upper-right side of the figure. The star symbols indicate approximate mixed phase drizzle water content (MPWC) values calculated from the CCP-CIP data (indicated in the table at the bottom-right side of the figure). The temperature in °C of rain or ice initiation ($D_r$ and $D_i$, respectively) is indicated by colored numbers close to the circle or star symbols.

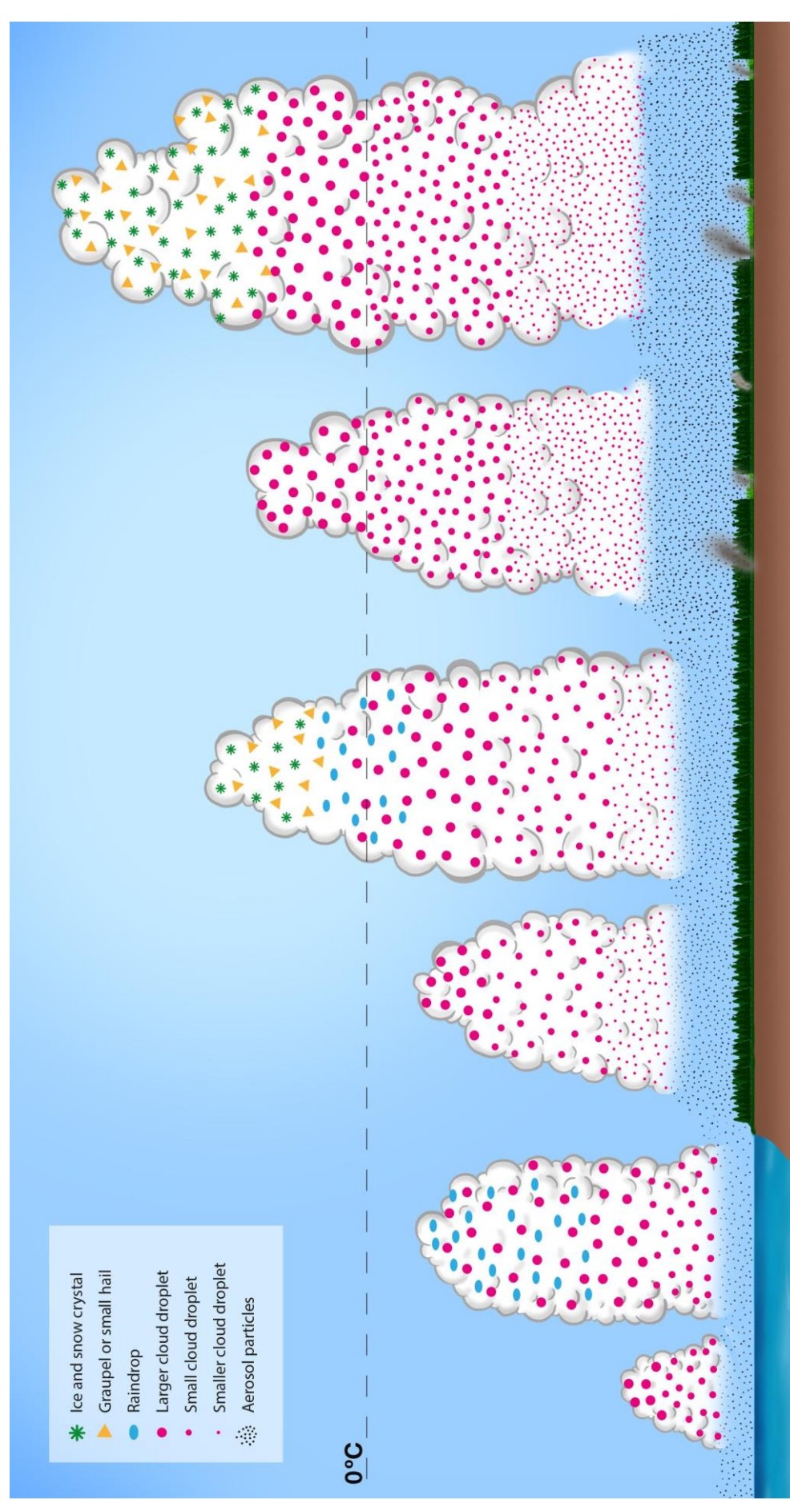

**Figure 17.** General characteristics of growing convective cumulus formed over the Amazon basin during the dry season. The heights of cloud base are higher over continental Amazon due to the smaller relative humidity in comparison with the maritime region. Convective clouds formed over the Atlantic Ocean, near the Brazilian coast, have smaller cloud droplets concentration ($N_d$) at cloud base due to the smaller concentration of aerosol and updraft speeds below cloud base. The initiation of warm rain ($D_r$) is observed at lower cloud depths (~2 km or ~10°C) from collision and coalescence processes. When convective clouds are more continental, larger aerosol concentration and updrafts are observed below cloud base, leading to larger $N_d$ nucleated at cloud base (as observed above forested and deforested regions). Over the forest $D_r$ is observed near 0°C, whilst for the deforestation arc region the collision and coalescence processes are totally suppressed and the formation of ice particles took place at higher altitudes in the clouds in very polluted conditions, because the resulting smaller cloud droplets froze at colder temperatures compared to the larger drops in the less polluted cases.

