# Peer review of "Further evidence for CCN aerosol concentrations determining the height of warm rain and ice initiation in convective clouds over the Amazon basin"

_Atmospheric Chemistry and Physics, 2016_

## Referee Comment (RC1) · Anonymous Referee #1 · 3 Apr 2017

**Reviewer B**

Title: Aerosol concentration determines the height of warm rain and ice initiation in convective clouds over the Amazon basin.

Submitted: Braga et al

To: ACP

Date: March 30 2017

Decision: Rejected

**Summary:**

This manuscript studies the effects of aerosol particle number concentration on the initiation of raindrops and ice hydrometeors in growing convective cumulus over the Amazon. Data from aerosol and cloud probes on board of the HALO aircraft are used. The values of the estimated $Na$ at cloud base were applied to classify the atmospheric conditions where convective clouds developed as a function of aerosol particle number concentration (i.e., clean, polluted, and very polluted regions). Main conclusion was that cloud depth, assuming adiabatic assumptions) is related to aerosol characteristics at the surface.

**Major issues:**

This paper claims that cloud depth and precip initiation are related to aerosol number concentrations at the cloud base.  I argue that precip formation in the clouds are related to saturation rate, updrafts and shear (turbulence), T gradients, stability, and particle type. I really do not like simplified convective clouds dependency on only aerosol concentration at the cloud base. Doing this you ignored non-adiabatic terms that include radiation, turbulence, and mixing can play important role for cloud structure. In fact difference between the adiabatic ref and observed indicates that. There are several issues with figs such as averaging time, RWC but images do not show any, and Fig. 1b concept.

Why this work uses reff rather than use of full spectra for precip?  Ref can be 12 micron but no droplets can be at that size range. Is this for a climate study or cloud study? Author should emphasize this.

Ref usually is almost constant above 3000 m in the plots but you show like exp curves.

You mention rain water content but none of the images is shown for rain droplets.

You are using different time segments for each case that is not acceptable (see figures top titles).

You need to show time series of ref, lwc, nd and ni for 2 cases (polluted and clean and pitch angle). Your physical values can be affected by aircraft INS system conditions.

Fig. 1b; shows a conceptual sketch but I feel that is not true. Droplets cant grow continuously to the cloud top level. Any work that reference the similar conditions cannot be considered for publication.

**Major issues can be listed as**

1. DWC=0.01 g m-3 used for DWC to get rain out of the cloud.  What this is selected not clear. DWC can be 0.01 g m-3 but 5 droplets ot 100 of droplets. I argue that this is not an acceptable assumption for cloud structure?

2. You say that rain occurred at higher levels in more polluted clouds; no RH profile is shown or icing detector (pg 10) ; how do you know the cloud structure and particle phase? In fact none of your figures show rain or cloud droplets at larger sizes.

3. Figure 1b; this figure really mean nothing for me, and doesn't help anything about understanding of cloud macro structure. We know that aerosols can play significant role in cloud development, specifically for stratiform clouds. Below the cloud base aerosols act as nuclei, and they are activated within the cloud base, then dynamics of the system play a role faster that diffusional growth of particles. Certainly mixing happens at the base and edges of the system. Conditions are not adiabatic clearly. In fact if saturation is high enough and lots of moisture exist, many droplets forms. Do you show any vertical profile of the cloud T, w, and qv? In fact when cloud top increases at the upper troposphere not many particle reach to cloud top. Therefore, cloud depth is not only function of Na but vertical air velocity which is lowest at the cloud top at the upper troposphere. In fact because of collision/coalescence/aggregation processes, many particles fall down as precip before reaching the cloud top. Therefore, I will not accept your claims here for your work.

4. Fig.2; most of ref is between 11-14 micron for precip occurrence>50%; then how accurate to use ref as a precip condition. Precip is function of Nd, Vt, wa, and shape. Differences of these parameters at various T, can affect reff strongly. Suggests that cloud depth is not only function of Na but other physical and dynamical para,eters.

5; I like to see time series of Nd, Ni, RH, and ref for clean and clean cases. Use of reff for a cloud microphysics case study may not be appropriate. Ref can be 15 micron with no rain or with rain depends on psd.

6; fig 4 may not be needed, this is well known.

7; figure 5;

I like to see time series of Nd, showing about 2000 cm-3, also Na, please provide.

This figure shows that Nd is constant above 0C, and increases to warmer T. To me this is an artifact. Please also show LWC time series. Also show the weighted line on the plot.

Fig. 6a; thi is shown for 25 sec but others are for 656 sec, please be consistent.

Fig6b; looks like no changes in reff are for T<0C. For a given T difference is about 2-3 micron; it means that uncertainty in reff can be up to 3 micron, then how can someone use your method for cloud depth.

Fig. 7a; Combination of cip and cdp? Based on image shown, there are cloud droplets, CIP doesn't show anything.

Fig 7b; doesn't show droplets or drizzle? Where is the rain water come from?

Fig 7c; same again where is the RWC come from? Image shows ice crystals or rimed particles.

Fig. 9; again for Dc>3000 m, re is constant. This shows that re-Dc relationship doesn't hold.

Fig. 10a; Nd is constant within the cloud, looks like well mixed layer. Then how ref increases with depth? Flight is only 3 mins? Looks like ref increases with LWC. Can you show reff versus LWC?

Fig 11a; assumed that drizzle has sizes >50 micron why we don't have RWC? Or precip?

Again image shows ice crystals, not drizzle.

Fig. 11b; image shows graupel, where is the rain coming from, you say 0.27 g m-3?

Fig. 12:

I really have issues with this figure. If you look at the lines for a given Na say 1 or 100; and then check the Da. Not easy to make any conclusion based on this figure. In fact any uncertainty for less than about 20 cm-3 can results no discrimination among these cases.

Fig 13; same as before. How do you know they are droplets?

13a; again like rimed particles. RWC=0 but you show only 5 sec spectra, you are not consistent with time averages.

13b; again not rain water but mostly they are graupel or rimed particles. Only 4 second average? Not consistent. If CWC=0.247 and RWC=0.27 and DWC=0.16, then rain amount is more than cloud water content? How do you explain this over 4 sec?

Fig. 16; why there is no RWC? You have particles more than 100 micron but not rain?

Fig. 17; Dc versus Na relationship can't be a linear relationship. Increasing aerosols doesnt result in larger Dc values. Other factors should be considered e.g. updrafts, stability, diabatic heating etc.

Fig. 18; as noted previously above 3000 m, ref is almost constant but here you show all DC values increase with increasing ref exponentially. This figure should be discussed based on my previous points.

Other point I am not comfortable is that almost largest WC is at the cloud top which cant be correct. This is function of Vf, and you never mentioned this.

Discussion section should focus on the uncertainty of the observations and method. Presently it is not focusing on observations/analysis uncertainties such as averaging over various time segments.

Conclusion section should have clear findings that are related to observations collected from the field project. It can be good idea to list them.

Minor points are not considered in this point.

---

## Author Comment (AC1) · 7 Apr 2017

Reviewer B

Title: Aerosol concentration determines the height of warm rain and ice initiation in convective clouds over the Amazon basin. Submitted: Braga et al To: ACP Date: March 30 2017 Decision: Rejected Summary: This manuscript studies the effects of aerosol particle number concentration on the initiation of raindrops and ice hydrometeors in growing convective cumulus over the Amazon. Data from aerosol and cloud probes on board of the HALO aircraft are used. The values of the estimated $N_a$ at cloud base were applied to classify the atmospheric conditions where convective clouds developed as a function of aerosol particle number concentration (i.e., clean, polluted,

and very polluted regions). Main conclusion was that cloud depth, assuming adiabatic assumptions) is related to aerosol characteristics at the surface.

A: The reviewer writes: "Main conclusion was that cloud depth, assuming adiabatic assumptions) is related to aerosol characteristics at the surface." Where did the reviewer get such an idea from? We did not make or even hint to such a claim.

Major issues:

This paper claims that cloud depth and precip initiation are related to aerosol number concentrations at the cloud base.

A: We did not claim that depth is related to aerosol number concentrations at the cloud base. We show that the height above cloud depth for rain initiation is related to cloud base drop concentrations.

I argue that precip formation in the clouds are related to saturation rate, updrafts and shear (turbulence), T gradients, stability, and particle type.

A: Indeed, precipitation initiation depends on saturation, which is determined by cloud base updraft and CCN. T gradient and shear determines the updraft and vertical extent of the cloud. Particle type is water drops. So where does the reviewer disagree with us?

I really do not like simplified convective clouds dependency on only aerosol concentration at the cloud base. Doing this you ignored non-adiabatic terms that include radiation, turbulence, and mixing can play important role for cloud structure.

A: We have shown in a series of papers referenced here why the adiabatic approximation to adiabatic re works. We also show the goodness of the simplified relationships. The disliking of the reviewer of this reality is not relevant to its validity.

In fact difference between the adiabatic ref and observed indicates that. There are several issues with figs such as averaging time, RWC but images do not show any, and

Fig. 1b concept.

A: This comment is incomprehensible.

Why this work uses reff rather than use of full spectra for precip? Ref can be 12 micron but no droplets can be at that size range. Is this for a climate study or cloud study? Author should emphasize this.

A: This comment is incomprehensible.

Ref usually is almost constant above 3000 m in the plots but you show like exp curves.

A: The lines are the adiabatic re. This is stated very clearly.

You mention rain water content but none of the images is shown for rain droplets.

A: Images of rain drops are show in Figures 11a, 11b, 14a, 14b.

You are using different time segments for each case that is not acceptable (see figures top titles).

A: Which figures?

You need to show time series of ref, lwc, nd and ni for 2 cases (polluted and clean and pitch angle). Your physical values can be affected by aircraft INS system conditions.

A: We present here a higher level data for all cases. The raw data are available for anyone to inspect from the ACRIDICON archive.

Fig. 1b; shows a conceptual sketch but I feel that is not true. Droplets cant grow continuously to the cloud top level. Any work that reference the similar conditions cannot be considered for publication.

A: Before making such invalid statements so strongly, we recommend that the reviewer will read: Beals, M. J., Fugal, J. P., Shaw, R. A., Lu, J., Spuler, S. M. and Stith, J. L.: Holographic measurements of inhomogeneous cloud mixing at the centimeter scale, Science, 350(6256), 87–90, doi:10.1126/science.aab0751, 2015.

Major issues can be listed as

1. DWC=0.01 g m-3 used for DWC to get rain out of the cloud. What this is selected not clear. DWC can be 0.01 g m-3 but 5 droplets ot 100 of droplets. I argue that this is not an acceptable assumption for cloud structure?

A: The reviewer writes that the threshold is "to get rain out of the cloud". This is not what we wrote. This threshold is used for in situ rain initiation. We wrote that "DWC, defined here as the mass of the drops integrated over the diameter range of 75–250 $\mu$m (Freud and Rosenfeld, 2012)". The reason for limiting the maximum drop size to 250 $\mu$m is because it has a terminal fall velocity of 1 m/s. This minimizes the chance that the rain fell from above and not initiated near the penetration height.

2. You say that rain occurred at higher levels in more polluted clouds; no RH profile is shown or icing detector (pg 10) ; how do you know the cloud structure and particle phase? In fact none of your figures show rain or cloud droplets at larger sizes.

A: Images of rain drops are show in Figures 11a, 11b, 14a, 14b. RH is irrelevant within the cloud. Why should we use icing detector while we have liquid water content and temperature, thereby quantifying supercooled liquid water content? Cloud phase is known by the hot wire liquid water content and CIP images.

3. Figure 1b; this figure really mean nothing for me, and doesn't help anything about understanding of cloud macro structure. We know that aerosols can play significant role in cloud development, specifically for stratiform clouds. Below the cloud base aerosols act as nuclei, and they are activated within the cloud base, then dynamics of the system play a role faster that diffusional growth of particles. Certainly mixing happens at the base and edges of the system. Conditions are not adiabatic clearly. In fact if saturation is high enough and lots of moisture exist, many droplets forms. Do you show any vertical profile of the cloud T, w, and qv? In fact when cloud top increases at the upper troposphere not many particle reach to cloud top. Therefore, cloud depth is not only function of Na but vertical air velocity which is lowest at the cloud top at the

upper troposphere. In fact because of collision/coalescence/aggregation processes, many particles fall down as precip before reaching the cloud top. Therefore, I will not accept your claims here for your work.

A: The captions of Figure 1b read: "Flight patterns below and in convective clouds during the ACRIDICON-CHUVA campaign". The title of the paper reads: "Aerosol concentrations determine the height of warm rain and ice initiation in convective clouds over the Amazon basin". It should have been very clear to the reviewer that our study focuses in the lower parts of the clouds between cloud base and height of precipitation initiation. All the objections of the reviewer pertain to processes that affect cloud microstructure and precipitation above the height of precipitation initiation.

4. Fig.2; most of ref is between 11-14 micron for precip occurrence>50%; then how accurate to use ref as a precip condition. Precip is function of Nd, Vt, wa, and shape. Differences of these parameters at various T, can affect reff strongly. Suggests that cloud depth is not only function of Na but other physical and dynamical para,eters.

A: Indeed, cloud depth is not a function of Na, and we never made such a strange claim.

5; I like to see time series of Nd, Ni, RH, and ref for clean and clean cases. Use of reff for a cloud microphysics case study may not be appropriate. Ref can be 15 micron with no rain or with rain depends on psd.

A: How is this comment relevant to the subject of the study?

6; fig 4 may not be needed, this is well known. A: Its validity has to be demonstrated to the subject clouds. 7; figure 5; I like to see time series of Nd, showing about 2000 cm-3, also Na, please provide.

A: We don't see what will the time series of Nd would add to the points made in this study? We have too many figures already.

This figure shows that Nd is constant above 0C, and increases to warmer T. To me this

is an artifact. Please also show LWC time series. Also show the weighted line on the plot.

A: We don't agree. Figure 5 shows clearly a steady decrease of Nd with height.

Fig. 6a; thi is shown for 25 sec but others are for 656 sec, please be consistent.

A: At cloud base we have much more sampling than at the narrow towers aloft. We took all the samples that we could get with the aircraft.

Fig6b; looks like no changes in reff are for T<0C. For a given T difference is about 2-3 micron; it means that uncertainty in reff can be up to 3 micron, then how can someone use your method for cloud depth.

A: The reviewer repeats his misconception about what we do in this paper.

Fig. 7a; Combination of cip and cdp? Based on image shown, there are cloud droplets, CIP doesn't show anything.

A: The images are made by the CIP. Indeed, the CIP shows only cloud droplets.

Fig 7b; doesn't show droplets or drizzle? Where is the rain water come from?

A: All CIP images are converted as if they were rain.

Fig 7c; same again where is the RWC come from? Image shows ice crystals or rimed particles. A: See response to the previous comment.

Fig. 9; again for Dc>3000 m, re is constant. This shows that re-Dc relationship doesn't hold.

A: We don't agree. re increases up to Dc=4500 m. At that height ice forms on expense of the cloud droplets, as shown in Figure 18.

Fig. 10a; Nd is constant within the cloud, looks like well mixed layer. Then how ref increases with depth? Flight is only 3 mins? Looks like ref increases with LWC. Can you show reff versus LWC?

A: The lack of decrease of Nd with height could be related to some secondary droplet nucleation, as indicated in Figure 11a.

Fig 11a; assumed that drizzle has sizes >50 micron why we don't have RWC? Or precip? Again image shows ice crystals, not drizzle.

A: The image shows nice rounded rain drops. It even shows one pair of drops while coalescing!

Fig. 11b; image shows graupel, where is the rain coming from, you say 0.27 g m-3?

A: The image shows large rounded rain drops. It even shows one pair of drops while coalescing!

Fig. 12: I really have issues with this figure. If you look at the lines for a given Na say 1 or 100; and then check the Da. Not easy to make any conclusion based on this figure. In fact any uncertainty for less than about 20 cm-3 can results no discrimination among these cases.

A: The figure clearly shows that while total PCASP in AC12 is 10 times the PCASP in AC19, concentrations of particles > 2 ïA▪m are 5 time larger in AC19 compared to AC12.

Fig 13; same as before. How do you know they are droplets?

A: The CDP is rather insensitive to ice particles, and the hot wire LWC was similar to the CDP LWC.

13a; again like rimed particles. RWC=0 but you show only 5 sec spectra, you are not consistent with time averages.

A: The reviewer probably means Figure 14a. We take the sample size that we can get under these challenging flight conditions.

13b; again not rain water but mostly they are graupel or rimed particles. Only 4 second

average? Not consistent. If CWC=0.247 and RWC=0.27 and DWC=0.16, then rain amount is more than cloud water content? How do you explain this over 4 sec?

A: Again, we take what we can get. At this height most cloud water was converted to precipitation.

Fig. 16; why there is no RWC? You have particles more than 100 micron but not rain?

A: See the CIP precipitation images.

Fig. 17; Dc versus Na relationship can't be a linear relationship. Increasing aerosols doesnt result in larger Dc values. Other factors should be considered e.g. updrafts, stability, diabatic heating etc.

A: These are our results, whether we like it or not, and whether it support or contradicts our concepts. We refer the reviewer to read: the paper which we reference in this context: Freud, E. and Rosenfeld, D.: Linear relation between convective cloud drop number concentration and depth for rain initiation, J. Geophys. Res. Atmos., 117(2), 1–13, doi:10.1029/2011JD016457, 2012.

Fig. 18; as noted previously above 3000 m, ref is almost constant but here you show all DC values increase with increasing ref exponentially. This figure should be discussed based on my previous points.

A: The figure caption states that the lines are the adiabatic cloud drop effective radius (rea).

Other point I am not comfortable is that almost largest WC is at the cloud top which cant be correct. This is function of Vf, and you never mentioned this.

A: We don't agree strongly with the reviewer, for reasons that we explained in a previous response.

Discussion section should focus on the uncertainty of the observations and method. Presently it is not focusing on observations/analysis uncertainties such as averaging

over various time segments.

A: The averaging over various time segments is part of the analysis.

Conclusion section should have clear findings that are related to observations collected from the field project. It can be good idea to list them.

A: The findings will be listed in the new version.

Minor points are not considered in this point.

---

## Referee Comment (RC2) · Anonymous Referee #2 · 25 Apr 2017

A strength of this paper is the presentation of cloud in-situ observations that addresses the question of how convective clouds are influenced by changing the concentration of aerosols. The authors have determined that the height of rain initiation given by $D_i$ is approximately 5 * the number concentration of cloud droplets at cloud base, $N_d$. This type of study is needed for developing parametrisations. It is difficult to obtain such a wide range of values of $N_d$ using a single location for the study.

There are a number of problems with the paper however.

1. There are no details about the environment of the clouds, the effect of different cloud bases, or the dynamics and history of the clouds. For example, there should be information for each cloud pass about the distance from cloud top, the horizontal

distribution along the track of the vertical wind and the location where different measurements were made. Also, it is important to know the history of the cloud before that pass. Does the cloud and the aircraft indeed follow the pattern illustrated in Fig 2? None of the diagrams in Supplementary Fig 1 seem similar to Fig 1b. An example is Figs. 7b-c and 8. How far was the pass below the cloud top? Supercooled raindrops would only be observed in the updraft region.

2. Are there multiple thermals? These can be important for the development of raindrops?

3. Turbulence enhancement of collision and coalescence and the enhancement of droplet growth due to entrainment and mixing are not considered in the analysis. Both of these processes would change the simple relationship between $D_i$ and $N_d$. Giant and ultra-giant aerosols are mentioned, but likewise the analysis does not consider them carefully.

4. The initiation of ice particles in convective clouds is complicated. The analysis presented does not discuss the critical aspects of the problem.

5. The paper is poorly constructed. The font is far too small and diagrams used in discussions are in different documents. Also there are too many similar figures that show very little and are not properly discussed, while more detailed analysis and accompanying figures are missing.

Specific comments.

1. Abstract. "Rain initiation" is a loose term. The authors should be more precise.

"Initiated as ice hydrometeors". The word "initiated" is confusing.

Does "polluted conditions" include biomass burning?

Say why smaller cloud droplets froze at lower temperatures compared to the larger [cloud?] droplets in the [un - or less?] polluted cases. And give the sizes.

"Entrainment and mixing almost completely inhomogeneous". It is the mixing that is either homogeneous or inhomogeneous. A value of $r_e$ close to the $r_{ea}$ value is not sufficient information to conclude that the mixing process is inhomogeneous. There could be only dilution and no evaporation.

"Secondary nucleation". It's not true that this process will necessarily inhibit the formation of rain, and may indeed enhance it by providing more small droplets for collisions. Secondary nucleation does not mean that the larger cloud droplets are absent necessarily. Addition of more smaller droplets shifts the value of $r_e$ to smaller sizes.

2. p2, lines 87-90. It is incorrect to assume that the mixing processes is completely inhomogeneous. And if it was, the vertical profile of the cloud drop effective radius would most likely not follow an idealized adiabatic parcel; there would be broadening.

3. Line 95. I am not in favour of the wording "raindrops start to form" or "rain initiation" (discussed throughout the paper) since it is a stochastic process.

4. p3, lines 112-113. See #2 above, and furthermore, the vertical values of $r_e$ would not necessarily be constrained by $N_d$ at cloud base.

5. p3, line 138. Downdrafts can be significant. This is a good reason why the vertical velocity time series and the distance below cloud top should be shown.

6. p6, line 238. What was the wind direction and where were the clouds relative to the opening of the river in Fig1a?

7. p8, line 329. Is it possible to show the CDP size distribution just below cloud base?

8. p8, lines 339-340. The adiabatic parcel model would presumably use the aerosol size distribution, including the giant ccn, to initiate the cloud drops.

9. p8, line 344. Electrification is not part of this paper. This statement is conjecture.

10. p8, lines 345-346. It is not evident that downdrafts commence after rain starts. More detailed analysis is needed. 11. p8, lines 346-349. The sentence does not make

sense.

12. p8, lines 349-350. More evidence is required. The stronger updraft in Fig 3a of the supplement could be due to environmental conditions.

13. p9, line 358. There should be more discussion of Figs 13 and 14. Also, again, the images and size distributions should be presented in the context of the updraft structure. For example, the larger particle in the top panel of Fig 14a looks like a graupel particle. Has particle recognition been performed?

14. p9, line 360. As above, it is not right to have some of the figures used in the arguments in the supplementary material and others included in the paper. The diagrams are used in the same way as figures in the manuscript. The authors make a strong statement in the paper based on two cases with one of them shown in the supplementary material.

15. p9, line 360 Change the last word in the line (this) to "the" since Fig 5S shows plots constructed for two levels.

16. What is the evidence that the images in Fig 5aS are spherical? Some of the larger particles in Fig 5bS do look spherical, but it is not possible to tell for the smaller particles.

17. And now back to Fig 14b... Representative images from all parts of the cloud pass should be shown in all cases. What is the difference between almost spherical particles in Fig 14b and the same in Fig5S?

18. p9, lines 364 to 371. There is no discussion about the effect of the difference in cloud-base temperature. The main problem with the discussion, however, is there is no mention of the effects of inhomogeneous mixing, or even the decrease in N observed in the two flights: e.g. N_max at 10 deg is the same (500/cc) in AC18 (Fig 4aS) as in AC09 (Fig 13a).

19. p9, lines 371-374. It is not clear what is meant by the association with vertical

velocities? The rate of condensational growth does not depend on vertical velocity.

20. p9, lines 382 - 384. It is not a relative increase. It is perhaps more surprising that there is such a decrease in N_max between the lowest and next level in AC07 (Fig 5). More analysis should be shown to support suggestions made about secondary activation.

21. p9, lines 387 - 388. Should it not be the reduced rate of production of raindrops due to lower collision and coalescence efficiencies? There is no process of inhibition or suppression in the cloud.

22. p9, lines 388 - 389. 300 m is not a great distance when making aircraft passes. Is the result significant? What is the explanation?

23. p10, lines 393 - 394. As with so many other statements in the paper, more analysis should be presented to support the statement. What is the variation of CCN and updraft speeds at cloud base, for example?

24. I believe the discussion and conclusions should be edited based on the referee comments.

---

## Referee Comment (RC3) · D. Baumgardner (Referee) · 27 Apr 2017

The authors have presented the case that cloud active aerosols at cloud base are responsible for determining the cloud depth at which precipitation forms. As pointed out in the introduction, this is not a new discovery and has been investigated in many regions by many researchers other than the ones that are heavily referenced in this paper. Although the failure to be more inclusive in mentioning these other studies is not a fatal flaw in this paper, it does weaken its overall premise and conclusions. There are more serious issues that I would like addressed before this study is published.

Instrument issues

[Figure]

I could not find in either this paper or the Braga et al (2016) sufficient discussion on the processing of spectrometer measurements. In particular:

1) Coincidence corrections. Lance (2012) clearly shows that the CDP (unmodified with secondary mask) and CAS seriously undercount at > 500 cm-3. Lance (2012) says nothing about interarrival times and coincidence. Interarrival is used for shattering, so I don't understand the justification for not correcting the concentrations. Many of the concentrations reported > 1000 cm-3 will likely be at least 50% larger whch will seriously impact the derived LWC and subsequent Na.

2) In the images from the CIP, there are many out of focus droplets (donuts). The Korolev (2007) correction has to be done, otherwise the derived water content will be an overestimate and the height of precipitation might be incorrect.

3) Was the PCASP operated with a heated inlet? If so, corrections are needed to size distribution.

4) A fair amount of the paper is devoted to illustrating that the CAS and CDP compare within expected uncertainties. Given that this has already been done in the Braga et al. (2016), this is redundant and doesn't add much new information to the results.

Science Issues

5) Modify title please. The current title is misleading and not correct. It currently implies that all aerosols determine the depth of precipitation initiation. The results do not support this strong of a statement. Some types of aerosols play a role in determining the height of warm rain initiation, i.e. CCN/IN and their concentration have an impact as is clearly shown in this paper. A more accurate title might be "Further evidence for the impact of cloud base CCN/IN on the height of precipitation initiation"

6) The determination of Na needs much more explanation. The Na vs Precipitation depth is key to the conclusions and needs amplification. Why should the slope of the LWC vs Mv relationship with height provide a good estimate of Na? I understand that

LWC = Nd*Mv but this is not discussed, nor is how Mv is derived. In addition, all the plots that determine Na should be shown. If they are anything like the one shown in Braga et al 2016, Fig. 14a, there can be a very large spread in values of LWC at each Mv and subsequent uncertainty in the Rea. Fig. 15 in Braga et al (2016) clearly show that there is a lot of dispersion when comparing Na and Nd. The best fit line in their Fig. 14a does not appear to fit the points and certainly can't justify reporting Na to such precision.

7) Nothing is said about the uncertainty in the determination of level of precipitation wrt to vertical motions and where the precipitation actually initiated, i.e. it could have actually been below the level of measurement before being lofted upwards. This uncertainty can be estimated using the measured vertical motions.

8) Nothing is said about the time it takes to make the measurements at the various cloud levels and how these levels were selected. This will give some idea of the time during which the cloud is growing an how long it took to initiate precipitation.

9) Secondary nucleation is a very poor term because in a classical parcel model in an updraft, new particle nucleation occurs above cloud base until there are no more cloud active CCN at the level of SS. The implication here is that new CCN are being entrained and that is why the Nd increases with altititude, but this is likely not the case. When running a parcel model with a prescribed updraft and CCN spectra, the supersaturation increases in altitude as the parcel rises adiabatically and cools. The CCN will activate depending on their SS spectra and the available water. This needs revising.

10) The relationship Dr = 5*Na needs revising to take into account the data processing and uncertainties that I raise above, and needs an error bar.

---

## Author Comment (AC2) · 28 May 2017

Interactive response to reviewer's comment on "Aerosol concentrations determine the height of warm rain and ice initiation in convective clouds over the Amazon basin" by Ramon Campos Braga et al. Anonymous Referee #2

The structure is response after the quoted text of the reviewer. The authors thank the referee for the general comments and advices. Furthermore, the advices of the referee are highly appreciated as well as the very valuable and constructive suggestions to increase the quality of the manuscript. We tried to address the points requested by the reviewer to the paper be considered for publication.

[Figure]

Reviewer's text: A strength of this paper is the presentation of cloud in-situ observations that addresses the question of how convective clouds are influenced by changing the concentration of aerosols. The authors have determined that the height of rain initiation given by D_i is approximately 5 * the number concentration of cloud droplets at cloud base, N_d. This type of study is needed for developing parametrisations. It is difficult to obtain such a wide range of values of N_d using a single location for the study.

There are a number of problems with the paper however.

1. There are no details about the environment of the clouds, the effect of different cloud bases, or the dynamics and history of the clouds. For example, there should be information for each cloud pass about the distance from cloud top, the horizontal distribution along the track of the vertical wind and the location where different measurements were made. Also, it is important to know the history of the cloud before that pass. Does the cloud and the aircraft indeed follow the pattern illustrated in Fig 2? None of the diagrams in Supplementary Fig 1 seem similar to Fig 1b. An example is Figs. 7b-c and 8. How far was the pass below the cloud top? Supercooled raindrops would only be observed in the updraft region.

A: As we mentioned in the manuscript, convective clouds develop as clusters. During the flights, the flight scientist found a region with growing convective cumulus with different stages of development from very shallow to very deep clouds. The cloud passes occurred near the tops of growing convective cumulus were performed about 100-300 m below cloud top, as estimated by the flight scientist, assisted by the pilots (we already highlighted these details while describing Figure 1 at Introduction section). Figure 1 is highly schematic and illustrates the ideas that we measured successively higher clouds near their tops in the same cloud cluster. This type of cloud profile flight was adopted because it provides the closest information as possible about cloud particles formation as a function of cloud depth near the top of growing convective towers. In addition, this type of flight is the safest because the pilots could see the end of the cloud, avoiding the risk of penetrating the cores of Cb. The details about

cloud base temperature and heights are available at Table 3. The details about the location of measurements (Figures 1) and vertical velocities (Figures 3) are available at supplementary material.

2. Are there multiple thermals? These can be important for the development of raindrops?

A: No cloud is composed of a single thermal. But the question of multiple thermal is relevant for rain formation only if raindrop that was formed above the cloud penetration level was recirculated to the penetration level. Since we were fairly close to cloud tops, this was not an issue.

3. Turbulence enhancement of collision and coalescence and the enhancement of droplet growth due to entrainment and mixing are not considered in the analysis. Both of these processes would change the simple relationship between $D_i$ and $N_d$. Giant and ultra-giant aerosols are mentioned, but likewise the analysis does not consider them carefully.

A: The effects of turbulence are inherently fully taken into account in the observed cloud properties, whether we acknowledge it or not. Cloud drop size distribution did not show a separate large mode below the height where cloud drop effective radius > 14 $\mu$m, where coalescence leads to fast rain initiation. This means that GCCN did not have a major impact on rain initiation. In any case, as for the turbulence, the reported relationships account for the occurrence of GCCN whether we acknowledge it or not.

4. The initiation of ice particles in convective clouds is complicated. The analysis presented does not discuss the critical aspects of the problem.

A: We just show what we observe. We have insufficient information to go beyond that for ice initiation.

5. The paper is poorly constructed. The font is far too small and diagrams used in discussions are in different documents. Also there are too many similar figures that

show very little and are not properly discussed, while more detailed analysis and accompanying figures are missing.

A: The font used is the required by ACP. We have chosen show many figures at supplementary material to provide the opportunity to the reader check our findings.

Specific comments.

1. Abstract. "Rain initiation" is a loose term. The authors should be more precise."Initiated as ice hydrometeors". The word "initiated" is confusing.

A: Rain initiation is a commonly used term. With respect to ice, the text was changed to: "the first observed precipitation particles were ice hydrometeors".

Does "polluted conditions" include biomass burning?

A: Yes.

Say why smaller cloud droplets froze at lower temperatures compared to the larger [cloud?] droplets in the [un - or less?] polluted cases. And give the sizes.

A: Smaller droplets are less likely to contain immersion freezing ice nuclei due to their lower volume, and are also less likely to meet contact ice nuclei due to their lower surface area. Smaller droplets are also less likely to incur ice multiplication processes. The effective radius of cloud droplets (re) which freezes at -9.1 °C was $\sim$ 11.5 $\mu$m for flight AC07, while for flight AC13 re was $\sim$10.2 $\mu$m at -14.1 °C. Both flights were performed in polluted conditions over the deforestation arc at Amazon were similar type of aerosols are found (mostly from biomass burning).

"Entrainment and mixing almost completely inhomogeneous". It is the mixing that is either homogeneous or inhomogeneous. A value of r_e close to the r_ea value is not sufficient information to conclude that the mixing process is inhomogeneous. There could be only dilution and no evaporation.

A: The mixing is nearly inhomogeneous.

"Secondary nucleation". It's not true that this process will necessarily inhibit the formation of rain, and may indeed enhance it by providing more small droplets for collisions. Secondary nucleation does not mean that the larger cloud droplets are absent necessarily. Addition of more smaller droplets shifts the value of r_e to smaller sizes.

A: The text says: "Secondary nucleation of droplets on aerosol particles from biomass burning and air pollution reduced re below rea, which further inhibited the formation of raindrops and ice particles and resulted in even higher altitudes for rain and ice initiation." Secondary nucleation slows down the growth with height of the primary nucleated drops at cloud base. Therefore, we support that secondary nucleation prevent the formation of raindrops and ice particles before the first raindrop/ice starts to form. Once new droplets are nucleated above cloud base, the condensational growth rate of the cloud droplets decreases due to the larger competition for the water vapor available. Then, the resulting cloud droplets take more time (requiring thicker cloud depths) to reach larger sizes via condensation process and initiate coagulation or freezing.

2. p2, lines 87-90. It is incorrect to assume that the mixing processes is completely inhomogeneous. And if it was, the vertical profile of the cloud drop effective radius would most likely not follow an idealized adiabatic parcel; there would be broadening.

A: The deviations from complete inhomogeneous mixing explain the deviations shown in many of the figures between the adiabatic and actual re. This is why we claim it is near inhomogeneous, but not ideally so.

3. Line 95. I am not in favour of the wording "raindrops start to form" or "rain initiation" (discussed throughout the paper) since it is a stochastic process.

A: Indeed everything is stochastic, but since the formation rate of raindrops is proportional to the 5th power of re, and re increases with height, it is quite OK to mention height for rain initiation in practical terms. This is what this paper is about.

4. p3, lines 112-113. See #2 above, and furthermore, the vertical values of r_e would

not necessarily be constrained by N_d at cloud base.

A: The previous studies cited here support these characteristics for convective clouds.

5. p3, line 138. Downdrafts can be significant. This is a good reason why the vertical velocity time series and the distance below cloud top should be shown.

A: The reason for limiting the maximum drop size to 250 $\mu$m for drizzle water content calculation was because it has a terminal fall velocity of up to 1 m/s. As convective clouds are a turbulent medium large updrafts and downdrafts than 1 m/s were found (this is shown at Figures 3 in supplementary material). Again, this is just an extra measure of caution on top of measuring clouds near their tops, where nothing can come from much greater heights.

6. p6, line 238. What was the wind direction and where were the clouds relative to the opening of the river in Fig1a?

A: The wind direction was east-north-easterly. See the location of the flight in clouds in Figure 1, at the northeastern most edge of the flight track.

7. p8, line 329. Is it possible to show the CDP size distribution just below cloud base? A: Yes. We show in the revised version of the manuscript (Figure 12) the mean total number concentration calculated with PCASP and CCP-CDP for ~200 s of measurements below cloud base during the flights AC12, AC18 and AC19 . The flight AC19 over the Atlantic Ocean (where we indicate the possibility of GCCN) presents higher concentration of particles > 1 $\mu$m. This is observed with PCASP and CCP-CDP. The mean total number concentration of large particles measured with CCP-CDP over the ocean is about 10 times greater than observed inland. These values highlight the difference between the size of aerosol which can activate as cloud droplet over the ocean and over Amazon basin.

8. p8, lines 339-340. The adiabatic parcel model would presumably use the aerosol size distribution, including the giant ccn, to initiate the cloud drops.

A: We merely calculated the adiabatic water content and divided by cloud base drop concentrations to obtain adiabatic rv. The re was calculated by re=1.08 rv.

9. p8, line 344. Electrification is not part of this paper. This statement is conjecture.

A: The text was changed from "the low remaining amount of cloud water suppresses the development of cloud electrification" to "the low remaining amount of cloud water reduced a key ingredient for cloud electrification".

10. p8, lines 345-346. It is not evident that downdrafts commence after rain starts. More detailed analysis is needed.

A: Observed downdrafts starts to be evident or more intense above 1660 m after rain starts for flight AC19. Above 1660 m, more downdrafts than updrafts were observed.

11. p8, lines 346-349. The sentence does not make sense.

A: The sentence was rewritten as follows: "The values of vertical velocities measured at flight AC19 (clean region) were smaller than measured for flight AC07 (very polluted region). However, for both cases updrafts are more evident during droplets growth via condensation and downdrafts are most notable when precipitation particles are observed in the cloud."

12. p8, lines 349-350. More evidence is required. The stronger updraft in Fig 3a of the supplement could be due to environmental conditions.

A: We cannot exclude the enhancement of the updrafts due to environmental conditions. New text: "Strong updrafts ($\sim$10 m s-1) are observed in polluted cases after ice starts to form, probably due to the latent heat release during freezing processes. An alternative explanation of updraft enhancement due to environmental conditions in these cases cannot be excluded."

13. p9, line 358. There should be more discussion of Figs 13 and 14. Also, again, the images and size distributions should be presented in the context of the updraft

structure. For example, the larger particle in the top panel of Fig 14a looks like a graupel particle. Has particle recognition been performed?

A: The CCP-CIP images were used to distinguish raindrops and ice particles during cloud passes. The hydrometeor type is identified visually by their shapes. The phase of the smaller CCP-CIP particles cannot be distinguished. We believe that at Fig14a we have not graupel particle but some raindrops. The general characteristics of vertical velocities were discussed already in the paper.

14. p9, line 360. As above, it is not right to have some of the figures used in the arguments in the supplementary material and others included in the paper. The diagrams are used in the same way as figures in the manuscript. The authors make a strong statement in the paper based on two cases with one of them shown in the supplementary material.

A: We just highlight what we found. We have already many figures in the paper and some of them we had to send for supplementary material because of editorial objections. We present here this question to the co-editor.

15. p9, line 360 Change the last word in the line (this) to "the" since Fig 5S shows plots constructed for two levels.

A: OK. Thanks.

16. What is the evidence that the images in Fig 5aS are spherical? Some of the larger particles in Fig 5bS do look spherical, but it is not possible to tell for the smaller particles.

A: Indeed, this is why we claim the rain/ice initiation to the larger particles. As we mention before we could not recognize the phase of particles, but only their shapes for precipitable ones.

17. And now back to Fig 14b... Representative images from all parts of the cloud pass should be shown in all cases. What is the difference between almost spherical

particles in Fig 14b and the same in Fig5S? A: Image 14b shows the first pass in which ice hydrometeors are observed mixed with supercooled rain drops. Rain drops were observed also at lower levels, as summarized in Figure 18. Figure 5S shows that in AC18 first rain drops are observed at the -5.7 C isotherm, and that they still remain liquid, or at least spherical, at the -11.4 C isotherm.

18. p9, lines 364 to 371. There is no discussion about the effect of the difference in cloud-base temperature. The main problem with the discussion, however, is there is no mention of the effects of inhomogeneous mixing, or even the decrease in N observed in the two flights: e.g. $N_{max}$ at 10 deg is the same (500/cc) in AC18 (Fig 4aS) as in AC09 (Fig 13a).

A: The inhomogeneous mixing is implicit when re is closer to rea by the reasons that we mentioned before. At the same temperature Nmax can be the same, but re is larger for AC09, because the 10 C isotherm is higher above cloud base in AC09 compared to AC18.

19. p9, lines 371-374. It is not clear what is meant by the association with vertical velocities? The rate of condensational growth does not depend on vertical velocity. A: The sentence was removed.

20. p9, lines 382 - 384. It is not a relative increase. It is perhaps more surprising that there is such a decrease in $N_{max}$ between the lowest and next level in AC07 (Fig 5). More analysis should be shown to support suggestions made about secondary activation.

A: We don't assert that it is due to secondary activation. We just raise the possibility.

21. p9, lines 387 - 388. Should it not be the reduced rate of production of raindrops due to lower collision and coalescence efficiencies? There is no process of inhibition or suppression in the cloud.

A: The text was modified to: "These results highlight the role of aerosols in inhibition of

raindrop formation due to inducing a larger Nd and respective lower re, which leads to suppression of collision and coalescence processes in very polluted regions."

22. p9, lines 388 - 389. 300 m is not a great distance when making aircraft passes. Is the result significant? What is the explanation? A: Indeed, it is not a significant difference, but nevertheless in the right direction, this is worth noting.

23. p10, lines 393 - 394. As with so many other statements in the paper, more analysis should be presented to support the statement. What is the variation of CCN and updraft speeds at cloud base, for example?

A: The acceleration of updrafts above the height of cloud base increases supersaturation and thus can induce secondary drop activation. For flights which we observed the increase of Nd with height, high aerosol concentration was observed indicating increased likelihood of secondary nucleation of droplets above cloud base.

24. I believe the discussion and conclusions should be edited based on the referee comments.

A: The conclusions did not change.
* * *

---

## Author Comment (AC3) · 28 May 2017

Interactive response to reviewer's comment on "Aerosol concentrations determine the height of warm rain and ice initiation in convective clouds over the Amazon basin" by Ramon Campos Braga et al.

The structure is response after the quoted text of the reviewer.

The authors thank Darrel Baumgardner for the general comments and advices. Furthermore, the advices of the referee are highly appreciated as well as the very valuable and constructive suggestions to increase the quality of the manuscript. We tried to address the points requested by the reviewer to the paper be considered for publication.

Reviewer's text: The authors have presented the case that cloud active aerosols at cloud base are responsible for determining the cloud depth at which precipitation forms. As pointed out in the introduction, this is not a new discovery and has been investigated in many regions by many researchers other than the ones that are heavily referenced in this paper. Although the failure to be more inclusive in mentioning these other studies is not a fatal flaw in this paper, it does weaken its overall premise and conclusions. There are more serious issues that I would like addressed before this study is published.

A: Four references to rain initiation were added.

Instrument issues

I could not find in either this paper or the Braga et al (2016) sufficient discussion on the processing of spectrometer measurements. In particular:

1) Coincidence corrections. Lance (2012) clearly shows that the CDP (unmodified with secondary mask) and CAS seriously undercount at > 500 cm-3. Lance (2012) says nothing about interarrival times and coincidence. Interarrival is used for shattering, so I don't understand the justification for not correcting the concentrations. Many of the concentrations reported > 1000 cm-3 will likely be at least 50% larger which will seriously impact the derived LWC and subsequent Na.

A: Both instruments have different set ups compared to the configuration described in Lance et al. Specifically for the CAS, the pin hole in front of the sizing detector was changed to a smaller diameter. This significantly reduced the number of coincident particles. The analysis addressing coincidence was done following the paper by Lance (2012). We compared the LWC from the hotwire and the PSDs assuming spherical particles. If coincidence occurs in the sampling volume, larger but fewer particles would have been detected, thus the LWC from the particle probes would be higher for higher number concentrations. The CAS showed rather lower LWC than the hotwire at higher number concentrations (> 1000 cm-3), which stands in contrast to the observations by Lance et al. Furthermore, we looked at the Poissonian probability density function

of the inter arrival times at high number concentrations (2000 cm-3). If coincidence occurs then a significant fraction of the inter arrival times should be at the lower end of the distribution (short inter arrival times) or even beyond the time resolution of the instrument. We could not find a significant fraction (< 5 %) at the lower end of the inter arrival time distribution. The CDP additionally measures the transit time. The transit time did not increase (unlike like the CDP in Lance et al. did) with the number of particles detected up to number concentrations of 1500 cm-3. Further, the good agreement of CAS and CDP regarding the number concentrations shows, if coincidence was an issue, it would be of similar magnitude for both instruments, which is very unlikely. The three independent analysis methods in addition to the good comparison of the probes proves that there are no indications of coincidence in our measurements.

2) In the images from the CIP, there are many out of focus droplets (donuts). The Korolev (2007) correction has to be done, otherwise the derived water content will be an overestimate and the height of precipitation might be incorrect. A: For the data processing of the CCP measurements, ice was assumed as the predominant particle phase in the mixed-state cloud conditions that were mainly given throughout the ACRIDICON CHUVA campaign. The ice assumption causes all images of droplets and ice particles to be treated and considered as particles (apart from shattering-induced particles) but the Poisson spot correction is then excluded. The Korolev correction is defined for liquid drops only and the SODA image processing disables this correction process once the ice-phase is selected. The assumption of ice density instead of water density implies a slight overestimation ($\sim$10 %) of the calculated rain water content for particles greater than 75 $\mu$m. This will be highlighted in the manuscript.

3) Was the PCASP operated with a heated inlet? If so, corrections are needed to size distribution.

A: No. PCASP was not operated with a heated inlet. We add a comment about this in the instrument description.

4) A fair amount of the paper is devoted to illustrating that the CAS and CDP compare within expected uncertainties. Given that this has already been done in the Braga et al. (2016), this is redundant and doesn't add much new information to the results.

A: We did not compare effective radius (re) calculated with CAS-DPOL and CCP-CDP as a function of mean volume radius and precipitation probability for at Braga et al. (2016). The results show that even with agreement the threshold of re for rain initiation is about 1 $\mu$m smaller for CAS-DPOL in comparison with CCP-CDP.

Science Issues

5) Modify title please. The current title is misleading and not correct. It currently implies that all aerosols determine the depth of precipitation initiation. The results do not support this strong of a statement. Some types of aerosols play a role in determining the height of warm rain initiation, i.e. CCN/IN and their concentration have an impact as is clearly shown in this paper. A more accurate title might be "Further evidence for the impact of cloud base CCN/IN on the height of precipitation initiation"

A: We have changed the title to: Further evidence for CCN aerosol concentrations determining the height of warm rain and ice initiation in convective clouds over the Amazon basin

6) The determination of Na needs much more explanation. The Na vs Precipitation depth is key to the conclusions and needs amplification. Why should the slope of the LWC vs Mv relationship with height provide a good estimate of Na? I understand that LWC = Nd*Mv but this is not discussed, nor is how Mv is derived. In addition, all the plots that determine Na should be shown. If they are anything like the one shown in Braga et al 2016, Fig. 14a, there can be a very large spread in values of LWC at each Mv and subsequent uncertainty in the Rea. Fig. 15 in Braga et al (2016) clearly show that there is a lot of dispersion when comparing Na and Nd. The best fit line in their Fig. 14a does not appear to fit the points and certainly can't justify reporting Na to such precision.

**[ACPD](ACPD)**

Interactive
comment

A: The reviewer wrote: "I understand that LWC = Nd*Mv". This is not quite so. The right expression is LWCa = Nda*Mva, where all are the adiabatic values. The whole idea of the methodology is that the actual re is similar to rea - the adiabatic effective radius, due to the nearly inhomogeneous nature of the mixing. The mixing does decorrelate LWC strongly from LWCa, while keeping re well correlated with rea. The methodology which use LWC vs. Mv relationship with height to estimate Na is well tested and validated at Freud and Rosenfeld (2011). The Na estimate is also explained and tested at Braga et al. (2016). Indeed there are uncertainties related to Na estimated mostly related when secondary nucleation takes place. The model does not predict that Nd increases with height, but decrease due to coalescence and inhomogeneous cloud mixing. The results suggest the occurrence of secondary activation with different strengths during flights AC08, AC12, AC13 and AC20 (see figures attached). Large updrafts were measured above cloud base during these flights which increase supersaturation inducing secondary activation. The increase of Nd with height was observed mostly when large aerosol amount was measured with PCASP and UHSAS above cloud base height. However, the estimation of Na have shown to be useful to discriminate clean from polluted environments and predict the height for rain initiation.

7) Nothing is said about the uncertainty in the determination of level of precipitation wrt to vertical motions and where the precipitation actually initiated, i.e. it could have actually been below the level of measurement before being lofted upwards. This uncertainty can be estimated using the measured vertical motions.

A: Doing that would require information that we don't have about the rate of rain formation with height, and will constitute a circular argumentation. The scatter in Figure 17 is the best that we can do for illustrating the uncertainty.

8) Nothing is said about the time it takes to make the measurements at the various cloud levels and how these levels were selected. This will give some idea of the time during which the cloud is growing an how long it took to initiate precipitation.

A: Since the measurements were not following individual growing cloud towers, these times would not advance such knowledge.

9) Secondary nucleation is a very poor term because in a classical parcel model in an updraft, new particle nucleation occurs above cloud base until there are no more cloud active CCN at the level of SS. The implication here is that new CCN are being entrained and that is why the Nd increases with altititude, but this is likely not the case. When running a parcel model with a prescribed updraft and CCN spectra, the supersaturation increases in altitude as the parcel rises adiabatically and cools. The CCN will activate depending on their SS spectra and the available water. This needs revising.

A: The secondary CCN activation was observed mainly in cloud segments with updrafts that were much stronger than at cloud base. This supports the narrow definition of secondary activation as defined by the reviewer. However, we do not exclude the possibility of additional CCN being entrained and activated above cloud base.

10) The relationship Dr = 5*Na needs revising to take into account the data processing and uncertainties that I raise above, and needs an error bar.

A: The uncertainty of Na calculation with CDP (14 %) is now included in the linear relationship. The linear relationship including Na uncertainty is $D\_r = (5 \pm 0.7) \cdot Na$ .

---

## Referee Report (RR1)

**Title:**

**Further evidence for CCN aerosol concentrations determining the height of warm rain and ice initiation in convective clouds over the Amazon basin**

**Date: Aug 2 2017**

Rev. 1

Decision: rejection

**General comments**

This is a second round of review process for me. I don't want to review this paper again if further corrections are needed. I feel that this paper needs to be better organized and data analysis should be done properly. I still see that my main concerns are not corrected properly, and text is not improved. Authors insist to keep text same as before after making some minor corrections. I will not go over again my points here but emphasized some points below:

Mainly figures do not represent what is said in the text.

Data analysis does not reflect proper averaging times.

Comparisons with adiabatic calculations were not discussed properly.

**Title says: Further evidence for CCN aerosol concentrations determining the height of warm rain and ice initiation in convective clouds over the Amazon basin.**

I don't believe that only CCN determines height of warm rain and ice initiation processes…. This is a misleading title, in fact IN or both CCN and IN at high levels plays an important role and never mentioned. How can adiabatic calculations be made for ref for comparisons or LWC, and used for comparisons if non-adiabatic terms are clearly dominant in a convective process at the certain phases of the storm??? At least 10 times adiabatic values are used for comparisons, in fact, convective clouds may deviate significantly from an adiabatic assumption.

Figures asked to be generated or removed are not included or deleted.

Fig. 1b is conceptually wrong and not true, if this is the case, what argument you have in the text. I feel this is pointed out previously.

Fits are provided may not represent data points distribution properly.

Icing detector and RH plots as indicated before are needed, without them you cant really say particles as droplets or ice crystals when particles are not falling above.

Images do not show statistically significant data points for droplets or rains. How do you know they are liquid?

Some text related to CDP and/or CIP probe are not correct.

How collisions and coalescence processes affect cloud macro and micro-properties, how about turbulence?

Conclusions are not provided properly and explained, this was mentioned in my previous review. I like to see not only Na characteristics affecting cloud properties but other parameters such as updrafts and mixing as well radiative processes. Conclusions provided are like for a conference paper.

In addition to my above points, as well as based on my previous points and responses received, I will not suggest this work for a publication in ACP.

---

## Referee Report (RR2)

Manuscript review
Further evidence for CCN aerosol concentrations determining the height of warm rain and ice initiation in convective clouds over the Amazon basin

Braga et al.

**General comments**

This manuscript is a revised version of a previous submission. Although the content has been somewhat improved from the original version, it is still not ready to be released for publication until most of my previous comments are better addressed than they were in the authors' original response. In addition, upon further evaluation of the paper, I have found other sections that need further clarification or modification.

To put my concerns about this paper in context, I will use this sentence in the introduction to highlight what I think is a potentially hazardous approach by modelers: ***"These parameterizations need to represent in simplified form the complex chain of events that occur in clouds."*** I understand the motivation for parameterizations of cloud microphysical processes in global scale models, and depending on the application of the parameterizations, simplifications can be implemented without seriously misrepresenting the cloud microphysics. In other cases, however, oversimplifications will produce results that are erroneous and misleading. Perhaps it was the first author's wording, and doesn't reflect the thinking of the other authors. If that is the case, then just remove this sentence because I doubt that any serious modeler would agree that the complex chain of events that occur in clouds can ever be represented by simplified parameterization.

What further worries me is that the results of this study, as currently presented, suggest that the height of precipitation initiation in any convective cloud can be represented by a single, integer multiplier of the cloud base droplet concentration. In the paper's current form, there are a limited number of caveats to this statement given in the abstract and the summary and yet the caveats are significant and need to be listed just as boldly as the simplified relationship between precipitation initiation height and cloud base droplet concentration. The motivation for forcing such a relationship is not obvious in the current paper but in reading the paper by Freud and Rosenfeld (2012), it seems that the driving force behind these parameterizations is to extract more information from satellite data about cloud properties. This is a worthy objective but not if accomplished at the cost of diminished scientific robustness.

In my comments below, I will further highlight and provide details of where this paper will need to be improved before I will accept it for publication. The bottom line is that I can't and will not allow the publication of a parameterization that can so easily be misused until it is properly justified.

**Editorial comments**

In my opinion it is my responsibility as the reviewer, a responsibility that I take very seriously, to help improve the paper I am reviewing by 1) identifying technical and factual errors or omissions,

2) requesting clarification when needed and 3) suggesting modifications that help to solidify the hypothesis put forward. In my follow-up review of the revised manuscript, I annotate my remarks with RR1, RR2 or RR3 so that the authors understand the motivation for my comments and suggestions in relationship to how I view my reviewer responsibility (RR).

The authors have sufficiently addressed my concern about the coincidence error losses but all my other comments lack responses that adequately address my comments or concerns. Giving the authors the benefit of the doubt, I will assume the responsibility for not having stated clearly enough the nature of my concerns. Hence I will repeat them here, but with enough detail that there should not be any confusion as to the nature of my comment and how I expect it to be addressed. I will also be more clear about the seriousness of my comments, i.e. those for which I expect concrete changes to the manuscript and those where I will accept lesser modifications as long as my comments receive a reasonable and scientifically defensible response. My comments are not necessarily in the same order as they were presented in the first review.

**Specific Comments, Questions and Suggestions**

1) Error propagation

In Braga et al. (2016) and relatively comprehensive uncertainty analysis is conducted of the number concentrations and sizing by the light scattering and imaging probes; however, this analysis is not taken into account in the current paper to estimate the expected uncertainty in determining $Na$, $Re$, $Rea$, $CLWC$, $Mv$, etc. This is a major omission (RR1) that must be rectified.

In their original response to my request to propagate the measurement uncertainties into the derivation of the $Dr$ vs $Na$ relationship, they state:

*"A: The uncertainty of Na calculation with CDP (14 %) is now included in the linear relationship. The linear relationship including Na uncertainty is $D\_r=(5\pm0.7)\cdot Na$".*

This is inadequate since it does not take into account the very large variations in the $CLWCa$ vs $Mv$ relationship that was shown in the Braga et al (2016) paper (Fig. 14a, redrawn below for a single flight), nor does it explain that an additional 30% unexplained correction has been applied to the $Na$ as also explained by Braga et al. (2016), i.e. "However, this methodology does not account for cloud mixing losses from droplet evaporation, and the Na estimates commonly overestimate the expected Nd by 30% (Freud et al., 2011). Therefore, in calculating Na we applied this 30% correction.".

The $Mv$ will have at least an uncertainty of 50% since it is derived from $rv$ (I think, although nowhere is $Mv$ ever explained how it is derived). In Fig. 14a, the best fit doesn't even go through most of the points at low or high $LWCa$ so how can a concentration be derived better than to the nearest 100 cm$^{-3}$, much less to the nearest 10$^{th}$! The supplementary material needs to show the same type of figure as Fig. 14 for all flights so that we can actually see how much deviation there is.

The uncertainty in $Na$ then propagates into the $r_{ea}$ and the derived $r_e$ will have an uncertainty of more than 20% if the uncertainties in size are properly propagated.

Finally, there is a very large uncertainty in the Dr, not only in its derivation of the DWC but also in the actual cloud depth where it is measured. There is an uncertainty of at least a maximum of $D_r$-$D_{r-1}$ where $D_{r-1}$ is the cloud depth at the previous cloud penetration where the $r_{13}$ threshold was not exceeded. This is because the re threshold could have been exceeded in the rising air mass at any point between the current and previous cloud penetration. This is not addressed at all and is a serious omission.

In summary, the ±0.7 value is unsupported and requires a much more robust derivation than is currently given.

[Figure]

Figure 14a from Braga et al. (2016)

2. Data processing (RR1)

In my previous review I stated: "In the images from the CIP, there are many out of focus droplets (donuts). The Korolev (2007) correction has to be done, otherwise the derived water content will be an overestimate and the height of precipitation might be incorrect."

To which the authors replied: *"A: For the data processing of the CCP measurements, ice was assumed as the predominant particle phase in the mixed-state cloud conditions that were mainly given throughout the ACRIDICON CHUVA campaign. The ice assumption causes all images of droplets and ice particles to be treated and considered as particles (apart from shattering-*

*induced particles) but the Poisson spot correction is then excluded. The Korolev correction is defined for liquid drops only and the SODA image processing disables this correction process once the ice-phase is selected. The assumption of ice density instead of water density implies a slight overestimation (_10 %) of the calculated rain water content for particles greater than 75 μm. This will be highlighted in the manuscript."*

I understand the issue of mixed phase, however, there are three flights, AC09, AC18 and AC19, that have a Dr that is obviously derived from all water images, so clearly the out-of-focus correction can and should be done. In addition, 2.2.1, it is stated "In this study, we deduced the existence of ice from the occurrence of visually non-spherical shapes of the shadows." Hence, water droplets can be detected in even mixed phase conditions so that these spherical images can be corrected to derive the true DWC. To not do so will bias the derived Dr and the subsequent rain initiation level. (RR1).

3. Comparisons of CDP vs CAS-POL and Re vs Rv (RR2 and RR3))

I will reiterate my suggestion that the CDP vs CAS-POL comparisons in Figs. 3 and 4 be removed. First of all, Braga et al. (2016) have more than shown how well these two instruments compare and secondly, there is no reason given in the text of the purpose of comparing Re vs Rv. Freud and Rosenfeld discuss this but it has no relevance in the current paper. These figures and the associated descriptive text should be removed.

4. Further clarification of the derivation of Na (RR2)

After reading Freud and Rosenfeld (2012) I was able to decipher the rather cryptic discussion of how Na is derived; however, the response to my original request for clarification raises a number of additional questions and related concerns. Here is the original response to my request for clarification:

*"A: The reviewer wrote: "I understand that LWC = Nd\*Mv". This is not quite so. The right expression is LWCa = Nda\*Mva, where all are the adiabatic values. The whole idea of the methodology is that the actual re is similar to rea - the adiabatic effective radius, due to the nearly inhomogeneous nature of the mixing. The mixing does decorrelate LWC strongly from LWCa, while keeping re well correlated with rea. The methodology which use LWC vs. Mv relationship with height to estimate Na is well tested and validated at Freud and Rosenfeld (2011). The Na estimate is also explained and tested at Braga et al. (2016). Indeed there are uncertainties related to Na estimated mostly related when secondary nucleation takes place. The model does not predict that Nd increases with height, but decrease due to coalescence and inhomogeneous cloud mixing. The results suggest the occurrence of secondary activation with different strengths during flights AC08, AC12, AC13 and AC20 (see figures attached). Large updrafts were measured above cloud base during these flights which increase supersaturation inducing secondary activation.*

This response seems contrary to lines 203: "b) The *Na* at cloud base is estimated through the vertical profile of *re*." and line 245: "The *Na* for the convective clusters is estimated based on the slope between the calculated CWC and the mean volume droplet(*Mv*)". This seems to indicate that the Na is being derived from measurements not the adiabatic values. In addition, Mv is never

defined, although I was finally able to deduce that it is somehow being derived from Rv, but this is never made clear.

This response also raises the issue of how adiabatic LWC and Re are derived, i.e. one must know not only the cloud base temperature and pressure, but vertical profiles of temperature, pressure and mixing ratio, as well. What vertical profiles are being used? This needs to be described at the very beginning, as well as the uncertainties involved, i.e. how much do these vertical profiles change over time, especially during the time period of the measurements from cloud base to cloud tops?

Then the comment regarding "Secondary nucleation" brought forward one of my other critiques regarding of the use of this term. Here is my previous comment:

"9) Secondary nucleation is a very poor term because in a classical parcel model in an updraft, new particle nucleation occurs above cloud base until there are no more cloud active CCN at the level of SS. The implication here is that new CCN are being entrained and that is why the Nd increases with altitude, but this is likely not the case. When running a parcel model with a prescribed updraft and CCN spectra, the supersaturation increases in altitude as the parcel rises adiabatically and cools. The CCN will activate depending on their SS spectra and the available water. This needs revising."

And this was the response:

*"A: The secondary CCN activation was observed mainly in cloud segments with updrafts that were much stronger than at cloud base. This supports the narrow definition of secondary activation as defined by the reviewer. However, we do not exclude the possibility of additional CCN being entrained and activated above cloud base."*

This response suggests that the authors did not understand my criticism, i.e. that I did not want them to use the term "Secondary Nucleation" or Secondary Activation" ANYWHERE in this manuscript for the reasons I stated. They are free to use the term "Additional activation/nucleation" or "Continuing activation/nucleation" but not "Secondary". If they wish to explain the possibility of entrainment of CCN above the cloud base, they are free to do so, as long as this would described as "additional" not "secondary"

5. Precipitation particles coming from below the measurement altitude (RR1)

This concerns a comment I presented above and the response of the authors to a similar comment I had made in my previous review. That comment:

"7) Nothing is said about the uncertainty in the determination of level of precipitation wrt to vertical motions and where the precipitation actually initiated, i.e. it could have actually been below the level of measurement before being lofted upwards. This uncertainty can be estimated using the measured vertical motions."

The authors' response:

*"A: Doing that would require information that we don't have about the rate of rain formation with height, and will constitute a circular argumentation. The scatter in Figure 17 is the best that we can do for illustrating the uncertainty."*

This response did not address my concern. In my comments above about the uncertainty in Dr, at the least, the maximum uncertainty can be estimated as the distance between the two measurement levels with and without the threshold being exceeded, e.g. if the Re and DWC had not been exceeded at the 3000 m level but is exceeded at the 4000 m level, given that it might have been exceeded at the 3001 m level that wasn't measured, the uncertainty in this case would be 1000 m. Hence, Table 3 has to have uncertainty bars on ALL of the quantities listed, including the Na.

6. Information on time between flight legs through clouds (RR3)

My previous comment:

"8) Nothing is said about the time it takes to make the measurements at the various cloud levels and how these levels were selected. This will give some idea of the time during which the cloud is growing an how long it took to initiate precipitation."

The authors' response:

*"A: Since the measurements were not following individual growing cloud towers, these times would not advance such knowledge."*

First of all, the first part of the response seems to fly in the face of what is written on line 126: "The aircraft obtained a composite vertical profile by penetrating young and rising convective elements, typically some 100-300 m below their tops." Secondly, I think that the amount of time from cloud base to each flight level is very germane to the question of how long it takes to initiate precipitation, given the discussion early on concerning the rate at which precipitation forms. Hence, in Table III I want to see a column that include the time after cloud base measurements that precipitation was identified. In this same Table I want to see vertical velocity added to the cloud base conditions, another column with the number of levels that were sampled for this date and another column that shows the maximum vertical velocity. This latter request is because it seems to me that from the Supplementary material, almost every flight had vertical velocities above cloud base that were larger than those at cloud base. This would invalidate the assumptions that are needed to use Na as a predictor of Dr.

7. Additional comments

In going through the revised manuscript, I had some remaining questions/comments:

Line 55: The authors use -36C as the threshold for homogeneous freezing but -38C is the value that is most commonly used, hence -36C should be changed to -37C, according to how they state this threshold.

Line 105: "The *Na* is calculated from *Na= CWCa/Mva*, where*CWCa*is the 105 adiabatic cloud water content (CWCa) as calculated from cloud base pressure and temperature, and *Mva* is the adiabatic mean volume droplet mass, as approximated from the actually measured mean volume droplet mass(*Mv*) by the cloud probe DSDs obtained during the cloud profiling measurements."

This further confuses me as it implies a mixture of adiabatic and observed quantities, i.e. why is Mva being approximated from the measured mean volume droplet mass, and in addition, why is Mv never explicitly defined in an equation?

Line 143: "The DWC is defined as the mass of the drops integrated over the diameter range of 75–250 µm (Freud and Rosenfeld, 2012)." Does this mean that any mass beyond 250 µm is excluded?"

Line 275: "The precipitation probability is calculated by integrating the measured DSDs exceeding certain DWC thresholds.". This statement needs a great deal of clarification. First of all, why are these being called "precipitation probabilities" and secondly, are the DSDs being integrated up to the Re that produces the threshold DWC?  Please be more explicit.

Line 295: This statement about the relationship between Dc and Rea needs to be qualified with the caveat that it only holds under certain strict conditions, e.g. maximum updraft at cloud base, no continuing activation of CCN above cloud base, etc.

Line 301: The maximum concentration was 2000 cm$^{-3}$, but what Nds were used to compare with the Nas?

Figure 8: Remove it. There is no relevant information here.

Discussion

Given that the end objective of this study is to support the Freud and Rosenfeld (2012) parameterization so that it can be used by the satellite community to derive microphysical properties, the discussion needs to be much more clear about the robustness of the Dr vs Na relationship with recommendations as to when is can be used and when it shouldn't be used.

The caveats that limit the use of the Dr vs Na relationship need to be bullets in the Summary so that future use of this parameterization is not used indiscriminately.

---

## Referee Report (RR3)

Title: Further evidence for CCN aerosol concentration

By Braga et al

Date: Oct 21 2017

Decision: Rejected

I do not think that this paper can be improved over a short time period. Issues are very serious for data analysis and organizing of the paper, and simplification of a research area related to aerosol-precipitation and cloud depth.

General comments

This paper focuses on precip initiation related to CCN in convective clouds, and its impact on cloud depth. This is an interesting research area but manuscript needs major improvements related to introduction, observations, and method sections, as well as figures. My previous points are not considered properly in this new version. Provided figures are not representative of the real conditions because of analysis issues. The Nd, LWC, and ref time series are not shown for at least for 1-flight that can be used to interpret analysis. How vertical profiles obtained using 1 Hz data are not presented properly for averages. Nd and ref profiles using T in y axis should be weighted over similar dNd or dref intervals, and sd should be shown. Then fits should be applied because of inhomogeneity of the data collected. Why LWC profiles are not given is another issue in this paper. If you are talking about adiabatic assumption, why LWCa and LWCm are not shown? In supplementary plots, specifically vertical air velocity plots means nothing unless aircraft position angles are shown, and time periods during turns are taken out of the analysis? Better provide time series of segments to represent real conditions.

Flight patterns are shown taken along the edges of Cbs where heavy mixing occurs, and suggest that data don't represent adiabatic conditions based on figure you provided; probably, because of this reason, LWCa and LWCm profiles are not likely provided. Also, some figures represent only 3 seconds of intervals (Fig. 14) or some represent only 247 seconds (Fig. 11). Therefore, one cannot make a decision or provide objective analysis. Also, in Fig. 11, Nd decreases with decreasing T (or Z), but ref increases with increasing Z. Usually reff increases with decreasing Nd (see Gultepe et al 1996). In this figure, # of points is about 247 (assuming 100 m/sec and using 247 sec flight duration). Can you explain how aircraft can climb up from 20C to -15C over 4 minutes? How accurate are these measurements when aircraft makes quick returns?

Na in same figure (11) is given as Nd which is 566 cm-3. But Nd is found less than 600 cm-3 at the cloud base, are you using instantaneous value here? Why not use averages ?. This analysis is very confusing and figs are not given properly. Weighted averages should be used not a max value or min value. In fact, Fig. 11a should show bin averages of Nd (dNd~50 cm-3) or similar

ones. In that plot, you also need to show LWC-T plot. Without this info, it is not possible to make fair conclusions.

Clearly, sampling of data and presenting observations are not presented properly (this is also valid for Fig. 12 in which dt=5 sec and 4 sec.). Gultepe et al 1999 suggested that variability over an averaging scale is an important issue for analyzing observations.

An article on aerosol effect on precip is also given by Menon and Delgenio (2007, chapter 3 of DOI: 10.1017/CBO9780511619472.005) where regional changes in precipitation, examined over India and China, were found to be *related to the amount of atmospheric heating, with higher atmospheric fluxes corresponding to larger changes (positive) in precipitation,* though we do not discount the influence of surface and meteorological conditions that may also lead to similar changes. This work clearly suggests that heating processes within the cloud important and that specifies precipitation and depth of convective system. I think only use of ref or Na at the cloud base can be very simplistic way to consider cloud top height changes. Therefor, authors should show relationships among ref, Nd, and LWC within the cloud system in their work.

Introduction section; some observations and part of method are given here but these need to move into proper sections. Text flow is not clear and very vogue in intro and method sections. Method section needs to be improved and explained properly as suggested above. I see method section is very poorly designed, providing these equations dont help for the comprehensive analysis of observations. See below for details.

**Major/minor issues:**

**Abstract**

LN31; Is this for only pollution aerosols or all "aerosols"?

Define ACR….CHUVA campaign.

LN 34 Provide method/assumptions first before providing results

Dr is confusing for defining height? Why not use Z, or h etc for height.

Also Na is used commonly for aerosol # concentration, please use Nd for droplets and Na for aerosols. If adiabatic droplets used for Na then show it as Nda.

Rain initiation is not only microphysics issue but dynamic of the system and environment. Changing of cloud height is related heat released in each layer because of mass change. Need to discuss this later.

Provide what assumptions are used first.

Measured ref was close to its adiabatic value? I understand that is possible in convective clouds but if mixing and precip occurs, it will cause to non-adiabatic conditions. In fact, size>50 micron can result in precip. Please clarify this in manuscript.

Biomass burning aerosols assuming darker and resulting heating may not be good nuclei in the convective systems anyway. Then, they do not play a role in rain initiation. Needs to be clarified with refs. Not all aerosols are good CCN and variability can be large, see above .

**1. introduction**

LN 55;  change to "cloud droplets form when humid air becomes supersaturated wrt water".

LN60, Rosenfeld and Woodley 2000, provide earlier reference for this, it is well known over 100 years.

LN65;  Provide a reference  as Gultepe and Isaac (AMS J of Climate, 1999) for Nd variability and Nd-Na relationships.  This paper clearly shows Nd-Na variability in the clouds representing various cloud types, clean and dirty environments. But most important it provides variability in Na versus Nd. Gultepe et al 1996 (AMS JClimate) provides Nd-Na relationships and k coefficient from various projects between rv and reff. These are earlier than references you provided, and earlier refs should be cited. k in this paper is given as 0.90 But it becomes reff=1.11rv that is ~10% larger than MVR.  Clearly, variability in Na and its relation to Nd are critical for many applications and these relationships are not unique.

LN70; are formed or formed?

LN 115; top parag; I feel this belongs to Observations section.

LN 120: This paragraps describes the goal of the paper, needs to be provided in the end of intro.

LN125-130; again this should be in the end of intro section. This is the goal of the paper.

LN 130, this parag belongs to observations/data section, not here.

LN135-145; this parag belongs to method section, not here.

LN 147; The DWC is defined………..as the mass of drops integrated over diameter range of 75-250 micron. This is not right, as you said it is drizzle water content (DWC), and therefore it is the water content of drizzle that is obtained by integration of (Nd xMd) over the spectral range representing CDP size range. Needs to be corrected.

What happens particles between 50 and 75 micron size range?

Page4; section 1.1 The scientific motivation for this study; take out "for this study".

I think no need for section 1.1 title, you don't have others anyway.

LN160; Say "The aircraft based in-situ measurements collected within convective clouds over the Amazon region of Brazil  were used to study precipitation initiation related to aerosol properties".

LN160-175 is too long; you said these before, and needs to be shortened.

**Section 2. Instruments; LN176**

This section should be "2. Project design and observations", briefly describe project location and flight patterns. Then go to observations, including instruments.

LN 178; 2.1 cloud particle measurements; 2.2.1 and 2.2.2 are ok, how about 2.4??? isn't that aerosols are part of cloud?

Section 2.3; please provide synoptic info on CCS, how and where they happened exactly.

**3.Method**

a) The relationship between ref and the probability of drizzle is defined???? This is not clear. You mean ref and drizzle LWC is described/analyzed? In what content?

b) That is not clear and based on what?

Goals of c) and d); these needs to be better organized/provided

You provided four steps here, but only 3.1 and 3.2 are given, they are not consistent with 4 steps given above. Need a road map here, confusing the way done.

Page 6; LN245;  CWC using Eq. 2 is not clear? How probes data are used? From what channels?

Page 9; LN 375; Analysis of ref and D……..; analysis should be in the method section not here. You should refer the analysis for the results given here.

4.2.3 Discussion should be a section on its own, not a subsection.

I am just passing results sections because of issues with other parts.

Other points from Gultepe et al (1996) suggest that ref max is about 12 micron which is obtained from fssp probe similar to cdp. If CIP is used, this increases to higher values of ref (>12 micron) for precip particles, as in your paper. Then, critical point for cloud depth is related to values of updrafts.  Droplet spectra changes although ref can be same, then this can affect D-ref or Nd relationships. This means that depth of the cloud is related to updrafts and latent heat released due to cooling/heating processes, not only cloud base max aerosols concentration or effective size.  In fact your data do not represent characteristics of convective clouds properly because flights are not from core of Cbs and dt time periods used for plots are different for collected data.

**5. Conclusions**

I suggest that you provide your findings in itemized way, what are they? As you said clouds are not adiabatic identities; agree that they are usually non-adiabatic in nature. Means these linear relationships suggested by you will not hold. Specifically convective clouds are usually non-adiabatic systems because of strong wind/turbulence effects, heating, and mixing and precip. Unfortunately, these are not discussed in the paper.

Figures:

Figure 1a; conceptually this figure is wrong, droplets do not grow continuously to the top of a convective cloud. Also, based on this, your measurements are not in the cloud but in the edges. This suggests your results may not represent adiabatic conditions. You need to show LWCa and LWCm profiles for an entire flight.

Figure using a satellite/synoptic map/image will help to visualize the system. A figure is needed.

Fig. 3; like to see particle Nd, aircraft heading, and wind time series for this time period. To me 2000 cm-3 is an artifact or just aerosols.

Figure 4; a plot for 25 second is statistically meaningless, please provide longer data set.

Figure 5a-c; why don't you say CIP images rather than CCP-CIP images.

Figure 7a; reff increases with height but LWC decreases, usually LWC increases with reff, explain (see Gultepe et al 1996).

Fig. 8; Nd~100 cm-3 but ref increases with height. No aerosol effects, needs clear explanation.

Fig. 10; Around 1 micron, no diff among the cases; Nd more than Na at larger sizes, how that happens? Sensor issues?

Fig. 11a; need to show same way for LWC, Nd, and Ref, otherwise no idea what is going on. Also, dt=247 second for flying from 20C to -20C, how good is the data?

Fig. 15; this plot should be taken out, fit is useless based on # of points and issues stated above.

Fig. 16; how those lines with color are obtained? What equation is used?

Fig. 17: caption; "As continental are the convective clouds?????" Not right sentence. This conceptual figure is also having issues similar to Fig. 1. Is this correct that more aerosols results in higher cloud tops? I don't think so. If stable environment exist, doesn't matter what you have as Na, especially if they are small. Other point, you have rain drops but do not fall????? Certainly they fall.

**Supplementary material;**

Fig. 6a; These figs need to show mean values over the x bins. Say 100 cm-3. Like to see time series of data for at lears 1 flight, with aircraft headings.

Profiles/time series should have LWC, Nd, and Reff, and mean/sd need to be shown for at least one flight. Providing 1 sec max values do not make sense.

---

## Editor Decision (ED1)

Editors Comments on the article:

I have carefully examined the three referees comments and provided my own to the authors. Two of the reviewers have rated the articles' significance and scientific quality poor. Dr. Baumgardner rated the latter two as fair. Issues relate to not adequately addressing the effects of mixing and instrument-related issues. On the basis of the reviews, I have little choice but to recommend rejection of the article in its present form. On the positive side, Referee 2 has a potential approach that could lead to an article that would give it a Good or Excellent rating. The suggestion is to use a theoretical model to examine what the effects of inhomogeneous mixing might be. I'd recommend that you consider that.

Sincerely,
Andy Heymsfield

Referee 2 comments:

The authors have not satisfactorily addressed several of the very reasonable and serious points made by the referees and editor. The primary concerns have to do with the effects of mixing and instrument-related issues. Referee 2 notes that the revised manuscript has changed very little, and that many of the problems listed in the first review remain in this manuscript.

**1) Scientific Significance**  Poor

**2) Scientific Quality**  Poor

**3) Presentation Quality**  Poor

- The implicit basis of the analysis presented in the Gamma phase space is that one is dealing with a Lagrangian case. But, inevitably, with any sort of microphysical measurements different samples of particle populations are being sampled.
- What are the effects of mixing on the PSDs?
- The reviewer suggests that you could use simple theoretical/modeling calculations to also help you assess how the DSD characteristics are being affect by homogeneous/inhomogeneous mixing.

**Referee 3 Comments (Baumgardner)**

The referee states that he content has been somewhat improved from the original version, but he notes that it is still not ready to be released for publication until most

of his previous comments are better addressed than they were in the authors' original response.

-

**1) Scientific Significance**  Good

| **2) Scientific Quality** | Fair | • The referee suggested some wording changes to the underlying thesis of the study. |
|---|---|---|
| **3) Presentation Quality** | Fair | • The referee notes and I agree that the results of this study, as currently presented, suggest that the height of precipitation initiation in any convective cloud can be represented by a |

single, integer multiplier of the cloud base droplet concentration. This is in line with interpretations by Rosenfeld using satellite data.
- The referee notes that there are numerous references to the measurements from the aircraft probes that need to be modified or added.

**Referee 4 Comments**

The referee states that "the authors have not satisfactorily addressed several of the very reasonable and serious points made by the referees and editor. As far as I can tell, the revised manuscript has changed very little. Many of the problems listed in the first review remain in this manuscript.

**1) Scientific Significance**  Poor

**2) Scientific Quality**     Poor

**3) Presentation Quality**   Poor

- The reviewer notes that the authors have offered very little discussion of the limitations of the observations and processes which can break or increase the error in the simple relationship. This was raised by Referee 2 as well. That referee did offer a suggestion for how to consider this point in more depth.
- The referee notes that there are no figures showing the dynamical structure of the clouds.

- An issue was raised about inhomogeneous mixing occurring within the clouds and the effect on the effective radius.

---

## Author Response (AR2)

**Responses from authors to reviewers of the manuscript:** "Further evidence for CCN aerosol concentrations determining the height of warm rain and ice initiation in convective clouds over the Amazon basin."

**Authors:** *Ramon Campos Braga[1], Daniel Rosenfeld[2], Ralf Weigel[3], Tina Jurkat[4], Meinrat O. Andreae[5,9], Manfred Wendisch[6], Ulrich Pöschl[5], Christiane Voigt[3,4], Christoph Mahnke[3,8], Stephan Borrmann[3,8], Rachel I. Albrecht[7], Sergej Molleker[8], Daniel A. Vila[1], Luiz A. T. Machado[1], and Lucas Grulich[10]*

**General comments**

We appreciate and thank the Editor for his considerable effort and care in the complex review of this paper.

The authors thank the referees for the general comments and advices. Furthermore, the advices of the referees are highly appreciated as well as the very valuable and constructive suggestions to increase the quality of the manuscript. We tried to address the points requested by the reviewers to the paper be considered for publication. Overall, we have improved the focus of the paper highlighting our objectives and the novelty of our study.

In the new version of the manuscript we have removed two Figures (3-4) as requested by the reviewers and an additional schematic Figure which summarizes our findings was added in the discussion section.

In the following text the reviewers' comments are highlighted in blue color and the authors responses are provided below.

================================================================

**Referee number 1**

Report 1:

The authors have not satisfactorily addressed several of the very reasonable and serious points made by the referees and editor. As far as I can tell, the revised manuscript has changed very little. Many of the problems listed in the first review remain in this manuscript.

These include:
1. The main result of the paper is to produce a relationship between the height of the first measurement of drizzle drops and the estimated number of cloud drops at cloud base (Fig 17). The problem with the approach and the resulting relationship is the lack of attention to detail. The authors have offered very little discussion of the limitations of the observations and processes which can break or increase the error in the simple relationship. Ultra-giant aerosols, enhancement of collision and coalescence due to turbulence and increased supersaturation due to entrainment and mixing will all potentially weaken the relationship.

A: All these factors were not measurable. In the case of giant CCN we found no evidence for large concentrations (as shown at Figure 10), but the observation of rain initiation for $r_e$ ~12 um (see Figure 9a) is an indication of the presence of GCCN (this was also observed at previous studies e.g. Freud and Rosenfeld, 2012 and Konwar et al., 2012).

The observed results mean that all the processes that are the subject of the reviewer's concern are secondary in their importance to the process that determines the documented relationships, as described in the manuscript. We add the text below in the manuscript.

"The linear relationship between $N_a$ and $D_{13}$ indicates a regression slope of about 5 m $(cm^{-3})^{-1}$ between $D_{13}$ and the calculated $N_a$ for the Amazon during the dry-to-wet season. This value is slightly larger than the values observed by

Freud and Rosenfeld (2012) for other locations around the globe (e.g., India and Israel). These clear linear relationships found between $N_a$ and $D_r$ ($\sim D_{13}$) for different regions highlight the efficiency of the adiabatic parcel model to estimate the height of rain initiation within convective clouds. Additional cloud processes associated such as GCCN, cloud and mixing with ambient air and other processes that are not accounted for in this study produce deviations which are already included within the observed uncertainty of the linear relationship $N_a$-$D_r$.."

These very concerns of the reviewer render the scientific significance the findings of this paper as quite high, because the reviewer was concerned that these other processes would mask the relationships between $N_a$ and $D_r$, but in fact they do not. This realization worth publishing.

R1: 2. There are no figures in the paper to show that the measurements were made in the main updraft near cloud top to avoid observing raindrops formed at higher levels and falling in the downdrafts around the cloud edges. What is the evidence that the cloud top was not higher at an earlier time? In fact there are no figures showing the dynamical structure of the clouds.

A: Figure 2 illustrates the flight pattern near cloud tops. It is documented in the videos of the nose camera. Again, if there was a risk of rain falling from above, which is minimized as we described in the manuscript, it would decreased the strength of observed relationships. To the extent that the data was contaminated by rain from above, it only demonstrates that this did not happen to the extent of masking the relationships, which is independently hypothesized and observed previously, as described in section 3.1. See below the relevant texts from the manuscript.

"…It is assumed that rain (or ice) formation starts when calculated DWC exceeds 0.01 g m$^{-3}$ (Freud and Rosenfeld, 2012). For rain initiation in liquid phase the DWC threshold is ~10% greater due to the overestimation of DWC during CIP measurements in warm clouds (as stated at Section 2.2.1). The small terminal fall speed of the drizzle drops ($\leq 1$ m s$^{-1}$) allows to focus on in-situ rain (or ice) initiation while minimizing the amount of DSDs affected by rain drops fallen from above into the region of measurements. In addition, cloud passes with rain were eliminated when cloud tops were visibly much higher than the penetration level (> ~1000 m), based on the videos recorded by the HALO's cockpit forward-looking camera. However, cloud tops higher than few hundred meters above the penetration level occurred only rarely."

R1: 3. There is the persistent argument that the almost constant value of the effective radius is evidence of inhomogeneous mixing...

A: The reviewer misses the point that inhomogeneous mixing leads to adiabatic effective radius, and not a fixed one. The fact that the measured $r_e$ is close to the estimated adiabatic $r_e$ highlights the observations that the behavior of cloud mixing with air is nearly inhomogeneous, and therefore the effective radius behaves nearly as in adiabatic cloud. This is explained better now at section 3.2.

R1: If inhomogeneous mixing had occurred, there would likely be an increase in supersaturation and hence in the size of the largest drops as well as activation of CCN from the drops evaporated due to the mixing. Importantly, the raindrops would most likely form at a lower level than in a cloud region containing the number of cloud drops activated at cloud base and moved in an adiabatic parcel. Similar arguments can be made for turbulent enhancement.

A: The main process that we highlight as responsible for the relationships between Na and height for rain initiation (Dr) is evidently the dominant one. Additional processes such as turbulence etc. produce deviation which are already included in the uncertainty of Na and Dr relationship. We add the text below to the manuscript.

"The linear relationship between $N_a$ and $D_{13}$ indicates a regression slope of about 5 m $(cm^{-3})^{-1}$ between $D_{13}$ and the calculated $N_a$ for the Amazon during the dry-to-wet season. This value is slightly larger than the values observed by Freud and Rosenfeld (2012) for other locations around the globe (e.g., India and Israel). These clear linear relationships found between $N_a$ and $D_r$ ($\sim D_{13}$) for different regions highlight the efficiency of the adiabatic parcel model to estimate the height of rain initiation within convective clouds. Additional cloud processes associated such as GCCN, cloud and mixing with ambient air and other processes that are not accounted for in this study produce deviations which are already included within the observed uncertainty of the linear relationship $N_a$-$D_r$.."

R1: 4. As the authors admit themselves, they have insufficient information to examine ice initiation in these clouds.

A: We show where first ice is found in CIP images. The initiation of ice can be visually ascribed for sizes greater than ~ 0.25 mm and it does not mean that frozen smaller particles cannot be present. This is commented at the manuscript as shown below:

"Table 3 shows the cloud depth above cloud base at which warm rain initiation ($D_r$) occurs (i.e., DWC > 0.01 g m$^{-3}$) for all flights as a function of estimated $N_a$. The $D_r$ is taken as the cloud depth for ice initiation ($D_i$) if ice particles are evident in the CCP-CIP images. Here, the $D_i$ is visually ascribed for sizes greater than ~ 0.25 mm and it does not mean that frozen smaller particles cannot be present."

===================================================================

**Referee number 2**

Report 2:

R2: This is a second round of review process for me. I don't want to review this paper again if further corrections are needed. I feel that this paper needs to be better organized and data analysis should be done properly. I still see that my main concerns are not corrected properly, and text is not improved. Authors insist to keep text same as before after making some minor corrections. I will not go over again my points here but emphasized some points below:

Mainly figures do not represent what is said in the text.

Data analysis does not reflect proper averaging times

A: Averaging times were dictated by the length of cloud passes.

R2: Comparisons with adiabatic calculations were not discussed properly.

A: More explanations is added now at section 3.2 to clarify what is done.

R2: **Title says: Further evidence for CCN aerosol concentrations determining the height of warm rain and ice initiation in convective clouds over the Amazon basin.**

I don't believe that only CCN determines height of warm rain and ice initiation processes…. This is a misleading title, in fact IN or both CCN and IN at high levels plays an important role and never mentioned.

A: The main point of the manuscript is that Na affects Dr. Na is determined mainly by CCN and cloud base updraft. This is shown at Braga et al. (2017).

CCN do affect ice initiation much more strongly than IN in by determining the extent of the Hallett Mossop ice multiplication processes. This is stated in the manuscript in the Introduction, see the text below:

"Ice multiplication is an important mechanism that masks the primary ice nucleation activity when cloud droplets are sufficiently large to promote also warm rain by coalescence, at the temperatures of -3 to -8 °C (Hallet and Mossop, 1974)."

R2: How can adiabatic calculations be made for ref for comparisons or LWC, and used for comparisons if non-adiabatic terms are clearly dominant in a convective process at the certain phases of the storm??? At least 10 times adiabatic values are used for comparisons, in fact, convective clouds may deviate significantly from an adiabatic assumption.

A: The reviewer ignored the large body of literature, described and referenced in the Introduction of the manuscript, showing that the predominant inhomogeneous mixing leads to nearly adiabatic effective radius while LWC can still vary greatly below adiabatic values in the very same cloud volume.

R2: Figures asked to be generated or removed are not included or deleted.

A: We have excluded in the new version Figures 3 and 4 as it was requested.

R2: Fig. 1b is conceptually wrong and not true, if this is the case, what argument you have in the text. I feel this is pointed out previously.

A: But Fig. 1 describes, as well as an illustration can capture, the way that we did the flights and the clouds penetrations. This statement of the reviewer accentuates the notion that he has misconceptions with respect to what the paper is all about.

R2: Fits are provided may not represent data points distribution properly.

A: The fits were generated objectively and show all the data points encountered within the cloud passes during the vertical profiling parts of the flights.

R2: Icing detector and RH plots as indicated before are needed, without them you cant really say particles as droplets or ice crystals when particles are not falling above.

A: Why should RH be relevant within a supercooled water cloud?

The reviewer appears to have ignored the following text in the manuscript:

"The hydrometeor type is identified visually by their shapes. The phase of the smaller CCP-CIP particles cannot be distinguished. Therefore, the precipitation is considered as mixed phase when ice particles are identified, and the combined DWC and RWC are redefined as mixed phase water content (MPWC)."

R2: Images do not show statistically significant data points for droplets or rains. How do you know they are liquid?

A: It is sufficient to identify visually a single ice particle for determining ice initiation. This was done for particles with sizes > ~0.25 mm (precipitating particles).

R2: Some text related to CDP and/or CIP probe are not correct.
How collisions and coalescence processes affect cloud macro and micro-properties, how about turbulence?

A: We have no information to address this question. But we can state that most variability in Dr is explained by Na, leaving less room for the impacts of these additional factors. The variability in the relationships between Na and Dr is quantified now and shown at Figure 15.

R2: Conclusions are not provided properly and explained, this was mentioned in my previous review.

I like to see not only Na characteristics affecting cloud properties but other parameters such as updrafts and mixing as well radiative processes. Conclusions provided are like for a conference paper.

A: These measurements are not available for the full cloud columns. The fact that Na explains much of the variability in Dr means that there is little room left for all these other processes. The reviewer does not accept this as a possibility. These results found and shown in the manuscript put the relatively small magnitude of these processes in the correct perspective, at least for the Amazon and other regions as Israel and India (shown at Freud and Rosenfeld, 2012). We add the text below in the manuscript.

"The linear relationship between $N_a$ and $D_{13}$ indicates a regression slope of about 5 m $(cm^{-3})^{-1}$ between $D_{13}$ and the calculated $N_a$ for the Amazon during the dry-to-wet season. This value is slightly larger than the values observed by Freud and Rosenfeld (2012) for other locations around the globe (e.g., India and Israel). These clear linear relationships found between $N_a$ and $D_r$ ($\sim D_{13}$) for different regions highlight the efficiency of the adiabatic parcel model to estimate the height of rain initiation within convective clouds. Additional cloud processes associated such as GCCN, cloud and mixing with ambient air and other processes that are not accounted for in this study produce deviations which are already included within the observed uncertainty of the linear relationship $N_a$-$D_r$.."

==================================================================

**Referee number 3**

Report 3:

R3: This manuscript is a revised version of a previous submission. Although the content has been somewhat improved from the original version, it is still not ready to be released for publication until most of my previous comments are better addressed than they were in the authors' original response. In addition, upon further evaluation of the paper, I have found other sections that need further clarification or modification.

To put my concerns about this paper in context, I will use this sentence in the introduction to highlight what I think is a potentially hazardous approach by modelers: ***"These parameterizations need to represent in simplified form the complex chain of events that occur in clouds."*** I understand the motivation for parameterizations of cloud microphysical processes in global scale models, and depending on the application of the parameterizations, simplifications can be implemented without seriously misrepresenting the cloud microphysics. In other cases, however, oversimplifications will produce results that are erroneous and misleading. Perhaps it was the first author's wording, and doesn't reflect the thinking of the other authors. If that is the case, then just remove this sentence because I doubt that any serious modeler would agree that the complex chain of events that occur in clouds can ever be represented by simplified parameterization.

A: The sentence is removed. However, much of the point of this paper is that the relationships between cloud base updraft, CCN, Na, Re and Dr obey rather simple rules that are suitable for parameterization. All three reviewers have problems in accepting that other processes such as mixing, turbulence, additional drop activation and even updrafts well above cloud base play a secondary role. This is exactly what makes this paper so important scientifically! If the claim of the paper is really true, it is important news, at least for the reviewers, and more likely to the readers. This does not mean that these other processes are not important. But at least in the clouds that we sampled

their variability from cloud to cloud apparently was not sufficiently large to dominate the relationships between CCN, Na, Re and Dr. We add the text below in the manuscript.

"The linear relationship between $N_a$ and $D_{13}$ indicates a regression slope of about 5 m $(cm^{-3})^{-1}$ between $D_{13}$ and the calculated $N_a$ for the Amazon during the dry-to-wet season. This value is slightly larger than the values observed by Freud and Rosenfeld (2012) for other locations around the globe (e.g., India and Israel). These clear linear relationships found between $N_a$ and $D_r$ ($\sim D_{13}$) for different regions highlight the efficiency of the adiabatic parcel model to estimate the height of rain initiation within convective clouds. Additional cloud processes associated such as GCCN, cloud and mixing with ambient air and other processes that are not accounted for in this study produce deviations which are already included within the observed uncertainty of the linear relationship $N_a$-$D_r$.."

R3: What further worries me is that the results of this study, as currently presented, suggest that the height of precipitation initiation in any convective cloud can be represented by a single, integer multiplier of the cloud base droplet concentration. In the paper's current form, there are a limited number of caveats to this statement given in the abstract and the summary and yet the caveats are significant and need to be listed just as boldly as the simplified relationship between precipitation initiation height and cloud base droplet concentration.

A: The message that we get from the reviewer is that we should explicitly point out all the competing processes that could affect the Na-Dr relationships and discuss how they would potentially affect these relationships. A crucial part of the discussion would be the fact that at the bottom line most variability in Dr was explained by Nd (we quantify it better in the new version), thus leaving less room for these other processes. The previous response also addresses this point.

In addition, we have highlighted in the new version of the manuscript the uncertainty regarding the Na-Dr relationships. The uncertainty of the linear relationship is mentioned at abstract and summary and is in mean terms of about 21 %, but it does not mask the linear relationship which was found also at different regions around the globe (e.g. Israel and India).

R3: The motivation for forcing such a relationship is not obvious in the current paper but in reading the paper by Freud and Rosenfeld (2012), it seems that the driving force behind these parameterizations is to extract more information from satellite data about cloud properties. This is a worthy objective but not if accomplished at the cost of diminished scientific robustness.

In my comments below, I will further highlight and provide details of where this paper will need to be improved before I will accept it for publication. The bottom line is that I can't and will not allow the publication of a parameterization that can so easily be misused until it is properly justified.

**Editorial comments**

In my opinion it is my responsibility as the reviewer, a responsibility that I take very seriously, to help improve the paper I am reviewing by 1) identifying technical and factual errors or omissions,

2) requesting clarification when needed and 3) suggesting modifications that help to solidify the hypothesis put forward. In my follow-up review of the revised manuscript, I annotate my remarks with RR1, RR2 or RR3 so that the authors understand the motivation for my comments and suggestions in relationship to how I view my reviewer responsibility (RR).

The authors have sufficiently addressed my concern about the coincidence error losses but all my other comments lack responses that adequately address my comments or concerns. Giving the authors the benefit of the doubt, I will assume the responsibility for not having stated clearly enough the nature of my concerns. Hence I will repeat them here, but with enough detail that there should not be any confusion as to the nature of my comment and how I expect it to be addressed. I will also be more clear about the seriousness of my comments, i.e. those for which I expect concrete changes to the manuscript and those where I will accept lesser modifications as long as my comments receive a reasonable and scientifically defensible response. My comments are not necessarily in the same order as they were presented in the first review.

**Specific Comments, Questions and Suggestions**

1) Error propagation

R3: In Braga et al. (2016) and relatively comprehensive uncertainty analysis is conducted of the number concentrations and sizing by the light scattering and imaging probes; however, this analysis is not taken into account in the current paper to estimate the expected uncertainty in determining Na, Re, Rea, CLWC, Mv, etc. This is a major omission (RR1) that must be rectified.

In their original response to my request to propagate the measurement uncertainties into the derivation of the Dr vs Na relationship, they state:

*"A: The uncertainty of Na calculation with CDP (14 %) is now included in the linear relationship. The linear relationship including Na uncertainty is D_r=(5±0.7)·Na" .*

A: We have now performed the error propagation where possible. This is shown at Appendix A and Figure 15 in the new manuscript.

This is inadequate since it does not take into account the very large variations in the CLWCa vs Mv relationship that was shown in the Braga et al (2016) paper (Fig. 14a, redrawn below for a single flight), nor does it explain that an additional 30% unexplained correction has been applied to the Na as also explained by Braga et al. (2016), i.e. "However, this methodology does not account for cloud mixing losses from droplet evaporation, and the Na estimates commonly overestimate the expected Nd by 30% (Freud et al., 2011). Therefore, in calculating Na we applied this 30% correction.".

A: The reason that $r_e$ can remain near adiabatic while CLWC can vary greatly in the same cloud volume is explained in Section 3.2. We have added the explanation for the 30% correction in Na (Freud et al., 2011) in section 3.2.

R3: The Mv will have at least an uncertainty of 50% since it is derived from rv (I think, although nowhere is Mv ever explained how it is derived). In Fig. 14a, the best fit doesn't even go through most of the points at low or high LWCa so how can a concentration be derived better than to the nearest 100 cm-3, much less to the nearest

10th! The supplementary material needs to show the same type of figure as Fig. 14 for all flights so that we can actually see how much deviation there is.

A: Yes, Mv is the mass of a water sphere having the radius Rv. This is explained in section 3.2 as follow:

"The uncertainties calculations of cloud properties estimated from cloud probes were described in Braga et al. (2017). The uncertainties of $r_e$, $r_v$, $r_{ea}$, $r_{va}$ are about 10%, while for CWC and $M_v$ the uncertainties are about 30%."

R3: The uncertainty in Na then propagates into the rea and the derived re will have an uncertainty of more than 20% if the uncertainties in size are properly propagated.

A: We have recalculated the uncertainty of the retrieved Na. The uncertainty of $N_a$ is ~21% in mean terms. The calculation is shown at Appendix A and the values at Table 3.

R3: Finally, there is a very large uncertainty in the Dr, not only in its derivation of the DWC but also in the actual cloud depth where it is measured. There is an uncertainty of at least a maximum of Dr-Dr-1 where Dr-1 is the cloud depth at the previous cloud penetration where the r13 threshold was not exceeded. This is because the re threshold could have been exceeded in the rising air mass at any point between the current and previous cloud penetration. This is not addressed at all and is a serious omission.

A: We marked the interval of Dr-Dr-1 as the uncertainty for Dr at Table 3. The uncertainty in $r_e$ does not change Dr and Dr-1.

R3: In summary, the ±0.7 value is unsupported and requires a much more robust derivation than is currently given.

A: We have recalculated the Na-Dr relationship with the error propagation. The results is shown at Figure 15 and Table 3 for each flight. The uncertainty of $N_a$ is ~21% in mean terms and the linear relationship of Na-Dr is $D_r = (5\pm1.06) \cdot N_a$.

In my previous review I stated: "In the images from the CIP, there are many out of focus droplets (donuts). The Korolev (2007) correction has to be done, otherwise the derived water content will be an overestimate and the height of precipitation might be incorrect."

To which the authors replied: *"A: For the data processing of the CCP measurements, ice was assumed as the predominant particle phase in the mixed-state cloud conditions that were mainly given throughout the ACRIDICON CHUVA campaign. The ice assumption causes all images of droplets and ice particles to be treated and considered as particles (apart from shattering-induced particles) but the Poisson spot correction is then excluded. The Korolev correction is defined for liquid drops only and the SODA image processing disables this correction process once the ice-phase is selected. The assumption of ice density instead of water density implies a slight overestimation (_10 %) of the calculated rain water content for particles greater than 75 µm. This will be highlighted in the manuscript."*

I understand the issue of mixed phase, however, there are three flights, AC09, AC18 and AC19, that have a Dr that is obviously derived from all water images, so clearly the out-of-focus correction can and should be done. In addition, 2.2.1, it is stated "In this study, we deduced the existence of ice from the occurrence of visually non-spherical shapes of the shadows." Hence, water droplets can be detected in even mixed phase conditions so that these spherical images can be corrected to derive the true DWC. To not do so will bias the derived Dr and the subsequent rain initiation level. (RR1).

A: The donuts in the CIP images cannot be removed. But since they are particles that are out of focus this does not matter, as we get the quantitative information from the particles that are in focus. Per the reviewer's request, we recalculated the DWC with CCP-CIP assuming water as particle density for flights AC09, AC18 and AC19. The calculated Dr did not changed, but the DWC is smaller by about ~10-15% when assuming water density instead of ice density. This is presented in section 3.1, as shown in the new text:

"Table 3 shows the cloud depth above cloud base at which warm rain initiation ($D_r$) occurs (i.e., DWC > 0.01 g m$^{-3}$) for all flights as a function of estimated $N_a$. The $D_r$ is taken as the cloud depth for ice initiation ($D_i$) if ice particles are evident in the CCP-CIP images. Here, the $D_i$ is visually ascribed for sizes greater than ~ 0.25 mm and it does not mean that frozen smaller particles cannot be present. The assumption of water or ice density as the predominant particle phase on DWC calculation based on CCP-CIP probe did not impact $D_r$ and $D_i$ measured because the DWC threshold (i.e., DWC > 0.01 g m$^{-3}$) for warm rain or ice initiation was achieved at the same cloud depth for both particles densities. Additional details about the cloud profiling characteristics for each flight as the number of altitude levels sampled (NLS), highest cloud depth without raindrop ($D_{r-1}$) or ice particles ($D_{i-1}$) etc. are also available in Table 3. Furthermore, Appendix A discusses the uncertainty calculations of the estimated parameters of cloud properties."

R3: 3. Comparisons of CDP vs CAS-POL and Re vs Rv (RR2 and RR3))
I will reiterate my suggestion that the CDP vs CAS-POL comparisons in Figs. 3 and 4 be removed. First of all, Braga et al. (2016) have more than shown how well these two instruments compare and secondly, there is no reason given in the text of the purpose of comparing Re vs Rv. Freud and Rosenfeld discuss this but it has no relevance in the current paper. These figures and the associated descriptive text should be removed.

A: We have removed Figures 3 and 4.

R3: 4. Further clarification of the derivation of Na (RR2)
After reading Freud and Rosenfeld (2012) I was able to decipher the rather cryptic discussion of how Na is derived; however, the response to my original request for clarification raises a number of additional questions and related concerns. Here is the original response to my request for clarification:
*"A: The reviewer wrote: "I understand that LWC = Nd*Mv". This is not quite so. The right expression is LWCa = Nda*Mva, where all are the adiabatic values. The whole idea of the methodology is that the actual re is similar to rea - the adiabatic effective radius, due to the nearly inhomogeneous nature of the mixing. The mixing does decorrelate LWC strongly from LWCa, while keeping re well correlated with rea. The methodology which use LWC vs. Mv relationship with height to estimate Na is well tested and validated at Freud and Rosenfeld (2011). The Na estimate is also explained and tested at Braga et al. (2016). Indeed there*

*are uncertainties related to Na estimated mostly related when secondary nucleation takes place. The model does not predict that Nd increases with height, but decrease due to coalescence and inhomogeneous cloud mixing. The results suggest the occurrence of secondary activation with different strengths during flights AC08, AC12, AC13 and AC20 (see figures attached). Large updrafts were measured above cloud base during these flights which increase supersaturation inducing secondary activation.*

This response seems contrary to lines 203: "b) The *Na* at cloud base is estimated through the vertical profile of *re*." and line 245: "The *Na* for the convective clusters is estimated based on the slope between the calculated CWC and the mean volume droplet(*Mv*)". This seems to indicate that the Na is being derived from measurements not the adiabatic values.

A: Na at cloud base is indeed estimated through the vertical profile of Re, which is expected to be close to Rea. This vertical profile is measured in a cluster of adjacent clouds. Indeed, additional activation reduces the actual Re compared to Rea and thus induces a positive bias in the computed Na. We address it in the new version at section 3.2 as follow:

"The $N_a$ for the convective clusters is estimated based on the slope between the calculated adiabatic CWC (CWC$_a$) and the mean volume mass of the droplets ($M_v$), which is the mass of a water sphere having the radius $r_v$. Mv is calculated for 1-s DSD measurements of CAS-DPOL and CCP-CDP for non-precipitating cloud passes (Braga et al., 2017). The underlying assumption is that the measured $r_v$ is approximating the adiabatic $r_v$ ($r_{va}$) due to the nearly inhomogeneous mixing behavior of the clouds with the ambient air (Beals et al., 2015). Therefore, the measured $M_v$ approximates the adiabatic $M_v$ ($M_{va}$, where $M_{va} = CWC_a / N_a$)."

R3: This seems to indicate that the Na is being derived from measurements not the adiabatic values. In addition, Mv is never defined, although I was finally able to deduce that it is somehow being derived from Rv, but this is never made clear.
This response also raises the issue of how adiabatic LWC and Re are derived, i.e. one must know not only the cloud base temperature and pressure, but vertical profiles of temperature, pressure and mixing ratio, as well. What vertical profiles are being used? This needs to be described at the very beginning, as well as the uncertainties involved,

i.e. how much do these vertical profiles change over time, especially during the time period of the measurements from cloud base to cloud tops?

A: Since these are convective clouds, we assume simply a moist adiabatic lapse rate within the clouds.

R3: Then the comment regarding "Secondary nucleation" brought forward one of my other critiques regarding of the use of this term. Here is my previous comment:
"9) Secondary nucleation is a very poor term because in a classical parcel model in an updraft, new particle nucleation occurs above cloud base until there are no more cloud active CCN at the level of SS. The implication here is that new CCN are being entrained and that is why the Nd increases with altitude, but this is likely not the case. When running a parcel model with a prescribed updraft and CCN spectra, the supersaturation increases in altitude as the parcel rises adiabatically and cools. The CCN will activate depending on their SS spectra and the available water. This needs revising."
And this was the response:

*"A: The secondary CCN activation was observed mainly in cloud segments with updrafts that were much stronger than at cloud base. This supports the narrow definition of secondary activation as defined by the reviewer. However, we do not exclude the possibility of additional CCN being entrained and activated above cloud base."*

This response suggests that the authors did not understand my criticism, i.e. that I did not want them to use the term "Secondary Nucleation" or Secondary Activation" ANYWHERE in this manuscript for the reasons I stated. They are free to use the term "Additional activation/nucleation" or "Continuing activation/nucleation" but not "Secondary". If they wish to explain the possibility of entrainment of CCN above the cloud base, they are free to do so, as long as this would described as "additional" not "secondary"

A: We have corrected the term to additional CCN activation in the whole text.

R3:5. Precipitation particles coming from below the measurement altitude (RR1)

This concerns a comment I presented above and the response of the authors to a similar comment I had made in my previous review. That comment:

"7) Nothing is said about the uncertainty in the determination of level of precipitation wrt to vertical motions and where the precipitation actually initiated, i.e. it could have actually been below the level of measurement before being lofted upwards. This uncertainty can be estimated using the measured vertical motions."

The authors' response: *"A: Doing that would require information that we don't have about the rate of rain formation with height, and will constitute a circular argumentation. The scatter in Figure 17 is the best that we can do for illustrating the uncertainty."*

This response did not address my concern. In my comments above about the uncertainty in Dr, at the least, the maximum uncertainty can be estimated as the distance between the two measurement levels with and without the threshold being exceeded, e.g. if the Re and DWC had not been exceeded at the 3000 m level but is exceeded at the 4000 m level, given that it might have been exceeded at the 3001 m level that wasn't measured, the uncertainty in this case would be 1000 m. Hence, Table 3 has to have uncertainty bars on ALL of the quantities listed, including the Na.

A: We have included uncertainty range on all the quantities in Table 3 and Figure 15.

R3: 6. Information on time between flight legs through clouds (RR3)

My previous comment:

"8) Nothing is said about the time it takes to make the measurements at the various cloud levels and how these levels were selected. This will give some idea of the time during which the cloud is growing an how long it took to initiate precipitation."

The authors' response:

*"A: Since the measurements were not following individual growing cloud towers, these times would not advance such knowledge."*

First of all, the first part of the response seems to fly in the face of what is written on line 126: "The aircraft obtained a composite vertical profile by penetrating young and rising convective elements, typically some 100-300 m below their tops."

A: We don't see where is the problem. We assume that the time of making the vertical profile (less than an hour) is smaller than the time for changes in cloud base temperature, pressure and CCN.

R3: Secondly, I think that the amount of time from cloud base to each flight level is very germane to the question of how long it takes to initiate precipitation, given the discussion early on concerning the rate at which precipitation forms.

A: The time of cloud growth from cloud base to each level might be important, but the time of aircraft ascent is irrelevant to the cloud processes.

R3: Hence, in Table III I want to see a column that include the time after cloud base measurements that precipitation was identified.

A: We can't see how it would be relevant, because cloud elements that reach precipitating threshold usually already exist at the time that cloud base is measured. Please see Figure 1.

R3: In this same Table I want to see vertical velocity added to the cloud base conditions, another column with the number of levels that were sampled for this date and another column that shows the maximum vertical velocity.

A: We have added Wmax and the number of levels that were sampled to Table 3. All vertical velocities from cloud base to the last penetration height are provided at the supplementary material.

R3: This latter request is because it seems to me that from the Supplementary material, almost every flight had vertical velocities above cloud base that were larger than those

at cloud base. This would invalidate the assumptions that are needed to use Na as a predictor of Dr.

A: For additional CCN activation to occur well above cloud base the updraft should be MUCH higher than at cloud base, because of the existence of Smax there. Therefore, having vertical acceleration of the updraft does not mean always additional CCN activation.

R3: 7. Additional comments
In going through the revised manuscript, I had some remaining questions/comments:
Line 55: The authors use -36C as the threshold for homogeneous freezing but -38C is the value that is most commonly used, hence -36C should be changed to -37C, according to how they state this threshold.

A: We have changed it to -37C.

R3: Line 105: "The $Na$ is calculated from $Na= CWCa/Mva$, where $CWCa$ is the adiabatic cloud water content (CWCa) as calculated from cloud base pressure and temperature, and $Mva$ is the adiabatic mean volume droplet mass, as approximated from the actually measured mean volume droplet mass($Mv$) by the cloud probe DSDs obtained during the cloud profiling measurements."
This further confuses me as it implies a mixture of adiabatic and observed quantities, i.e. why is Mva being approximated from the measured mean volume droplet mass, and in addition, why is Mv never explicitly defined in an equation?

A: The underlying assumption, which will now be stated explicitly, is that the measured Rv is approximating Rva due to the nearly inhomogeneous mixing.  Therefore, the measured Mv approximates Mva.

The reviewer gathered correctly what is Mv and Mva (the mass of droplet having radius Rv and Rva, respectively.

A better explanation is provided at the new version at section 3.2. See below the new text in the new version.

"The $N_a$ for the convective clusters is estimated based on the slope between the calculated adiabatic CWC ($CWC_a$) and the mean volume mass of the droplets ($M_v$), which is the mass of a water sphere having the radius $r_v$. $Mv$ is calculated for 1-s DSD measurements of CAS-DPOL and CCP-CDP for non-precipitating cloud passes (Braga et al., 2017). The underlying assumption is that the measured $r_v$ is approximating the adiabatic $r_v$ ($r_{va}$) due to the nearly inhomogeneous mixing behavior of the clouds with the ambient air (Beals et al., 2015). Therefore, the measured $M_v$ approximates the adiabatic $M_v$ ($M_{va}$, where $M_{va} = CWC_a / N_a$)..."

R3: Line 143: "The DWC is defined as the mass of the drops integrated over the diameter range of 75–250 μm (Freud and Rosenfeld, 2012)." Does this mean that any mass beyond 250 μm is excluded?"

A: Yes.

R3: Line 275: "The precipitation probability is calculated by integrating the measured DSDs exceeding certain DWC thresholds.". This statement needs a great deal of clarification. First of all, why are these being called "precipitation probabilities" and secondly, are the DSDs being integrated up to the Re that produces the threshold DWC? Please be more explicit.

A: DWC is integrated between 75 and 250 μm, as the reviewer quoted above.

We have changed the sentence in the new version as shown below:

"The probability of precipitation is the fraction of 1-Hz in-cloud measurements which exceed certain DWC thresholds (i.e. for DWC > 0.01 g m$^{-3}$) as a function of $r_e$ value."

R3: Line 295: This statement about the relationship between Dc and Rea needs to be qualified with the caveat that it only holds under certain strict conditions, e.g. maximum updraft at cloud base, no continuing activation of CCN above cloud base, etc.

A: we addressed this point by the following added text:

"The $N_a$ calculation does not take into account the possibility of new nucleation above cloud base (Freud et al., 2011). Braga et al. (2017) have shown that the assumption of adiabatic growth of droplets via condensation from

cloud base to higher levels within cloud can lead to an overestimation by ~20-30% of the number of droplets at cloud base when calculating $N_a$ in cases with additional droplet nucleation above cloud base."

The differences between the estimated $r_{ea}$ and the measured $r_e$ as a function of $D_c$ is highlighted at section 4.2.3 as follow:

"This additional CCN activation leads to smaller $r_e$. For flights where additional CCN activation was significant, the differences between the estimated $r_{ea}$ and the $r_e$ measurements at same height are larger, because the adiabatic estimation does not consider the additional CCN activation of droplets above cloud base and thus overestimates the observed size."

In addition, the presence of GCCN also can produce differences in the height predicted by the adiabatic model to the rain initiation. In this case we justify in the manuscript (section 4.2.3) our findings for flight AC19 as follow:

"For the flight in cleanest conditions (AC19), the presence of larger aerosol particles (possibly GCCN from sea spray) below cloud base leads to a faster growth of cloud droplets via condensation with height, and consequently $r_e$ is smaller than 13 μm (see Figure 9a) for warm rain initiation. A similar decrease of $r_e$ for rain initiation over ocean was observed by Konwar et al. (2012)."

R3: Line 301: The maximum concentration was 2000 cm-3, but what Nds were used to compare with the Nas?

A: Nd was not directly compared to Na in this study. This was already done in Braga et al., 2017.

R3: Figure 8: Remove it. There is no relevant information here.

A: This image shows the convective nature of the clouds. We prefer to keep it.

Discussion
Given that the end objective of this study is to support the Freud and Rosenfeld (2012) parameterization so that it can be used by the satellite community to derive microphysical properties, the discussion needs to be much more clear about the

robustness of the Dr vs Na relationship with recommendations as to when is can be used and when it shouldn't be used.

The caveats that limit the use of the Dr vs Na relationship need to be bullets in the Summary so that future use of this parameterization is not used indiscriminately.

A: We have given now a better explanation about the scientific significance and motivation of this study. In the new version of the manuscript (section 1.1) we highlight the scientific motivation of our study as follow:

"**1.1. The scientific motivation for this study**

The *in situ* measurements of cloud properties were collected within convective clouds formed over the Amazon from cloud base up to cloud top above the glaciated level. These measurements provided a unique opportunity to evaluate previous theoretical knowledge about aerosol impacts on convective clouds characteristics over the Amazon. In this study the impact of $N_a$ (adiabatic cloud drop concentrations) in determining the initiation of rain and ice within convective clouds is evaluated. This is performed through the analysis between the calculated $N_a$, $D_r$ and $D_i$ for several different environmental conditions over the Amazon (cloud base updrafts, aerosol concentration, surface cover etc.). The relationship of $N_a$ and $D_r$ was previously analyzed for regions of Israel and India where a linear relationship was found ($D_r \approx 4 \cdot N_a$) [Freud and Rosenfeld, 2012]. For Amazon region a similar analysis is performed here also taking in account the impact of $N_a$ in $D_i$. This is the first study which analysis the impact of $N_a$ on $D_r$ and $D_i$ at Amazon region using *in situ* measurements of convective cloud properties. The results obtained from comparisons of $N_a$ estimates and the measured effective number of droplets nucleated at cloud base ($N_d$*), shown at Braga et al. (2017) for the same flights in the Amazon region, support the methodology of deriving $N_a$ based on the rate of $r_e$ growth with cloud depth, and under the assumption that the entrainment and mixing of air into convective clouds is extremely inhomogeneous. This is important because the characteristics of convective clouds based on $N_a$ values can be extended in space and time by their application to satellite-calculated $N_a$ (which is obtained with the same parameterization that has been recently developed from the satellite-retrieved vertical evolution of $r_e$ in convective clouds) [Rosenfeld et al., 2014b].

Regarding the robustness of Na-Dr relationship we better highlight the caveats that can limit the use of the equation provided at section 4.2.3 and at the introduction and summary. These limitations were already shown in the previous responses.

**References**

Braga, R. C., Rosenfeld, D., Weigel, R., Jurkat, T., Andreae, M. O., Wendisch, M., Pöhlker, M. L., Klimach, T., Pöschl, U., Pöhlker, C., Voigt, C., Mahnke, C., Borrmann, S., Albrecht, R. I., Molleker, S., Vila, D. A., Machado, L. A. T., and Artaxo, P.: Comparing parameterized versus measured microphysical properties of tropical convective cloud bases during the ACRIDICON–CHUVA campaign, Atmos. Chem. Phys., 17, 7365-7386, https://doi.org/10.5194/acp-17-7365-2017, 2017.

Freud, E., Rosenfeld, D. and Kulkarni, J. R.: Resolving both entrainment-mixing and number of activated CCN in deep convective clouds, Atmos. Chem. Phys., 11(24), 12887–12900, doi:10.5194/acp-11-12887-2011, 2011.

Freud, E. and Rosenfeld, D.: Linear relation between convective cloud drop number concentration and depth for rain initiation, J. Geophys. Res. Atmos., 117(2), 1–13, doi:10.1029/2011JD016457, 2012.

Konwar, M., Maheskumar, R. S., Kulkarni, J. R., Freud, E., Goswami, B. N. and Rosenfeld, D.: Aerosol control on depth of warm rain in convective clouds, J. Geophys. Res. Atmos., 117(13), 1–10, doi:10.1029/2012JD017585,2012, 2012.

---

## Author Response (AR3)

**Manuscript:** Further evidence for CCN aerosol concentrations determining the height of warm rain and ice initiation in convective clouds over the Amazon basin

**Manuscript code:** ACP-2017-1155

**Authors:** Ramon Campos Braga, Daniel Rosenfeld, Ralf Weigel, Tina Jurkat, Meinrat O. Andreae, Manfred Wendisch, Ulrich Pöschl, Christiane Voigt, Christoph Mahnke, Stephan Borrmann, Rachel I. Albrecht, Sergej Molleker, Daniel A. Vila, Luiz A. T. Machado, and Lucas Grulich

**Comments to the Editor and the reviewers**

We appreciate and thank the Editor for his considerable effort and care in the complex review of this paper.

We thank the Editor and the reviewers for the comments, which helped improving the manuscript. Below are our responses interleaved with the reviewers' comments.

###############################################################################

**Co-Editor Decision: Publish subject to minor revisions (review by editor)** (24 Oct 2017) by

Andrew Heymsfield

***Comments to the Author:***
Three reviews are now available for your manuscript. Reviewer 1 recommends acceptance of the article in its current form. Reviewer 2 suggests some modifications of the article and recommends acceptance with minor revision. Reviewer 3 has significant concerns. Given the 3 reviews, my recommendation is to address Reviewer 2's concerns and try the best you can to respond to Reviewer 3's comments. I'll be reviewing your resubmission--it won't take long for me to go through the manuscript and then a decision will be made.

**Referee #1**

Although the authors and I do not agree on a number of points in the paper, it is their prerogative to defend their point of view, which they did in their responses, and I will not further delay the publication of this study.

**Referee #2**

1. The authors should change a sentence in their added text: "These clear ... within convective clouds in this study.

" And: "Additional ... would produce deviations that are likely to be the cause of the observed scatter in the results."

A: Ok. We have made these changes.

2. The authors do not describe inhomogeneous mixing as it was originally defined. They should describe what they mean specifically in relation to the dilution or evaporation of drops.

A: The following text was added. We thank the reviewer for pointing out the need to add this much needed text.

"The inhomogeneous mixing occurs when evaporation rate of cloud droplets exceeds significantly the mixing rate of the cloud with ambient air. This causes the droplets that are at the boundary of the entrained air filament to evaporate completely and moisten that air until it is saturated. Further mixing of the saturated entrained air would not cause additional evaporation, but only decreasing of $N_d$ and LWC, while maintaining $r_e$ of the remaining droplets unaffected."

3. I remain unconvinced about including ice initiation in the paper.

A: We document what we measured. There was insufficient data for a separate paper, but it deserves to be published. The compromise solution is to include it in this paper.

**Referee #3**

Title: Further evidence for CCN aerosol concentration

By Braga et al

Date: Oct 21 2017

Decision: Rejected

I do not think that this paper can be improved over a short time period. Issues are very serious for data analysis and organizing of the paper, and simplification of a research area related to aerosol-precipitation and cloud depth.

*General comments*

This paper focuses on precip initiation related to CCN in convective clouds, and its impact on cloud depth. This is an interesting research area but manuscript needs major improvements related to introduction, observations, and method sections, as well as figures. My previous points are not considered properly in this new version. Provided figures are not representative of the real conditions because of analysis issues.

A: The figures represent exactly all the points within clouds during the conduct of vertical profiles within clouds. We don't know what else could be possibly done.

The $N_d$, LWC, and ref time series are not shown for at least for 1-flight that can be used to interpret analysis.

A: We don't see how it can be relevant to the questions addressed by the study as well as to issues raised by the reviewer. The values of $N_d$, CWC, $r_e$, $r_{ea}$ as a function of height or temperature are shown in several graphics in the paper (e.g. Figures 3,4,7,8 etc.). The initial time is at lowest height and the end time is approximately the highest height. The approximate start time of the cloud profile are shown at the top of the figures.

How vertical profiles obtained using 1 Hz data are not presented properly for averages. Nd and ref profiles using T in y axis should be weighted over similar dNd or dref intervals, and sd should be shown.

A: Since all data points are shown, the actual scatter is displayed, as well as the best fit for the relationships. This is sufficient, in our view.

Then fits should be applied because of inhomogeneity of the data collected.

A: Fits are used for calculating Na.

Why LWC profiles are not given is another issue in this paper. If you are talking about adiabatic assumption, why LWCa and LWCm are not shown?

A: Since we use the adiabatic LWC, the actual LWC has little relevance to the scientific questions of this study, because of the inhomogeneous nature of the mixing.

In supplementary plots, specifically vertical air velocity plots means nothing unless aircraft position angles are shown, and time periods during turns are taken out of the analysis? Better provide time series of segments to represent real conditions.

A: All cloud penetrations were done in straight flight segments. No turning was conducted within clouds.

Flight patterns are shown taken along the edges of Cbs where heavy mixing occurs, and suggest that data don't represent adiabatic conditions based on figure you provided; probably, because of this reason, LWCa and LWCm profiles are not likely provided.

A: The reviewer appears to miss the fundamental point that makes this study possible, that Reff remains adiabatic while LWC can be greatly diluted in the mixing process.

Also, some figures represent only 3 seconds of intervals (Fig. 14)

A: This was a 3 second pass. This is all that we could get at this altitude for tops of growing convective clouds. But this provides valid good data.

or some represent only 247 seconds (Fig. 11).

A: Figure 11 provides the full vertical profile composed of all cloud penetrations. These last two comments show that the reviewer appears not to distinguish between composites and single cloud penetrations.

Therefore, one cannot make a decision or provide objective analysis.

A: Based on our response to the previous comments, this conclusion of the reviewer is unfounded.

Also, in Fig. 11, Nd decreases with decreasing T (or Z), but ref increases with increasing Z. Usually reff increases with decreasing Nd (see Gultepe et al 1996). In this figure, # of points is about 247 (assuming 100 m/sec and using 247 sec flight duration). Can you explain how aircraft can climb up from 20C to -15C over 4 minutes? How accurate are these measurements when aircraft makes quick returns?

A: Figure 11 is a composition of all the passes that the aircraft conducted in this cloud cluster within nearly an hour.

Na in same figure (11) is given as Nd which is 566 cm-3. But Nd is found less than 600 cm-3 at the cloud base, are you using instantaneous value here? Why not use averages ?.

A: Only maximum values near cloud base can approximate Na.

This analysis is very confusing and figs are not given properly. Weighted averages should be used not a max value or min value. In fact, Fig. 11a should show bin averages of Nd (dNd~50

cm-3) or similar ones. In that plot, you also need to show LWC-T plot. Without this info, it is not possible to make fair conclusions.

A: The figure shows the actual 1Hz data.  LWC-T has little relevance to the research question.

Clearly, sampling of data and presenting observations are not presented properly (this is also valid for Fig. 12 in which dt=5 sec and 4 sec.). Gultepe et al 1999 suggested that variability over an averaging scale is an important issue for analyzing observations.

A: Gultepe measured stratiform clouds, which require utterly different approach than convective clouds. Nevertheless, the reviewer wants us to apply Gultepe's approach, which is highly unsuitable for vertical profiling of convective clouds.

An article on aerosol effect on precip is also given by Menon and Delgenio (2007, chapter 3 of DOI: 10.1017/CBO9780511619472.005) where regional changes in precipitation, examined over India and China, were found to be related to the amount of atmospheric heating, with higher atmospheric fluxes corresponding to larger changes (positive) in precipitation, though we do not discount the influence of surface and meteorological conditions that may also lead to similar changes. This work clearly suggests that heating processes within the cloud important and that specifies precipitation and depth of convective system. I think only use of ref or Na at the cloud base can be very simplistic way to consider cloud top height changes. Therefor, authors should show relationships among ref, Nd, and LWC within the cloud system in their work.

A: Heating processes can be relevant only at much higher aerosol loads that occurred in Manaus. In addition, the heating occurs at longer time scales than the individual convective updrafts. Finally, where does the reviewer get the idea that "This work clearly suggests that heating processes within the cloud important"?

Introduction section; some observations and part of method are given here but these need to move into proper sections. Text flow is not clear and very vogue in intro and method sections. Method section needs to be improved and explained properly as suggested above. I see method section is very poorly designed, providing these equations dont help for the comprehensive analysis of observations. See below for details.

A: See below answers.

*Major/minor issues:*

*Abstract*

LN31; Is this for only pollution aerosols or all "aerosols"?

A: Not only pollution aerosols. The word "pollution" is deleted.

Define ACR….CHUVA campaign.

A: We prefer to do it in the text. It is defined at Introduction section and at Table 1.

LN 34 Provide method/assumptions first before providing results

A: It is OK to do so in an abstract, to capture the attention of the reader.

Dr is confusing for defining height? Why not use Z, or h etc for height.

A: Dr is Depth for Rain initiation. It is explained in the text.

Also Na is used commonly for aerosol # concentration, please use Nd for droplets and Na for aerosols. If adiabatic droplets used for Na then show it as Nda.

A: Our previous papers (Braga et al., 2017, Freud et al., 2011 and Freud and Rosenfeld, 2012) have named the adiabatic number of droplets as Na. We prefer to keep this nomenclature.

Rain initiation is not only microphysics issue but dynamic of the system and environment. Changing of cloud height is related heat released in each layer because of mass change. Need to discuss this later.

A: Much of the importance of this study is showing that Dr is not affected much by these other considerations. This is highlighted in the text, for example in the abstract we state:

"The results show that the height of rain initiation by collision and coalescence processes ($D_r$, in units of meters above cloud base) is linearly correlated with the number concentration of droplets ($N_d$ in cm$^{-3}$) nucleated at cloud base ($D_r \approx 5 \cdot N_d$). Additional cloud processes associated to $D_r$ such as GCCN, cloud and mixing with ambient air and other processes produce deviation of ~21% in the linear relationship, but it does not mask the clear relationship between $D_r$-$N_d$ which was also found at different regions around the globe (e.g. Israel and India)."

Provide what assumptions are used first.

A: It is OK to do so in an abstract, to capture the attention of the reader.

Measured ref was close to its adiabatic value? I understand that is possible in convective clouds but if mixing and precip occurs, it will cause to non-adiabatic conditions. In fact, size>50 micron can result in precip. Please clarify this in manuscript.

A: The reason for this is the inhomogeneous nature of the mixing, as already stated in the latter part of the abstract.

Biomass burning aerosols assuming darker and resulting heating may not be good nuclei in the convective systems anyway. Then, they do not play a role in rain initiation. Needs to be clarified with refs. Not all aerosols are good CCN and variability can be large, see above .

A: The aerosols have been documented to serve as CCN even when originated from smoke.

**1. Introduction**

LN 55; change to "cloud droplets form when humid air becomes supersaturated wrt water".

A: OK.

LN60, Rosenfeld and Woodley 2000, provide earlier reference for this, it is well known over 100 years.

A: It was first documented in situ in clouds by Rosenfeld and Woodley (2000).

LN65; Provide a reference as Gultepe and Isaac (AMS J of Climate, 1999) for Nd variability and Nd-Na relationships. This paper clearly shows Nd-Na variability in the clouds representing various cloud types, clean and dirty environments.

A: Our study does not deal with the relationships between Na and Nd. We are interested in the adiabatic Nd, which cannot be obtained from averages of Nd.

But most important it provides variability in Na versus Nd. Gultepe et al 1996 (AMS JClimate) provides Nd-Na relationships and k coefficient from various projects between rv and reff. These are earlier than references you provided, and earlier refs should be cited. k in this paper is given as 0.90 But it becomes reff=1.11rv that is ~10% larger than MVR. Clearly, variability in Na and its relation to Nd are critical for many applications and these relationships are not unique.

A: Here we focus only on convective clouds, where this variability is much smaller than in marine stratocumulus.

LN70; are formed or formed?

A: Are formed.

LN 115; top parag; I feel this belongs to Observations section.

A: The introduction has to address in some way also the observations.

LN 120: This paragraps describes the goal of the paper, needs to be provided in the end of intro.

LN125-130; again this should be in the end of intro section. This is the goal of the paper.

A: It is OK for the introduction to have a promo for the goal of the paper.

LN 130, this parag belongs to observations/data section, not here.

A: The introduction provides a bit of everything to paint a coherent picture of the issue.

LN135-145; this parag belongs to method section, not here.

A: The introduction provides a bit of everything to paint a coherent picture of the issue.

LN 147; The DWC is defined...........as the mass of drops integrated over diameter range of 75-250 micron. This is not right, as you said it is drizzle water content (DWC), and therefore it is the water content of drizzle that is obtained by integration of (Nd xMd) over the spectral range representing CDP size range. Needs to be corrected.

A: The reviewer did not point out anything that needs to be corrected. We use CCP-CIP.

What happens particles between 50 and 75 micron size range?

A: It is not counted as DWC. The first bin size from CCP-CIP is 25-75 µm. Then, as we don't know if the particle size is > 50 µm we did not use this information.

Page4; section 1.1 The scientific motivation for this study; take out "for this study".

A: OK.

I think no need for section 1.1 title, you don't have others anyway.

A: OK.

LN160; Say "The aircraft based in-situ measurements collected within convective clouds over the Amazon region of Brazil were used to study precipitation initiation related to aerosol properties".

A: We prefer the existing wording.

LN160-175 is too long; you said these before, and needs to be shortened.

A: This added description is needed, based on previous comments of reviewers.

**Section 2. Instruments; LN176**

This section should be "2. Project design and observations", briefly describe project location and flight patterns. Then go to observations, including instruments.

A: But we describe here only the instrumentation.

LN 178; 2.1 cloud particle measurements; 2.2.1 and 2.2.2 are ok, how about 2.4??? isn't that aerosols are part of cloud?

A: The comment would have been correct if it was 2.2.4. But it is 2.4.

Section 2.3; please provide synoptic info on CCS, how and where they happened exactly.

A: It is given in Figure 1a and Figure 1 at supplementary material.

**3.Method**

a) The relationship between ref and the probability of drizzle is defined???? This is not clear. You mean ref and drizzle LWC is described/analyzed? In what content?

A: "Defined" was replaced with "found"

b) That is not clear and based on what?

A: That is explained later in the paper.

Goals of c) and d); these needs to be better organized/provided

A: This defines exactly what we do.

You provided four steps here, but only 3.1 and 3.2 are given, they are not consistent with 4 steps given above. Need a road map here, confusing the way done.

A: We indicate now at each item of the method section where we describe the calculations of cloud properties in analysis.

Page 6; LN245; CWC using Eq. 2 is not clear? How probes data are used? From what channels?

A: We define the size range of cloud droplets at Table 2. This is state in the text as follow:

"The calculations of DWC, RWC, and MPWC are done in similar fashion to CWC but with different cloud probes and particle size ranges (see Table 2)."

Page 9; LN 375; Analysis of ref and D……..; analysis should be in the method section not here. You should refer the analysis for the results given here.

A: Changed to "Results of analysis..."

4.2.3 Discussion should be a section on its own, not a subsection.

I am just passing results sections because of issues with other parts.

A: Ok. This is a new section now.

Other points from Gultepe et al (1996) suggest that ref max is about 12 micron which is obtained from fssp probe similar to cdp. If CIP is used, this increases to higher values of ref (>12 micron) for precip particles, as in your paper.

A: We used only CCP-CDP or CAS-DPOL for calculating ref.

Then, critical point for cloud depth is related to values of updrafts. Droplet spectra changes although ref can be same, then this can affect D-ref or Nd relationships. This means that depth of the cloud is related to updrafts and latent heat released due to cooling/heating processes, not only cloud base max aerosols concentration or effective size. In fact your data do not

represent characteristics of convective clouds properly because flights are not from core of Cbs and dt time periods used for plots are different for collected data.

A: We did not relate cloud depth to Nd. We related Dr to Nd. This is a big difference.

**5. Conclusions**

I suggest that you provide your findings in itemized way, what are they?

A: Ok, this is done in the new version.

As you said clouds are not adiabatic identities; agree that they are usually non-adiabatic in nature. Means these linear relationships suggested by you will not hold.

A: A major point of the paper is that linear relationship between Na and Dr hold despite the mixing non adiabaticity of LWC.

Specifically convective clouds are usually non-adiabatic systems because of strong wind/turbulence effects, heating, and mixing and precip. Unfortunately, these are not discussed in the paper.

A: This is true, fortunate, and being discussed in the paper and in the conclusions.

**Figures:**

Figure 1a; conceptually this figure is wrong, droplets do not grow continuously to the top of a convective cloud. Also, based on this, your measurements are not in the cloud but in the edges. This suggests your results may not represent adiabatic conditions. You need to show LWCa and LWCm profiles for an entire flight.

A: The reviewer means Figure 1b. We have demonstrated that clouds do grow from base to cloud top, and that their reff is nearly adiabatic while their LWC is not. This is explained in the paper in great detail.

Figure using a satellite/synoptic map/image will help to visualize the system. A figure is needed.

A: We can't see how it would be more helpful than Figure 1b.

Fig. 3; like to see particle Nd, aircraft heading, and wind time series for this time period. To me 2000 cm-3 is an artifact or just aerosols.

A: These added data would not be relevant to the figure. The Nd was repeated in more than one cloud droplet probe.

Figure 4; a plot for 25 second is statistically meaningless, please provide longer data set.

A: The cloud pass lasted 25 seconds, and this is how it looked like. @5 seconds is very long for a convective cloud pass.

Figure 5a-c; why don't you say CIP images rather than CCP-CIP images.

A: Because this is the full name of the instrument.

Figure 7a; reff increases with height but LWC decreases, usually LWC increases with reff, explain (see Gultepe et al 1996).

A: Can't compare marine Sc with Cb in the Amazon.

Fig. 8; Nd~100 cm-3 but ref increases with height. No aerosol effects, needs clear explanation.

A: This is what happens if the same cloud droplets continue to grow with height by condensation.

Fig. 10; Around 1 micron, no diff among the cases; Nd more than Na at larger sizes, how that happens? Sensor issues?

A: Different instruments.

Fig. 11a; need to show same way for LWC, Nd, and Ref, otherwise no idea what is going on.

A: Nd and Ref are shown. LWC has little relevance.

Also, dt=247 second for flying from 20C to -20C, how good is the data?

A: How can the reviewer imagine that we fly vertically 6 km in 247 s through cloud?!?! This is a composite of all cloud passes in this vertical profile, which took much longer.

Fig. 15; this plot should be taken out, fit is useless based on # of points and issues stated above.

A: We don't agree, based on issues stated above with the reviewer's comments.

Fig. 16; how those lines with color are obtained? What equation is used?

A: The figure captions read: estimated cloud droplet adiabatic effective radius ($r_{ea}$) (colored lines)

Fig. 17: caption; "As continental are the convective clouds?????" Not right sentence.

A: Should be: When convective clouds are more continental.

This conceptual figure is also having issues similar to Fig. 1. Is this correct that more aerosols results in higher cloud tops? I don't think so. If stable environment exist, doesn't matter what you have as Na, especially if they are small. Other point, you have rain drops but do not fall????? Certainly they fall.

A: The figure is not intended to convey that deeper clouds develop when there are more aerosols. It is not implied in any way by the captions. The figure represents the hydrometeors

size evolution with depth that was measured with CCP-CIP in convective clouds as a function of Amazon region.

**Supplementary material;**

Fig. 6a; These figs need to show mean values over the x bins. Say 100 cm-3. Like to see time series of data for at lears 1 flight, with aircraft headings.

A: The figure does not show any means.

Profiles/time series should have LWC, Nd, and Reff, and mean/sd need to be shown for at least one flight. Providing 1 sec max values do not make sense.

A: The values of Nd, CWC, re, rea as a function of height or temperature are shown in several graphics in the paper (e.g. Figures 3,4,7,8 etc.). The initial time is at lowest height and the end time are approximately the highest height. 
[revised manuscript text omitted]